# Towards Spectroscopy: Susceptibility Clusters in Language Models

**Andrew Gordon** [1 2]  **Garrett Baker** [1 2]  **George Wang** [2]  **William Snell** [2]  **Stan van Wingerden** [2]  **Daniel Murfet** [2]

## Abstract

Spectroscopy infers the internal structure of physical systems by measuring their response to perturbations. We apply this principle to neural networks: perturbing the data distribution by upweighting a token $y$ in context $x$, we measure the model's response via susceptibilities $\chi_{xy}$, which are covariances between component-level observables and the perturbation computed over a localized Gibbs posterior via stochastic gradient Langevin dynamics (SGLD). Theoretically, we show that susceptibilities decompose as a sum over *modes* of the data distribution, explaining why tokens that follow their contexts "for similar reasons" cluster together in susceptibility space. Empirically, we apply this methodology to Pythia-14M, developing a conductance-based clustering algorithm that identifies 510 interpretable clusters ranging from grammatical patterns to code structure to mathematical notation. Comparing to sparse autoencoders, 50% of our clusters match SAE features, validating that both methods recover similar structure.

## 1. Introduction

A fundamental challenge in science is inferring the internal structure of complex systems. In deep learning this takes the form of interpretability: understanding how internal structure in neural networks underlies generalization. For physical matter, a key interpretability technique is *spectroscopy* – probing systems with perturbations and inferring structure from their responses. The insight that makes spectroscopy powerful is that complex systems have *normal modes*, characteristic patterns of internal organization. Probes that excite these modes produce strong, distinctive responses; the spectrum of responses thus maps the modes and reveals structure.

**What is the spectrum of a language model?** Following Baker et al. (2025), we take perturbations to the data distribution as probes – specifically, upweighting particular tokens $y$ in contexts $x$. The response is how the model's internal components adjust, measured by susceptibility vectors $\chi_{xy}$. And the "normal modes" are patterns in the data distribution – regularities that explain why certain tokens follow certain contexts. If the model has developed internal structure sensitive to a pattern, it will respond strongly when probed with tokens following that pattern. In this paper we study susceptibility vectors extensively for Pythia-14M (Biderman et al., 2023), using data from the Pile (Gao et al., 2020); see Figure 1.

**Our main finding is that Pythia-14M has developed specialized responses to hundreds of identifiable patterns in its training data.** We find 510 interpretable clusters in susceptibility space, ranging from linguistic structures (sentence boundaries, prepositions) to dataset-specific regularities (LaTeX, code syntax). These clusters are the spectral signature of modes: token pairs that follow similar patterns have similar susceptibility vectors, causing them to cluster together. The theoretical framework of Section 2.3 makes this precise via a decomposition of susceptibilities into mode contributions.

Our contributions:

- **We develop a clustering methodology** based on conductance that identifies clusters directly from the susceptibility data yielding 510 highly interpretable clusters (Section 3.2).

- **We provide a theoretical link between clusters and patterns** in the data distribution via the mode decomposition above (Section 2.3).

- **We compare susceptibility clusters to SAE features**, finding that 50% of clusters have a matched feature in Pythia-70M SAEs (Appendix H).

- **We show that these patterns persist at scale**: measuring conductance of Pythia-14M clusters in larger models (up to 1.4B), we find they remain coherent, indicating the clusters are genuine rather than artifacts of the small model (Section 4.4).

[1]Equal Contributor [2]Timaeus. Correspondence to: Andrew Gordon <andrew@timaeus.co>.

*Proceedings of the 43rd International Conference on Machine Learning*, Seoul, South Korea. PMLR 306, 2026. Copyright 2026 by the author(s).

In Baker et al. (2025) the discovery of internal structure in neural networks through analysis of susceptibility data was termed *structural inference*. The results of this paper demonstrate that **structural inference reveals rich, interpretable structure in Pythia-14M**: the model has internalized responses to hundreds of patterns, from basic syntax to domain-specific conventions, and this structure persists across model scales. The framework connecting susceptibilities to modes provides a principled basis for understanding how patterns in training data give rise to organization in trained models.

## 2. Background

### 2.1. Tokens

We denote by $\Sigma$ the set of tokens. All Pythia models use the GPT-NeoX tokenizer (Biderman et al., 2023), a byte-pair encoding (BPE) tokenizer with $|\Sigma| = 50,304$ tokens. We often consider a token $y \in \Sigma$ in context $x \in \Sigma^k$. When presenting these we typically only give $y$ with at most some small segment of $x$, e.g. `wa` `vel` `ength` where only the last two tokens of the context $x$ are given and we indicate the token to be predicted with a solid black outline.

### 2.2. Susceptibility Space

A *susceptibility vector* $\chi_{xy}$ is associated to a token sequence $x \in \Sigma^k$ and next token $y \in \Sigma$, where $p(y|x, w)$ is the model's predictive distribution with parameters $w$.

Given a component $C$ of the network (such as an attention head), write $W = U \times C$ for the decomposition of the parameter space into $C$ and the remaining parameters $U$. Given trained weights $w^* = (u^*, c^*)$, define the observable

$$\phi_C(w) = \delta(u - u^*)\big[L(w) - L(w^*)\big] \qquad (1)$$

where $\delta$ is a Dirac delta, $L(w) = \mathbb{E}_{xy}[\ell_{xy}(w)]$ is the population loss, and $\ell_{xy}(w) = -\log p(y|x, w)$ is the per-token loss. This observable measures variations in loss due to the component $C$ alone. The *per-token susceptibility* is

$$\chi_{xy}^C = -\text{Cov}\Big[\phi_C(w), \ \ell_{xy}(w) - L(w)\Big] \qquad (2)$$

where the covariance is taken with respect to a tempered posterior localized to a neighborhood of $w^*$; see Baker et al. (2025) for details. Intuitively, $\chi_{xy}^C$ measures the first-order response of the component $C$ to upweighting the token sequence $xy$ in the data distribution: it captures how fluctuations in $C$'s behavior covary with fluctuations in the prediction of $y$ given $x$ specifically. Aggregating over components gives the susceptibility vector

$$\chi_{xy} = \big(\chi_{xy}^{C_1}, \ldots, \chi_{xy}^{C_H}\big) \in \mathbb{R}^H \qquad (3)$$

which provides an overall response profile of the model to "fluctuations" in the data distribution. Baker et al. (2025);

Wang et al. (2025) claim that this quantity is sensitive to *how* the model computes the prediction of $y$ in context $x$.

We can use susceptibilities to relate *structure in the data* with *structure in the model*: if $\{x_i y_i\}_{i=1}^n$ is a set of token sequences, then the singular value decomposition of the $n \times H$ response matrix $X$ with rows $\chi_{x_i y_i}$ couples principal components (linear combinations of token sequences) with parts of the model. The main result of Baker et al. (2025) is that in a 3M parameter attention-only transformer this couples induction patterns (structure in the data) to the induction circuit (structure in the model). In Wang et al. (2025) this was used to study the development of the induction circuit (and other structure) over training. In short, it has been established in these works that susceptibilities can be used to study internal structure in very small language models.

### 2.3. The spectrum of a language model

Informally, the reason that $y \in \Sigma$ follows $x \in \Sigma^k$ in natural language can be attributed to various *patterns* or *modes* in the data distribution. These modes can be defined formally via singular value decomposition of the conditional distribution $q(y|x)$, viewed as a matrix indexed by contexts and continuations. Such spectral decompositions have a long history in NLP and learning theory (Anandkumar et al., 2014; Hsu et al., 2012); here we follow the formulation of Chen & Murfet (2025). We provide a simplified example that illustrates the concept in Appendix C.5.

Consider the conditional distribution $q(y|x)$ as defining a linear map from contexts to distributions over continuations. Formally, let $\mathscr{H} = L^2(\Sigma^k, q; \mathbb{R}^\Sigma)$ be the space of functions from contexts to vectors over tokens, equipped with $\langle f, g \rangle_\mathscr{H} = \int \langle f(x), g(x) \rangle_{\mathbb{R}^\Sigma} q(x)\, dx$. The conditional distribution defines an element $\mathcal{C} \in \mathscr{H}$ by $\mathcal{C}(x) = \sum_y q(y|x)\, y$. The model parameter $w$ also defines an element $\Phi(w) \in \mathscr{H}$, given by $\Phi(w)(x) := \sum_y \ell_{xy}(w) y$.

Applying SVD to $\mathcal{C}$ yields singular values $s_\alpha$ with right singular vectors $v_\alpha$ (context patterns) and left singular vectors $u_\alpha$ (continuation patterns). These define basis elements $e_{\alpha\beta}$ for $\mathscr{H}$ where $e_{\alpha\beta}(x)(y)$ is the loading of $x$ on the right singular vector $v_\alpha$ times the loading of $y$ on the left singular vector $u_\beta$. We call this the *mode basis*. For more details see Appendix C.1.

The per-token susceptibility (2) corresponds to a perturbation of the data distribution that upweights the single pair $(x, y)$: we show in Appendix C.2 that $\chi_{xy}$ is proportional to the susceptibility for the perturbation $q' = (1-\epsilon)q + \epsilon\delta_{(x,y)}$. Expanding this perturbation in the mode basis gives coefficients $s_{\alpha\beta}(xy) = e_{\alpha\beta}(x)(y)$, which we call the *propensities*. These measure the extent to which the mode pair $(\alpha, \beta)$ is responsible for $y$ following $x$ in the true distribution, and depend only on the data distribution $q$, not on the model.

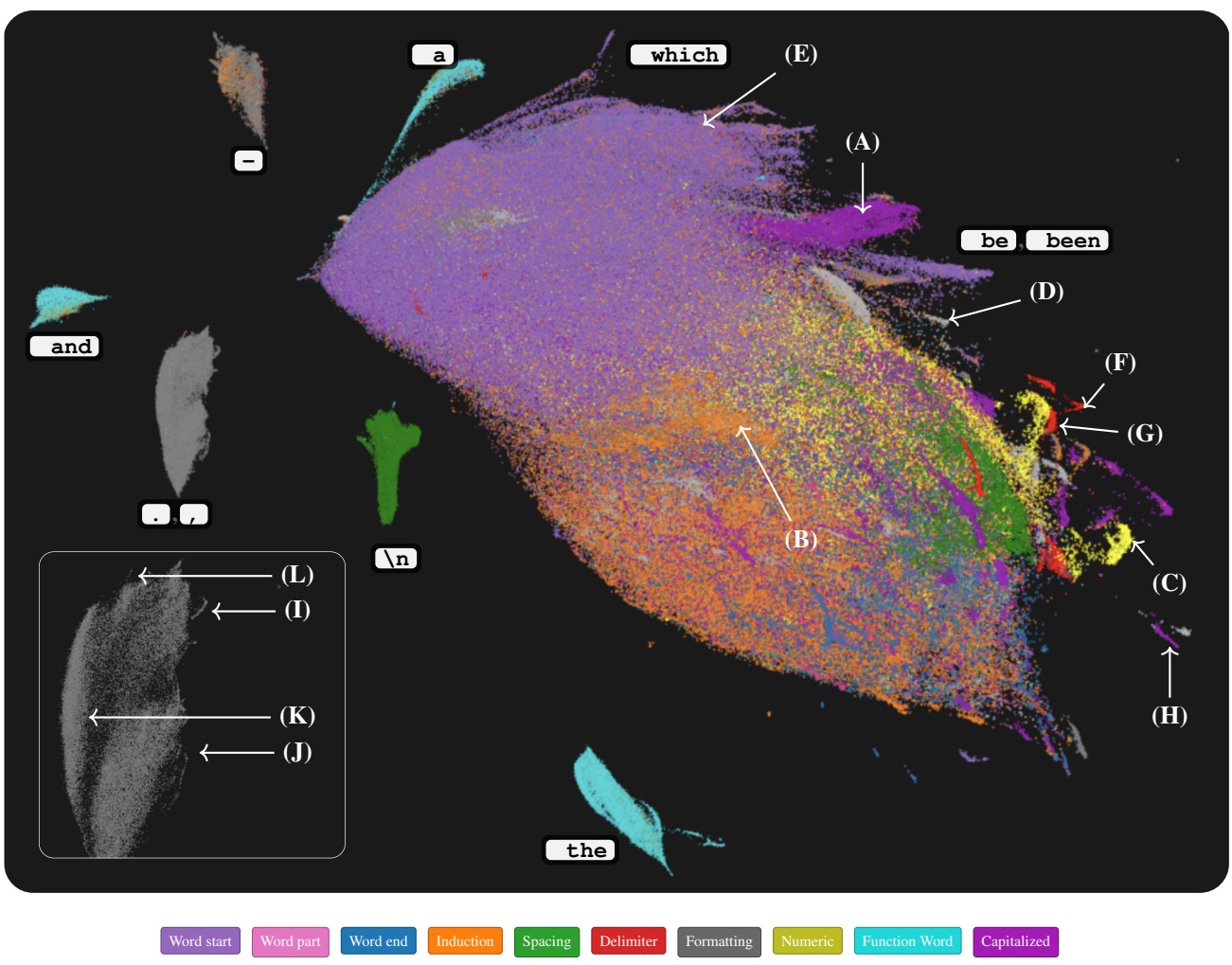

*Figure 1.* **The spectrum of Pythia-14M.** Representation using UMAP of 780,000 susceptibility vectors $\{\chi_{x_i y_i}\}_i$ computed for a 14M parameter language model. Shown is one view on a 3D point cloud. Each point represents a token $y$ in context $x$, colored by pattern type (legend above, see Table 6). Marked are some external bodies for token sequences where $y$ is a particular token and $x$ varies (e.g. there is a green body for $y =$ `\n`). Descriptions of clusters (A)-(L) can be found in Table 1. *Bottom left inset:* zoomed in view of the body of tokens `.`, `,`.

**Transform from tokens to modes**  Each mode pair $(\alpha, \beta)$ induces a characteristic response

$$\chi^C_{\alpha\beta} = -\mathrm{Cov}\Big[\phi_C(w),\ \Phi_{\alpha\beta}(w)\Big] \qquad (4)$$

where $\Phi_{\alpha\beta}(w) = \langle \Phi(w), e_{\alpha\beta}\rangle_{\mathscr{H}}$ projects the model's loss profile onto the basis element. The mode susceptibility depends on the model but not on the particular token sequence. Aggregating over components defines the vector $\chi_{\alpha\beta}$. One can show that (Lemma C.3)

$$\chi_{xy} = \sum_{\alpha,\beta} s_{\alpha\beta}(xy)\chi_{\alpha\beta} - \bar{\chi} \qquad (5)$$

where $\bar{\chi}$ is uniquely determined since $\mathbb{E}_q[\chi_{xy}] = 0$. The diagonal terms $\chi_{\alpha\alpha}$ are particularly important: they measure

the model's response to mode $\alpha$. Early in training, before the model has developed internal structure sensitive to a given pattern, we expect $\|\chi_{\alpha\alpha}\| \approx 0$. As structure specialized for that mode emerges, the response grows: $\|\chi_{\alpha\alpha}\| \gg 0$. Empirical evidence for this picture can be found in Baker et al. (2025); Wang et al. (2025).

**Similar sequences have similar susceptibilities**  The mode basis derived from $q$ provides a nontrivial sense in which $xy$ can be *similar to but distinct from* $x'y'$. Let us define

$$s(xy) := \big(s_{\alpha\beta}(xy)\big)_{\alpha,\beta}$$

in the space $P$ of propensity vectors. This map is injective: if $s(xy) = s(x'y')$ then $x = x', y = y'$. If the distinction between $xy$ and $x'y'$ is subtle we might have

*Table 1.* **A selection of clusters** from Figure 1. The clusters span natural language (A, D, E), mathematics (B, C, H, G, L), and code-like syntax (J). In the fourth column we show whether the cluster has a single shared $y$ token. For a complete list of clusters see Appendix J.1.

| Label | Description | Examples | Single | Cluster |
|---|---|---|---|---|
| (A) | Sentence starters | `. Finally`, `. Thus` | No | **C180**, **280**, **305** |
| (B) | Variable names in mathematics | `s ** 2 , 3 * t` | No | **C51**, **58**, **72** |
| (C) | Exponent 2 | `a ** 2`, `b ** 2` | Yes | **C148** |
| (D) | Enclitics | `I 'm`, `I 've` | No | **C238** |
| (E) | Logical implication | `thus`, `therefore` | No | **C424** |
| (F) | Left brackets in multiple choice | `( a )`, `( b )` | Yes | **C241**, **316** |
| (G) | Left brackets in functions | `p ( n )`, `w ( y )` | Yes | **C81**, **95**, **349** |
| (H) | True and False | `True`, `False` | No | **C250** |
| (I) | Pointed abbreviations | `U . S .` | Yes | **C308**, **474** |
| (J) | Comma in list of email addresses | `ECT @ ECT ,` | Yes | **C154** |
| (K) | End of sentence in general language | `as Trustee .` | Yes | - |
| (L) | End of sentence in math | `0 = g * d .` | Yes | **C94** |

$s_{\alpha\beta}(xy) \approx s_{\alpha\beta}(x'y')$ for all modes $\alpha, \beta < \gamma$ below some cutoff (arranging modes by singular value) with the distinction lying in some mode $\geq \gamma$. Thus $s(xy) \approx s(x'y')$ if $y$ follows $x$ for reasons similar to why $y'$ follows $x'$.

If we let $\iota : P \longrightarrow \mathbb{R}^H$ be the linear transformation sending the basis element corresponding to $\alpha, \beta$ to $\chi_{\alpha\beta}$ then (5) says $\chi_{xy} = \iota(s(xy)) - \bar{\chi}$. Thus if $y$ follows $x$ for reasons similar to why $y'$ follows $x'$ in the data distribution, we will have $s(xy) \approx s(x'y')$ (note that this has nothing to do with the model) and thus $\chi_{xy} \approx \chi_{x'y'}$. In the other direction, if the distinction between $xy$ and $x'y'$ lies only in some mode $\gamma$ the model doesn't "understand" (so $\chi_{\gamma\gamma} \approx 0$) then we can have $\chi_{xy} \approx \chi_{x'y'}$. The model *separates token sequences in susceptibility space according to the modes it understands*.

**Clusters as spectral lines.** This analysis clarifies the connection between clusters and the spectroscopy framing. In physical spectroscopy, a *spectral line* is a strong, consistent response observed across a family of probes. Analogously, when we observe a cluster of token sequences $\{x_i y_i\}$ the decomposition (5) provides a candidate explanation: these sequences share strong propensities $s_{\alpha\alpha}(x_i y_i)$ on some mode $\alpha$, and the model has developed a strong response $\|\chi_{\alpha\alpha}\| \gg 0$ to that mode. The cluster is the observable signature of internal structure specialized for pattern $\alpha$.

In conclusion: the $\chi_{\alpha\beta}$ measure the fundamental response of the model to mode pairs in the data distribution. These quantities are not directly observable since computing the modes would require the full conditional distribution $q(y|x)$, which is unknown. Instead, we observe the responses $\chi_{xy}$ and hypothesize that clusters correspond to dominant modes. The theory provides a principled explanation for why clustering should reveal meaningful structure in a neural network.

## 3. Methodology

### 3.1. Data and susceptibilities

We sample 60,000 token sequences from each of 13 Pile subsets (Gao et al., 2020), yielding 780,000 context-continuation pairs $(x, y)$ for analysis. The subsets span diverse domains including code (`github-code`), scientific text (`arxiv`, `pubmed_central`), legal documents (`freelaw`), and general web text (`pile-cc`); the full list is given in Appendix E.

For each token sequence, we compute susceptibility vectors following Baker et al. (2025). Susceptibilities are estimated via preconditioned Stochastic Gradient Langevin Dynamics (pSGLD), which samples from a localized posterior centered at the trained model checkpoint. For each component $C$, the per-token susceptibility $\chi_{xy}^C$ is computed as a covariance between the component's observable and the per-token loss, estimated from posterior samples. Aggregating across all $H$ components yields the susceptibility vector $\chi_{xy} \in \mathbb{R}^H$. Full hyperparameter settings are given in Appendix B.2.

**Visualization.** We visualize the high-dimensional susceptibility vectors $\{\chi_{x_i y_i}\} \subseteq \mathbb{R}^H$ using UMAP, a standard dimensionality reduction technique. In this paper, the role of UMAP is purely illustrative: no claims are based solely on examining the UMAP. It serves as a visual device for communicating quantitative insights that are derived and supported by other means (principally, the conductance-based clustering of Section 3.2). While it is reasonable to question the faithfulness of UMAP embeddings, this concern has no bearing on our main results.

## 3.2. Clustering

To identify interpretable clusters in the high-dimensional susceptibility data, we adapted a PageRank-based local clustering algorithm from Andersen et al. (2009) to an iterated setting.

**Preprocessing.** The susceptibility matrix is standardized column-wise (zero mean, unit variance) and then row-wise (zero mean). The row standardization removes the similarities between different component susceptibilities that would otherwise dominate Euclidean distances.

**Graph construction.** We represent the preprocessed susceptibility data as a symmetrized $k$-nearest-neighbor graph, where each node corresponds to a token sequence $xy$ and edges connect nearby points in $\mathbb{R}^H$. Edge weights use a self-tuning radial basis function that adapts to local density variations, ensuring meaningful connectivity.

**Local clustering via conductance.** The *conductance* of a vertex subset $S$ measures how well-separated it is from the rest of the graph:

$$\text{Cond}(S) = \frac{\text{(total degree of edges leaving } S)}{\text{(total degree of } S)}$$

Low-conductance sets have many internal connections but few external ones: precisely our intuition for a cluster. Following Andersen et al. (2009), given a seed node, we rank all nodes by their personal-PageRank score and search for a minimum-conductance prefix of this ranking.

**Iterative discovery.** Random seed nodes are usually in the dense main body of the data cloud, not a cluster. We therefore apply the algorithm iteratively: after each clustering attempt, we remove the discovered nodes from consideration. Early iterations typically encounter portions of the main body (and reject as having too high a conductance). As these are progressively excluded, the algorithm identifies the genuine clusters. Full algorithmic details and parameter settings are given in Appendix B.1.

## 4. Results

### 4.1. Susceptibility clusters reflect structure, not tokens

We show susceptibility geometry reflects computational role and not merely token identity by examining two complementary phenomena: *context sensitivity* (same token, different clusters) and *functional grouping* (different tokens, same cluster).

**Context sensitivity.** As shown in the first section of Table 2, the model places the same $y$ token into disjoint clusters depending on its syntactic role. When `*` appears as a multiplication operator (e.g. in $3 * 4$), it falls into clusters **C15** and **C57**, which are dominated by mathematical

contexts. However, when `*` functions as markdown formatting (e.g. *\*italics\**), it appears in the unrelated cluster **C504**.

**Functional Grouping.** Conversely, the second section of Table 2 demonstrates that the model abstracts away specific token identities when they share a functional role. For example, the tokens `h`, `u`, `y`, `o`, and `w` are grouped into a single cluster (**C58**) specifically when they appear as variables following a multiplication operator.

### 4.2. Patterns in Pythia-14M

We now survey the patterns Pythia-14M has learned to respond to. Figure 2 shows example token-context pairs from a selection of clusters, illustrating the range of patterns: from universal linguistic structures (prepositions, sentence boundaries) to dataset-specific conventions (LaTeX math mode, HTML tags, Python syntax). Clusters also vary in their level of abstraction, from low-level syntactic patterns (a specific token in a specific position) to higher-level patterns that abstract over token identity to capture functional roles; see Appendix J.4 for discussion.

Appendix J.5 shows 30 randomly chosen examples of longer token-context pairs for the same clusters in Figure 2 as well as an evaluation of whether or not they adhere to a common theme. Based on evaluating this random sample, we estimate that the majority of clusters have more than 90% of their token-context pairs well described by a single theme.

Our clustering algorithm (Section 3.2) identifies 510 clusters in the susceptibility data, listed in Table 9. Cluster labels were generated by an LLM (Claude Opus 4.5) and manually reviewed for correctness; for auto-interpretability methods see Bills et al. (2023); Paulo et al. (2025). For a higher-level taxonomy organizing these clusters by type, see Appendix J.3.

### 4.3. Comparison to SAEs

The susceptibility clusters shown in Table 2 are recognizably similar to patterns found in sparse auto-encoder (SAE) features (Yun et al., 2021; Cunningham et al., 2024; Bricken et al., 2023). To validate that these methods find similar structure we compare susceptibility clusters in Pythia-14M (Appendix J.1) to SAE features for Pythia-70M computed in Lan et al. (2025).

We use the residual stream features from layers 2-4 and define a susceptibility cluster to *match* with an SAE feature if the feature has an unusually large activation on the $y$ tokens across the cluster (and has a low baseline activation on the $y$ tokens of a random selection of other clusters). For full details of the methodology see Appendix H. **We find that out of 510 susceptibility clusters, 259 (50.8%) have**

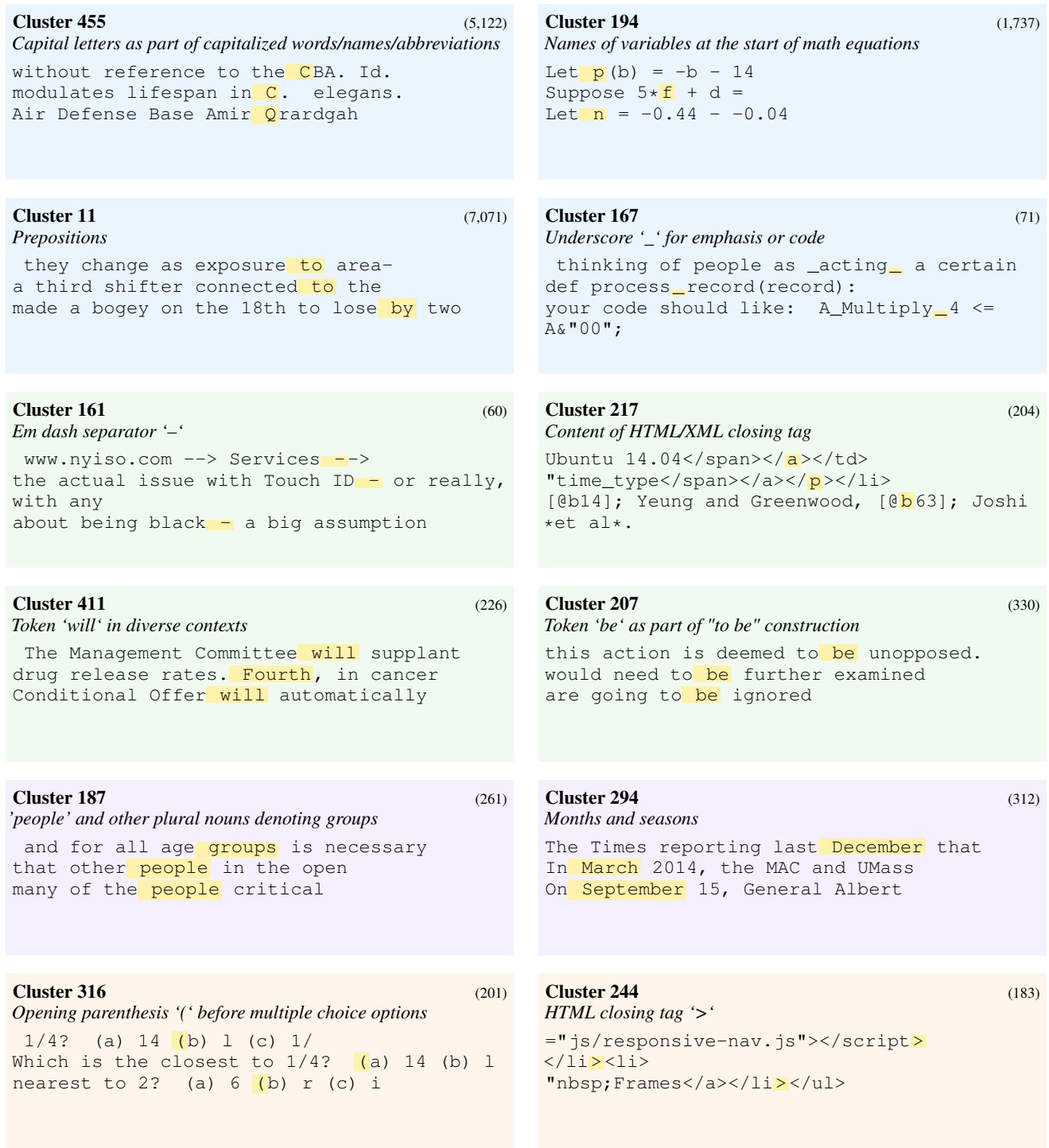

**Cluster 455** (5,122)
*Capital letters as part of capitalized words/names/abbreviations*
```
without reference to the CBA. Id.
modulates lifespan in C. elegans.
Air Defense Base Amir Qrardgah
```

**Cluster 194** (1,737)
*Names of variables at the start of math equations*
```
Let p(b) = −b − 14
Suppose 5*f + d =
Let n = −0.44 − −0.04
```

**Cluster 11** (7,071)
*Prepositions*
```
 they change as exposure to area-
a third shifter connected to the
made a bogey on the 18th to lose by two
```

**Cluster 167** (71)
*Underscore '_' for emphasis or code*
```
 thinking of people as _acting_ a certain
def process_record(record):
your code should like:  A_Multiply_4 <=
A&"00";
```

**Cluster 161** (60)
*Em dash separator '–'*
```
 www.nyiso.com --> Services --> 
the actual issue with Touch ID – or really,
with any
about being black – a big assumption
```

**Cluster 217** (204)
*Content of HTML/XML closing tag*
```
Ubuntu 14.04</a></td>
"time_type</a></p></li>
[@b14]; Yeung and Greenwood, [@b63]; Joshi
*et al*.
```

**Cluster 411** (226)
*Token 'will' in diverse contexts*
```
 The Management Committee will supplant
drug release rates. Fourth, in cancer
Conditional Offer will automatically
```

**Cluster 207** (330)
*Token 'be' as part of "to be" construction*
```
this action is deemed to be unopposed.
would need to be further examined
are going to be ignored
```

**Cluster 187** (261)
*'people' and other plural nouns denoting groups*
```
 and for all age groups is necessary
that other people in the open
many of the people critical
```

**Cluster 294** (312)
*Months and seasons*
```
The Times reporting last December that
In March 2014, the MAC and UMass
On September 15, General Albert
```

**Cluster 316** (201)
*Opening parenthesis '(' before multiple choice options*
```
 1/4?  (a) 14 (b) l (c) 1/
Which is the closest to 1/4?  (a) 14 (b) l
nearest to 2?  (a) 6 (b) r (c) i
```

**Cluster 244** (183)
*HTML closing tag '>'*
```
="js/responsive-nav.js"></script>
</li><li>
"nbsp;Frames</a></li></ul>
```

*Figure 2.* **Examples of clusters**. For each, the final token is highlighted in a selection of contexts. These clusters were selected for either having high entropy in final or penultimate tokens (blue), completely at random (green), as examples of semantic/meaning based clusters (purple), or as examples of syntactic/grammatical clusters (orange). The size of the cluster is shown in the top right.

*Table 2.* **Bidirectional abstraction over token identity.** Clusters capture functional roles rather than surface forms: (top) disparate tokens group together when serving equivalent functions; (bottom) identical tokens separate into distinct clusters based on context. This demonstrates that susceptibility vectors encode abstract syntactic/semantic categories.

| Token(s) | Context/Function | Cluster | Description |
|---|---|---|---|
| *Functional equivalence: different tokens → same cluster* | | | |
| `h`, `u`, `y`, `o`, `w` | Variables after `*` | **C58** | Multiplication operands |
| `will`, `must`, `should` | Modal auxiliaries | **C63** | Following subject noun/pronoun |
| `4`, `5`, `6`, `7` | Numeric exponents | **C143** | Following `**` operator |
| `to`, `on`, `by`, `in` | Prepositions | **C11** | Following passive verbs |
| *Context sensitivity: same token → different clusters* | | | |
| `*` | Math operator | **C15**, **57** | Multiplication in expressions |
| | Formatting | **C504** | Markdown emphasis marker |
| `/` | Division | **C224** | Fraction bar in math |
| | Date separator | **C3** | YYYY/MM/DD format |
| | Path/URL | **C322** | Directory or domain separator |
| `(` | Function call | **C81** | Opening arguments: $f(x)$ |
| | Enumeration | **C241** | List items: $(a)$ |

**a matched SAE feature**.

We caution against overinterpreting the match rate. SAE features are sparse but not singleton – multiple features can fire on any given input, reflecting the hypothesis that inputs are explained by combinations of features. By contrast, our clustering assigns each token pair to just one cluster. A token pair explained by multiple patterns may match an SAE feature for only one of those patterns, or may be assigned to a cluster capturing a different pattern. The comparison validates cross-method consistency but does not establish a one-to-one correspondence between the two representations.

While our quantitative comparison is to the Pythia-70M SAEs of Lan et al. (2025), we note that there is also qualitative similarity to the features in Bricken et al. (2023) which studies a one-layer transformer with a 512-neuron transformer MLP. The 4096 features (A/1) released there closely resemble the relatively low-level syntactic and structural features of language captured by our clusters with many "flavors" of `.`, `,`, `\n`, `and`, `of` and parts of mathematics and code.

### 4.4. Scaling to larger models

A natural question is whether the structure we identify in Pythia-14M also exists in larger Pythia models. To test this, we measured the conductance of Pythia-14M clusters when the graph structure is defined by susceptibilities from larger models. If a cluster corresponds to a genuine pattern, tokens in that cluster should remain nearby in susceptibility space even when susceptibilities are computed from a different model. We find this is indeed the case: Pythia-14M clusters have conductance significantly below 1 in all larger

models tested, from 31M to 1.4B parameters (Figure 4 in Appendix F). Random vertex sets, by contrast, have conductance $\approx 1$. This provides evidence that the clusters in Pythia-14M reflect structure in the data that larger models also learn to respond to.

Clustering on larger models yields fewer clusters (241–358 compared to 510 for Pythia-14M; see Appendix F). We hypothesize multiple causes: difficulty sampling higher-dimensional parameter spaces, noisier Euclidean metrics in higher dimensions, and the fact that larger models respond to more modes may all contribute to diffusing the cluster structure. Techniques like dictionary learning on susceptibility vectors may be necessary to recover more structure in larger models.

## 5. Related work

**Prior susceptibility work**   Susceptibility analysis was introduced by Baker et al. (2025) and applied to study development over training by Wang et al. (2025), both focusing on a 3M parameter attention-only transformer. Related ideas appear in the Bayesian influence function of Kreer et al. (2025) and the loss kernel of Adam et al. (2025), which also use SGLD sampling to compute covariances for data attribution and interpretability. The present paper extends this work in several directions: we develop the theoretical connection between susceptibilities and modes of the data distribution (Section 2.3), introduce a systematic clustering methodology based on conductance (Section 3.2), and apply these tools to Pythia-14M, which despite being only modestly larger exhibits significantly richer structure – yielding 510 interpretable clusters compared to the handful of patterns

identified in prior work.

**SAEs** Yun et al. (2021) apply dictionary learning to the activations of a 12-layer BERT model and find transformer factors that include low-level word features but also more complex high-level features involving multiple distinct tokens. Cunningham et al. (2024) mainly study Pythia-70M and Pythia-410M but in the paper do not give details on the SAE features discovered. Marks et al. (2025) make use of the clustering technique (based on gradients) from Michaud et al. (2023) and then use SAE features evaluated on these clusters to discover circuits. It is known that SAE features capture high level cross-lingual representations of grammatical concepts such as *plural* and *past tense* (Brinkmann et al., 2025). We note that our clusters are derived from 780,000 token sequences whereas e.g. the SAEs in Bricken et al. (2023) were trained with 8B data points.

**Quanta vs modes** The theoretical contribution of this paper is organized around *modes* of Chen & Murfet (2025). A very similar idea is the *quanta* of Michaud et al. (2023) which are defined in terms of the singular value decomposition of a matrix whose rows are gradients of the loss on a set of tokens in context $xy$. Spectral clustering based on this data is related to clustering by susceptibility vectors (in the sense that if the loss landscape were nondegenerate, there would be a formal relationship between them). While Michaud et al. (2023) do not provide the full set of clusters they do note that "most clusters involve the prediction of the same token" (i.e. $y$ in our notation). There are interesting exceptions, including newlines in length limited text (our **C465**).

**Latent concepts** Given that deep neural networks learn representations, it is natural to suppose that the activations of neural language models should group words together in high-dimensional space based on syntactic and semantic relationships, in such a way that clusters represent latent concepts (Mikolov et al., 2013; Reif et al., 2019; Hewitt & Manning, 2019). Dalvi et al. (2022) find 1000 clusters based on activations in a BERT model (110M parameters) and they give 183 labeled clusters in Dalvi et al. (2022, Appendix B.3). For a survey of methods before SAEs see Sajjad et al. (2022).

**Limitations** Our study focuses exclusively on the Pythia model family trained on the Pile, so our findings may not generalize to other architectures or training distributions. The clustering results depend on hyperparameter choices (conductance threshold, $k$-nearest neighbors); while we find the results robust to reasonable variations, different settings could yield different cluster boundaries. Our SAE comparison uses features from Pythia-70M rather than Pythia-14M due to availability, which may affect the match rate. Finally, while we provide evidence that Pythia-14M clusters persist in larger models (Section 4.4), susceptibility analysis has

not yet been demonstrated at frontier scale.

## 6. Conclusion

This paper establishes susceptibility analysis as a form of *spectroscopy* for neural networks: a method for inferring internal structure from how the model responds to patterns in the data distribution. The 510 interpretable clusters we identify in Pythia-14M – 50.8% of which match SAE features – reflect the model's learned responses to regularities ranging from universal linguistic structures to dataset-specific patterns. The theoretical framework (Section 2.3) provides a principled connection between these patterns, formalized as modes of the data distribution, and the model's internal responses.

Beyond interpretation, susceptibilities may also enable *intervention*: if susceptibilities measure the first-order response of structural coordinates to shifts in the data distribution, this relationship can be inverted to find data modifications that produce desired structural changes (Wang & Murfet, 2026). We leave exploration of this direction to future work.

Our detailed results focus on Pythia-14M, but we provide evidence that the patterns identified persist in larger models. The hundreds of interpretable clusters we find, and the theoretical framework connecting them to modes of the data distribution, suggest that spectroscopy may be a useful complement to existing interpretability methods.

## Impact Statement

This paper presents work whose goal is to advance the field of Machine Learning. There are many potential societal consequences of our work, none which we feel must be specifically highlighted here.

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

## Appendix Overview

The appendix provides supplementary material organized as follows:

- **Appendix A:** Table of notation used

- **Appendix B.1–Appendix B.2:** Details of the clustering algorithm and susceptibility hyperparameters.

- **Appendix C.1–Appendix C.5:** Theoretical material on susceptibilities, including the mode decomposition, per-token susceptibilities as data perturbations, per-pattern susceptibilities, and a toy model illustrating mode structure.

- **Appendix D:** Definitions of token pattern categories, with distribution across datasets.

- **Appendix E:** UMAP methodology and hyperparameters.

- **Appendix F:** Results from applying susceptibility analysis to larger Pythia models (31M–1.4B parameters).

- **Appendix G:** Additional examples of linked cluster networks including code blocks, mathematical reasoning, and LaTeX typesetting.

- **Appendix H:** Methodology for comparing susceptibility clusters to SAE features.

- **Appendix I:** Comparison of susceptibilities to a simpler Gaussian baseline.

- **Appendix J.1–Appendix J.4:** Complete list of 510 Pythia-14M clusters, taxonomy, and examples.

# A. Table of Notation

We include for convenience a list of the most important notation used throughout the paper.

*Table 3.* Common Notation.

| Notation | Meaning |
|---|---|
| $\Sigma$ | Token vocabulary |
| $(x, y)$ | Context–continuation pair |
| $q$ | True data distribution over context-token pairs |
| $p(-, w)$ | Model predictive distribution with parameters $w$ |
| $W$ | Model parameter space |
| $w^*$ | Trained parameter value |
| $C$ | A model component, e.g. an attention head |
| $\ell_{xy}(w)$ | Per-token loss $-\log p(y \mid x, w)$ |
| $L(w)$ | Population loss $\mathbb{E}_{q(x,y)}[\ell_{xy}(w)]$ |
| $L_n(w)$ | Empirical loss on dataset $D_n$ |
| $\phi(w)$ | Prior density over parameters |
| $\gamma$ | Localization strength around $w^*$ |
| $n\beta$ | Effective inverse temperature |
| $\epsilon$ | SGLD step size |
| $p_n^\beta(w)$ | Tempered posterior distribution over weights |
| $\phi_C(w)$ | Observable localized on component $C$ |
| $\chi_{xy}^C$ | Susceptibility of component $C$ to token pair $(x, y)$ |
| $\chi_{xy}$ | Susceptibility vector for $(x, y)$ |
| $\mathcal{H}$ | Hilbert space of next token functions |
| $s_\alpha$ | Singular value of the conditional distribution operator |
| $v_\alpha$ | Right singular vector / context mode |
| $u_\alpha$ | Left singular vector / continuation mode |
| $\Lambda, \Lambda^+$ | Index set for modes, positive modes |
| $e_{\alpha\beta}$ | Mode-basis element indexed by $(\alpha, \beta)$ |
| $s_{\alpha\beta}(xy)$ | Propensity of token pair $(x, y)$ for mode pair $(\alpha, \beta)$ |

# B. Experiment Details

## B.1. Details of the clustering algorithm

### B.1.1. GRAPH CONSTRUCTION

The susceptibility matrix $X \in \mathbb{R}^{n \times H}$, has different scales across columns, as well high degree of correlation between columns, both of which make Euclidean distance in $n$-dimensional space a poor basis for clustering. We preprocess it by standardizing each column to have zero mean and unit variance, then shifting each row to have zero mean.

We then construct a symmetrized $k$-nearest-neighbor graph with $k = 45$. Edges are weighted using a self-tuning radial basis function following Zelnik-Manor & Perona (2005). For two connected points $x$ and $y$, the edge weight is

$$w(x, y) = \exp\left(-\frac{\|x - y\|^2}{\sigma_x \cdot \sigma_y}\right)$$

where $\sigma_x$ is the distance from $x$ to its $k$-th nearest neighbor.

This self-tuning accounts for the increased distance between points in higher dimensions, and enforces a roughly equal average degree for distance graphs of datasets of varying dimensions.

### B.1.2. CONDUCTANCE AND LOCAL CLUSTERING

For a proper subset $S$ of vertices, the *conductance* is defined as

$$\text{Cond}(S) = \frac{w(S, \bar{S})}{\min(\text{vol}(S), \text{vol}(\bar{S}))}$$

where $w(S, \bar{S})$ is the total weight of edges crossing the boundary and $\text{vol}(S)$ is the sum of degrees within $S$. Conductance lies in $[0, 1]$, with low values indicating well-separated clusters.

The Andersen-Chung-Lang (ACL) Local clustering algorithm (Andersen et al., 2009) identifies low conductance sets containing a given seed by ordering all points in the graph with personal PageRank of that seed, and identifying low conductance sets among the prefixes of that ordering.

The personal PageRank of a seed, with teleport probability $\alpha$, is the stable state of a random walk on the graph which, at any step, returns to the seed with probability $\alpha$. Though this PageRank is determined by the global structure of the graph, it can be efficiently computed via the "push" approximation (Andersen et al., 2009), whose runtime does not depend on the number of points in the graph. The push approximation depends on a tolerance parameter $\epsilon$. It is designed to only explore nodes of the graph with predicted rank above $\epsilon$. We selected teleportation parameter $\alpha = 0.001$ and $\epsilon = 10^{-7}$ to cause push personal PageRank to assign nonzero rank to roughly 10k nodes.

### B.1.3. ITERATIVE CLUSTER DISCOVERY

Direct application of local clustering suffers from several drawbacks. On the one hand seeds in the dense main body produce candidate clusters encompassing most of the graph: if the seed is central in a large (i.e. much greater than 10k) point cloud of roughly uniform density, then the potential cluster will usually include every point with positive PageRank, since the visited points form a rough ball around the seed, and the volume increases faster than the weight of outgoing edges. On the other hand, if the seed is isolated, the potential cluster will be very small.

We address this with an iterative procedure (Algorithm 1) that progressively removes visited nodes. The aim is to find reasonable sized clusters while excluding excessively large clusters and isolated points.

Early seed selection almost always chooses points within a large main body, but repeated steps of the algorithm remove such points relatively quickly. After they are gone, the algorithm categorizes the remaining points into clusters

---

**Algorithm 1** Iterative Conductance-Based Clustering

---

**Require:** Susceptibility vectors $\{\chi_i\}_{i=1}^{n} \subset \mathbb{R}^H$
**Ensure:** List of clusters $\mathcal{C}$
  1: Construct $k$-NN graph $G$ with self-tuning weights
  2: $Unvisited \leftarrow \{1, \dots, n\}$
  3: $\mathcal{C} \leftarrow [\,]$
  4: **while** $|Unvisited| > 0.001 \cdot n$ **do**
  5:     Sample seed $s$ uniformly from $Unvisited$
  6:     Compute personalized PageRank $\pi$ from $s$ with parameters $(\alpha, \epsilon)$
  7:     Let $R \leftarrow \{i : \pi_i > 0\}$ be nodes with positive rank
  8:     Find minimum-conductance prefix $P$ of nodes sorted by $\pi$
  9:     **if** $|P| > 0.99 \cdot |R|$ **then**
 10:         $Unvisited \leftarrow Unvisited \setminus R$ {Main body}
 11:     **else if** $|P| < 20$ **then**
 12:         $Unvisited \leftarrow Unvisited \setminus \{s\}$ {Isolated}
 13:     **else**
 14:         Append $P$ to $\mathcal{C}$; $Unvisited \leftarrow Unvisited \setminus P$ {Cluster}
 15:     **end if**
 16: **end while**
 17: **return** $\mathcal{C}$

---

easily.

The parameter settings are summarized in Table 4.

The key insight is that early iterations almost always select seeds within the large, dense main body of the distribution. These attempts are rejected (the candidate cluster spans nearly all reachable nodes), but the rejection removes those nodes from future consideration. Once the main body is exhausted, subsequent seeds land in the peripheral structures, which the algorithm identifies as genuine low-conductance clusters.

### B.1.4. LOCAL CLUSTERING

Its steps are

1. To start, initialize *UnvisitedNodes* as a set of all nodes in the graph, *UnclusteredNodes* as an empty set, and *Clusters* as an empty list.

2. While *UnvisitedNodes* contains more than .1% of all nodes, choose a random node $x$ from the *UnvisitedNodes*.

3. Determine the potential cluster of $x$ using ACL

4. If the potential cluster contains more than 99% of all nodes given positive rank by that round PPR, then we did not find a cluster starting from $x$. Add all nodes with positive rank to *UnclusteredNodes* and remove them all from *UnvisitedNodes*. Return to step 2

5. If the potential cluster has size less than 20, then $x$ is an isolated point. Remove $x$ from *UnvisitedNodes* and return to step 2.

6. If neither of the above is true, the potential cluster found is acceptable. Append the potential cluster (as a set) as a new element to *Clusters* and remove every element in the potential cluster from *UnvisitedNodes*. Return to Step 2

### B.2. Susceptibilities hyperparameters

We compute susceptibilities similarly to Baker et al. (2025), with a few modifications to account for the increased size of the model.

Susceptibilities were computed using preconditioned Stochastic Gradient Langevin Dynamics (pSGLD) that used the RMSProp algorithm as a preconditioner. For Pythia-14m we used as hyperparameters $\gamma = 300$, $n\beta = 3$, $\varepsilon = 1e - 5$, batch size 16, 4 chains, and 100 draws, with 55 steps taken between each draw.

For the larger models, the hyperparameters are given in Table 5.

For additional details on the theory and implementation of susceptibilities used in this paper, please refer to the appendices of Baker et al. (2025).

### B.3. Compute Use

The data Pythia-14m experiments were collected using 4 compute nodes, each with 4 NVIDIA H200 GPUs. It took approximately 50 H200-hours.

Data for the other models was collected similarly, with Pythia-410m and Pythia-1.4b using 10 nodes of 4 NVIDIA H200s and the other models using 4 nodes. On these models, unlike with Pythia-14m, we did not log precise GPU-hour usage, but notes indicate that experiments on the largest model, Pythia 1.4b took 5 days of clock time to complete, which converts to 4,800 H200-hours, and that the runtime of the other models, in H200-hours, is roughly proportional to parameter count.

## C. Susceptibilities

We define the susceptibility $\chi_{xy}^{C}$ for a component $C$ of a neural network used to predict the next token $y$ given a context $x$, and explain how to think intuitively about what these scalar values mean. For full details see Baker et al. (2025).

We consider sequence models $p(y|x, w)$ that predict tokens $y \in \Sigma$ given sequences of tokens $x \in \Sigma^k$ for various $1 \le k \le K$ (called *contexts*) where $K$ is the maximum context length and $\Sigma$ is the set of tokens. The true distribution of

*Table 4.* Clustering algorithm parameters.

| Parameter | Value | Role |
|---|---|---|
| $k$ (neighbors) | 45 | Graph connectivity |
| $\alpha$ (teleport) | 0.001 | PageRank localization |
| $\epsilon$ (tolerance) | $10^{-7}$ | PPR approximation precision |
| Main body threshold | 0.99 | Reject if best prefix contains >99% of ranked nodes |
| Minimum cluster size | 20 | Reject isolated points |
| Termination threshold | 0.001 | Stop when <0.1% nodes remain |

*Table 5.* Experimental hyperparameters for each model.

| Model | $\gamma$ | $n\beta$ | $\varepsilon$ | Batch size | Chains | Draws | Steps Between Draws |
|---|---|---|---|---|---|---|---|
| Pythia-14M | 300 | 3 | 1e-5 | 16 | 4 | 100 | 55 |
| Pythia-31M | 300 | 3 | 1e-5 | 16 | 4 | 100 | 120 |
| Pythia-70M | 300 | 3 | 1e-5 | 16 | 4 | 100 | 200 |
| Pythia-160M | 300 | 3 | 1e-5 | 16 | 4 | 100 | 140 |
| Pythia-410M | 300 | 10 | 1e-5 | 16 | 4 | 100 | 160 |
| Pythia-1.4B | 300 | 10 | 1e-5 | 16 | 4 | 100 | 160 |

token sequences $(x, y)$ is denoted $q(x, y)$. The sequence models we have in mind are transformer neural networks, where $w \in W$ is the vector of weights. We set $X$ to be the disjoint union of $\Sigma^k$ over $1 \leq k \leq K$ and $Y = \Sigma$.

Given a dataset $D_n = \{(x_i, y_i)\}_{i=1}^n$, drawn i.i.d. from $q(x, y)$ we define

$$\ell_{xy}(w) = -\log p(y|x, w),$$

$$L_n(w) = \frac{1}{n} \sum_{i=1}^n \ell_{x_i y_i}(w).$$

The function $L_n(w)$ is the empirical negative log-likelihood and its average over the data distribution is denoted $L(w) = \mathbb{E}_{q(x,y)}[\ell_{xy}(w)]$. By a *component* of the neural network we mean some subset of the weights $C$ associated with a product decomposition $W = U \times C$. Given a parameter $w^* = (u^*, v^*)$ and writing $w = (u, v)$ for the decomposition of a general parameter, we define a generalized function on $W$ by

$$\phi_C(w) = \delta(u - u^*)\Big[L(w) - L(w^*)\Big] \tag{6}$$

where $\delta(u - u^*)$ is one if $u = u^*$ and zero otherwise. Given a prior $\varphi(w)$ on the parameter space, the *quenched posterior* at inverse temperature $\beta > 0$ and sample size $n$ is

$$p_n^\beta(w) = \frac{1}{Z_n^\beta} \exp\{-n\beta L(w)\}\varphi(w) \tag{7}$$

where

$$Z_n^\beta = \int \exp\{-n\beta L(w)\}\varphi(w)\,dw. \tag{8}$$

In practice, we use a *localized* version of this posterior centered at a trained parameter $w^*$, replacing $\varphi(w)$ with a Gaussian $\exp\{-\frac{\gamma}{2}\|w - w^*\|^2\}$ and the population loss with the empirical loss $L_n(w)$. This ensures sampling remains in a neighborhood of $w^*$; see Baker et al. (2025) and Appendix B.2 for details.

Given a generalized function $\phi(w)$ we define the expectation

$$\langle\phi\rangle_\beta = \int \phi(w)p_n^\beta(w)dw. \tag{9}$$

and given a function $\psi(w)$ the covariance with respect to the quenched posterior is

$$\mathrm{Cov}\left[\phi, \psi\right] = \langle\phi\,\psi\rangle_\beta - \langle\phi\rangle_\beta\langle\psi\rangle_\beta.$$

**Definition C.1.** The *per-token susceptibility* of $C$ for $(x, y) \in X \times Y$ is

$$\chi_{xy}^C := -\mathrm{Cov}\Big[\phi_C, \ell_{xy}(w) - L(w)\Big]. \tag{10}$$

### C.1. Modes

We briefly recall the framework of modes from Chen & Murfet (2025); Baker et al. (2025). Fix a finite alphabet $\Sigma$ and consider the Hilbert space $\mathscr{H} = L^2(\Sigma^k, q; \mathbb{R}^\Sigma)$ of functions from contexts $x \in \Sigma^k$ to vectors over tokens, with inner product

$$\langle f, g\rangle_{\mathscr{H}} = \int \langle f(x), g(x)\rangle q(x)\,dx.$$

The conditional distribution defines an element $\mathcal{C} \in \mathscr{H}$ by $\mathcal{C}(x) = \sum_y q(y|x)\,y$. Following Chen & Murfet (2025)

we use the inner product on $\mathscr{V} = \mathbb{R}^{\Sigma^k}$ defined on basis elements $x, x' \in \Sigma^k$ by

$$\langle x, x' \rangle_{\mathscr{V}} = q(x)^{-1} \delta_{x,x'}$$

where $\delta_{x,x'}$ is the Kronecker delta. This weighting corresponds to the standard whitening procedure for SVD of conditional distributions: contexts are weighted inversely to their frequency, ensuring the decomposition captures structure in the joint distribution rather than being dominated by frequent contexts. We denote by $\langle -, - \rangle$ the standard inner product on $\mathbb{R}^{\Sigma}$. For $x \in \Sigma^k$ let $\hat{x}^* : \mathscr{V} \to \mathbb{R}$ denote the linear functional $\hat{x}^*(-) = \langle -, x \rangle_{\mathscr{V}}$.

For $y \in \Sigma$ and $x \in \Sigma^k$, define $y \circ \hat{x}^* \in \mathscr{H}$ by $(y \circ \hat{x}^*)(x') = y \cdot \hat{x}^*(x') = y \cdot q(x)^{-1} \delta_{x,x'}$. The norm of this vector is $\|y \circ \hat{x}^*\|_{\mathscr{H}} = q(x)^{-1/2}$ so the elements $\{q(x)^{1/2} y \circ \hat{x}^*\}_{x \in \Sigma^k, y \in \Sigma}$ form an orthonormal basis of $\mathscr{H}$, which we call the *token basis*.

Applying SVD to $\mathcal{C}$ yields singular values $s_\alpha$ with right singular vectors $v_\alpha$ (context patterns) and left singular vectors $u_\alpha$ (continuation patterns). Here $\Lambda$ indexes right singular vectors, $\Lambda^+$ indexes left singular vectors for nonzero singular values, and $\Lambda^{++} \supseteq \Lambda^+$ is an extension to an orthonormal basis of $\mathbb{R}^\Sigma$. Define $e_{\alpha\beta} \in \mathscr{H}$ by $e_{\alpha\beta}(x)(y) = \hat{v}_\alpha^*(x)\langle u_\beta, y \rangle$ where $\hat{v}_\alpha^*(x) = \langle v_\alpha, x \rangle_{\mathscr{V}}$. The elements $\{e_{\alpha\beta}\}_{\alpha \in \Lambda, \beta \in \Lambda^{++}}$ form an orthonormal basis of $\mathscr{H}$, which we call the *modes basis*.

**Transfer coefficients.** The *propensity* $s_{\alpha\beta}(xy)$ is the coefficient relating these two bases:

$$s_{\alpha\beta}(xy) := \langle y \circ \hat{x}^*, e_{\alpha\beta} \rangle_{\mathscr{H}} = e_{\alpha\beta}(x)(y) = \hat{v}_\alpha^*(x) u_\beta^*(y) \tag{11}$$

where $u_\beta^*(y) = \langle u_\beta, y \rangle$. This gives the expansion

$$y \circ \hat{x}^* = \sum_{\alpha,\beta} s_{\alpha\beta}(xy) e_{\alpha\beta}$$

where sums over $\alpha, \beta$ mean $\alpha \in \Lambda$ and $\beta \in \Lambda^{++}$.

**Lemma C.2.** *As functions of $w$ we have*

$$\ell_{xy}(w) - L(w) = \sum_{\alpha,\beta} \left[ s_{\alpha\beta}(xy) - \delta_{\alpha,\beta} s_\alpha \right] \Phi_{\alpha\beta}(w)$$

*where* $\Phi(w)(x) = \sum_y \ell_{xy}(w) y$ *and* $\Phi_{\alpha\beta}(w) = \langle \Phi(w), e_{\alpha\beta} \rangle_{\mathscr{H}}$.

*Proof.* Using the expansion $y \circ \hat{x}^* = \sum_{\alpha,\beta} s_{\alpha\beta}(xy) e_{\alpha\beta}$ from (11), we repeat the calculation of Baker et al. (2025,

Lemma D.4) with $\Delta L = \ell_{xy} - L$

$$\begin{aligned}
\Delta L &= -\int (\delta_{x,x'} \delta_{y,y'} q(x)^{-1} - q(y'|x')) q(x') \\
&\quad \times \log p(y'|x', w) dx' dy' \\
&= -\int (y \circ \hat{x}^* - \mathcal{C})(x')(y') q(x') \\
&\quad \times \log p(y'|x', w) dx' dy' \\
&= -\sum_{\alpha,\gamma} \left[ s_{\alpha\gamma}(xy) - \delta_{\alpha,\gamma} s_\alpha \right] \\
&\quad \times \int e_{\alpha\gamma}(x')(y') q(x') \log p(y'|x', w) dx' dy' \\
&= \sum_{\alpha,\gamma} \left[ s_{\alpha\gamma}(xy) - \delta_{\alpha,\gamma} s_\alpha \right] \Phi_{\alpha\gamma}(w)
\end{aligned}$$

as claimed. $\qquad\square$

**Lemma C.3.** *Hence*

$$\chi_{xy} = \sum_{\alpha,\beta} s_{\alpha\beta}(xy) \chi_{\alpha\beta} - \bar{\chi} \tag{12}$$

*where $\chi_{\alpha\beta} \in \mathbb{R}^H$ is the vector with components $\chi_{\alpha\beta}^C = -\operatorname{Cov}[\phi_C, \Phi_{\alpha\beta}]$, the susceptibility of component $C$ for mode pair $(\alpha, \beta)$, and*

$$\bar{\chi} = \sum_\alpha s_\alpha \chi_{\alpha\alpha} \,.$$

**Sparsity.** The decomposition (12) is useful when the propensity profile $\{s_{\alpha\beta}(xy)\}_{\alpha,\beta}$ is sparse. This requires: (i) $x$ aligns with few context patterns $v_\alpha$, (ii) $y$ aligns with few continuation patterns $u_\beta$, and (iii) diagonal dominance, meaning $s_{\alpha\alpha}(xy)$ dominates over off-diagonal terms. This occurs when $(x, y)$ "follows for a clear reason" – a small number of patterns coherently explain why $y$ follows $x$.

### C.2. Per-token susceptibilities as data perturbations

We explain how to interpret the per-token susceptibility $\chi_{xy}$ in terms of perturbations of the data distribution. This clarifies the connection between per-token susceptibilities and the mode decomposition. Following Baker et al. (2025), the susceptibility for a perturbation $q \to q'$ of the data distribution decomposes as

$$\chi = \int q'(x, y) \chi_{xy} dx dy \,. \tag{13}$$

That is, the susceptibility is the $q'$-weighted average of per-token susceptibilities. Consider a perturbation that slightly upweights a single pair $(x_0, y_0)$

$$q' = (1 - \epsilon) q + \epsilon\, \delta_{(x_0, y_0)}$$

where $\delta_{(x_0, y_0)}$ is the point mass at $(x_0, y_0)$. Using (13):

$$\chi = (1 - \epsilon) \int q(x, y) \chi_{xy} \, dx \, dy + \epsilon \, \chi_{x_0 y_0}$$
$$= (1 - \epsilon) \, \mathbb{E}_q[\chi_{xy}] + \epsilon \, \chi_{x_0 y_0}$$
$$= \epsilon \, \chi_{x_0 y_0}$$

where we used $\mathbb{E}_q[\chi_{xy}] = 0$ (the per-token susceptibilities are centered). The per-token susceptibility $\chi_{xy}$ is proportional to the susceptibility for a perturbation that upweights the pair $(x, y)$.

### C.3. Inverting the decomposition

Using the orthonormality $\langle e_{\alpha\beta}, e_{\gamma\delta} \rangle_{\mathscr{H}} = \delta_{\alpha\gamma} \delta_{\beta\delta}$, we can invert (12). Observe

$$\int s_{\alpha\beta}(xy) q(x) \chi_{xy} \, dx \, dy$$
$$= \sum_{\gamma, \delta} \chi_{\gamma\delta} \int e_{\alpha\beta}(x)(y) \big( e_{\gamma\delta}(x)(y) - \delta_{\gamma,\delta} s_\gamma \big) q(x) \, dx \, dy$$
$$= \sum_{\gamma, \delta} \chi_{\gamma\delta} \Big[ \langle e_{\alpha\beta}, e_{\gamma\delta} \rangle_{\mathscr{H}} - \int q(x) \delta_{\gamma,\delta} s_\gamma \, dx \, dy \Big]$$
$$= \chi_{\alpha\beta} - |\Sigma| \bar{\chi}.$$

### C.4. Per-pattern susceptibilities

The mode structure of a data distribution is not known *a priori*, and one may instead work with patterns defined by interpretable token properties. Given a pattern category $\mathcal{P}$ (e.g., word-start tokens, induction patterns, spacing tokens), Wang et al. (2025) define the empirical *per-pattern susceptibility*

$$\hat{\chi}(\mathcal{P}) = \frac{1}{|\mathcal{P}|} \sum_{(x,y) \in \mathcal{P}} \chi_{xy} \tag{14}$$

where the sum is over token pairs classified as following pattern $\mathcal{P}$. This is the average susceptibility vector over tokens in the pattern class. The per-pattern susceptibility can be understood as an empirical approximation to the pure susceptibility $\chi_{\alpha\alpha}$. To see this, substitute (12) into (14):

$$\hat{\chi}(\mathcal{P}) = \sum_{\gamma, \delta} \chi_{\gamma\delta} \cdot \frac{1}{|\mathcal{P}|} \sum_{(x,y) \in \mathcal{P}} s_{\gamma\delta}(xy) - \bar{\chi}.$$

The coefficient $\frac{1}{|\mathcal{P}|} \sum_{(x,y) \in \mathcal{P}} s_{\gamma\delta}(xy)$ measures the average propensity of mode $(\gamma, \delta)$ within pattern $\mathcal{P}$. If pattern $\mathcal{P}$ is well-aligned with a mode $\alpha$ in the sense that:

(i) tokens $(x, y) \in \mathcal{P}$ have large diagonal propensity $s_{\alpha\alpha}(xy)$, and

(ii) propensities $s_{\gamma\delta}(xy)$ with $\gamma \neq \alpha$ and $\delta \neq \alpha$ are small relative to $|\mathcal{P}|$,

then $\hat{\chi}(\mathcal{P}) \approx c \cdot \chi_{\alpha\alpha} - \bar{\chi}$ for some constant $c > 0$ and with the vector $\bar{\chi}$ independent of $\mathcal{P}, \alpha$. In this case, the per-pattern susceptibility captures the same information as the pure susceptibility for the corresponding mode.

The per-pattern approach of Wang et al. (2025) uses indicator functions for pattern membership (uniform weights within each class), while the mode decomposition uses theoretically-derived weights $s_{\alpha\beta}(xy)$ from the SVD of the conditional distribution. The latter is more principled but requires knowledge of the mode structure; the former is practical when patterns are defined by interpretable token properties that happen to align with the underlying modes.

### C.5. Toy model of modes

In this section we study a simplified data distribution and the modes that it determines. As our starting point we take an observation about the clusters in Pythia-14M. The model "understands" two distinct ways that a sentence, concluded with a full stop, can be continued: with a capitalized word or a newline. The relevant clusters:

- **Capitalized clusters**: there are a number of clusters consisting of token sequences $xy$ where $x$ ends in a period token (i.e. $x = x'$ `.` for some $x'$) and the token $y$ de-tokenises to a space followed by a capital letter. In short, the clusters capture *capitalized words following full stops*. The examples: **C180** (sentence-initial `These`), **C280** (conjunctive adverbs like `However`, `Thus`, `Therefore`), **C305** (sentence-initial words like `In`, `At`, `Since`), along with related clusters **C36**, **C160**, **C351**, **C363**, and **C365** which capture other sentence-initial capitalized words after periods. Most of these appear in the UMAP near **C180** as shown in Figure 1.

- **Newline cluster**: the cluster **C189** consists mostly of `\n` following `.`. This cluster appears in the UMAP with the other `\n` tokens as shown in Figure 1.

Fix some $k > 0$ and let three distinct contexts $x_C, x_N, x_E \in \Sigma^k$ be chosen. Mathematically we make no further assumptions on these contexts, but informally, we think of these as all ending in `.` with the following distinctions:

- $x_C$ is the kind of context that is usually continued with a Capitalized word.

- $x_N$ is the kind of context that is usually continued with a Newline.

- $x_E$ can be continued Either way.

We let $y_C, y_N \in \Sigma$ denote continuations that begin with a capitalized word or a newline, respectively. In practice

of course $x_C, x_N, x_E$ and $y_C, y_N$ should be replaced by sets of tokens with their own distribution, but we treat only the simplest case (for the general idea see the collective bigrams of Chen & Murfet (2025)). Then as our conditional distributions we take

$$q(y_C|x_C) = 1, \quad q(y_N|x_C) = 0 \quad (x_C \text{ prefers capitalized})$$
$$q(y_C|x_N) = 0, \quad q(y_N|x_N) = 1 \quad (x_N \text{ prefers newline})$$
$$q(y_C|x_E) = a, \quad q(y_N|x_E) = b \quad (x_E \text{ is ambiguous})$$

where $a + b = 1$ and $a, b > 0$, with $q(x_C) = q(x_N) = q(x_E) = \frac{1}{3}$. The operator $\mathcal{C}$ is

$$M = \begin{matrix} & & x_C & x_N & x_E \\ & y_C & \\ & & \begin{pmatrix} 1 & 0 & a \\ 0 & 1 & b \end{pmatrix} \\ & y_N & \end{matrix}$$

where columns are contexts and rows are tokens. We compute the SVD of $\frac{1}{\sqrt{3}}M$ (Chen & Murfet, 2025, Remark 4.11). The eigenvalues of $MM^T$ are found from the characteristic polynomial

$$\lambda^2 - (3 - 2ab)\lambda + 2(1 - ab) = 0$$

which has discriminant $(1 - 2ab)^2$, giving $\lambda_1 = 2(1 - ab), \lambda_2 = 1$. The singular values of $\frac{1}{\sqrt{3}}M$ are

$$s_1 = \sqrt{\frac{2(1 - ab)}{3}}, \qquad s_2 = \frac{1}{\sqrt{3}}.$$

The left singular vectors (token patterns) are

$$u_1 \propto a \cdot y_C + b \cdot y_N \quad (\text{weighted average of continuations})$$
$$u_2 \propto b \cdot y_C - a \cdot y_N \quad (\text{capitalize vs. newline contrast})$$

and the right singular vectors (context patterns) are

$$v_1 \propto a \cdot x_C + b \cdot x_N + (1 - 2ab) \cdot x_E$$
$$v_2 \propto b \cdot x_C - a \cdot x_N + 0 \cdot x_E$$

The key observation is that the second mode $v_2$ has zero weight on $x_E$ regardless of the value of $a$. The ambiguous context does not participate in the discriminating mode – it has nothing to contribute to the distinction between capitalizing and not capitalizing.

In the special case $a = b = 1/2$ we have $s_1 = 1/\sqrt{2}$, $s_2 = 1/\sqrt{3}$, and $u_1 \propto y_C + y_N$, $u_2 \propto y_C - y_N$, $v_1 \propto x_C + x_N + x_E$, $v_2 \propto x_C - x_N$.

## D. Token pattern definitions

Following Baker et al. (2025); Wang et al. (2025), we classify tokens according to patterns that capture syntactic and structural properties. Table 6 defines the pattern categories

used throughout this paper, with the corresponding colors used in Figure 1 and other figures. Figure 3 shows how these patterns are distributed across the 13 Pile subsets used in our analysis; note that patterns are not mutually exclusive, so percentages need not sum to 100%.

Note the distinctions with respect to the tokenizer in Wang et al. (2025). The tokenizer used for the Pythia models contains dedicated tokens for spaces of various lengths. Let us denote by ▢ $\times n$ a token consisting of $n$ consecutive spaces. The token ID of ▢ is 209 while for $1 < n \leq 24$ the token ID of ▢ $\times n$ is $50278 - n$. In particular, the token consisting of 24 consecutive spaces (the largest number encoded by a single token) has ID 50254.

## E. UMAP

The data matrix $X$ has $l \times h$ columns (where $l$ is the number of layers and $h$ the number of heads per layer) and approximately 780,000 rows. Each row is the susceptibility vector $\chi_{xy}$ for a fixed neural network parameter $w$ where $(x, y) \sim q^l(x, y)$ as $1 \leq l \leq 13$ ranges over subsets of the Pile (Gao et al., 2020): `github-code`, `pile-cc`, `pubmed_abstracts`, `uspto_backgrounds`, `pubmed_central`, `stackexchange`, `wikipedia_en`, `freelaw`, `arxiv`, `dm_mathematics`, `enron_emails`, `hackernews`, and `nih_exporter`. We sample 60,000 token sequences from each dataset. The data matrix $X$ is standardized (that is, the columns have the mean subtracted and are rescaled to have unit standard deviation) before applying the UMAP algorithm.

### E.1. UMAP hyperparameters

The UMAP algorithm depends fundamentally on the choice of `n_neighbors` hyperparameter. The images in this paper were computed with `n_neighbors` equal to 45.

This hyperparameter governs how many nearest neighbors are taken into consideration when computing the local distances in the original embedding that the learned embedding tries to match. The value being too low can cause misleading clusters of data points in the visualization.

## F. Larger Models

We repeated this analysis on a collection of five more models with as many as 1.4B parameters (Figure 5). This list of models, as well as number of components per model and clusters found using the clustering algorithm in Appendix B.1 is shown in Table 7.

As discussed in Section 4.4, clustering performed better on Pythia-14M than on the larger models. Nonetheless the clusters found in higher models were highly coherent. They

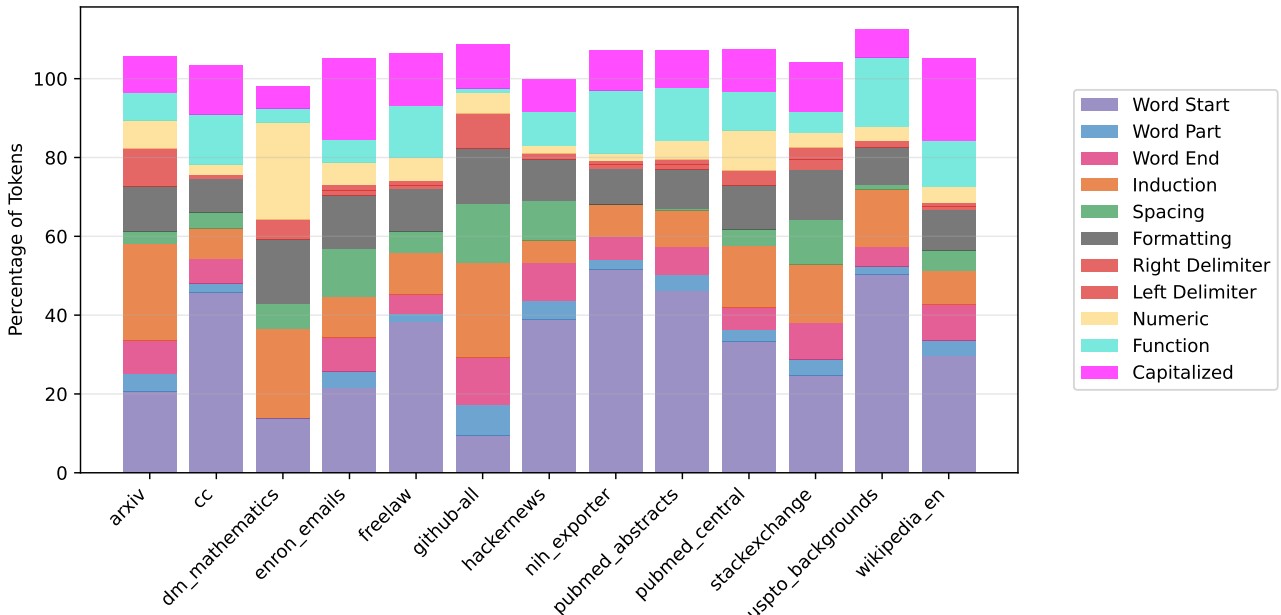

*Figure 3.* Percentages of tokens in each dataset which follow a given pattern. Note that not all patterns are mutually exclusive.

almost all corresponded to observable, interpretable patterns in the data. However, there were fewer overall clusters, and most of the patterns found were already seen in the Pythia-14M clusters. We plan to improve our data collection and analysis methods and hope to achieve similar or higher quality cluster detection models of this scale. See Appendix B.3 for details about the cost of computing susceptibilities on these models.

We took the clusters found for Pythia-14M and measured their conductance where the graph structure on the data is given by the susceptibility values found in the larger models instead, as shown in Figure 4.

## G. Linked Clusters

Modes of the data distribution (Section 2.3) are defined for contexts of a fixed length $k$, but modes for different $k$ can be *linked*. Consider

$$\overbrace{\underbrace{x_1 x_2 \cdots x_{i-1}}_{} x_i \cdots x_k}^{x_{<i}} y$$
$$\underbrace{\phantom{x_1 x_2 \cdots x_{i-1} x_i \cdots x_k}}_{x}$$

where $x_i \in \Sigma$ and $y \in \Sigma$. The reason $y$ follows $x$ may depend on both (i) why $x_i$ follows $x_{<i}$ and (ii) the presence of $x_i$ in the context. These clusters form a kind of *network* where we imagine an edge from cluster $\mathcal{C}$ to cluster $\mathcal{C}'$ if the kind of token sequences $xy$ that end up in $\mathcal{C}'$ often have contexts of the form $x = ux'y'v$ where $x'y' \in \mathcal{C}$.

In this section, we show some examples of linked clusters.

**HTML tag syntax**    Consider the token sequence

C18,151,242,506
`<`   `a`   `href`   `="`   `http`   `://`   `www`
     C149   C291   C251         C195,286   C46

representing the beginning of a HTML anchor tag, and

C190,266      C52,170,244
`</`   `a`   `>`
     C217

representing the end. To predict `</` we need to know about the early tokens that "open" the HTML tag, and to know the correct closing tag `a` we need to know about the nature of the opening tag, which itself may be predictable from the earlier context (not shown).

There is a sequence of clusters corresponding to different positions in the HTML syntax, distinguishing between the opening syntax, the tag identity, and its attributes. The sequence begins with the opening angle bracket `<` (**C18**, **151**, **242**, **506**), followed by clusters for tag names like `a`, `div`, `span`, or `li` (**C149**, **377**, **426**). If attributes are present, there are clusters for keywords like `class` (**C330**) or `href` (**C197**), followed by the assignment operator `="` (**C251**). The opening tag is completed by `>` or compound tokens (**C52**, **56**, **170**). Finally, a separate set of clusters covers the beginning of the closing tag `</` (**C190**, **266**), the content of the closing tag (e.g. `a` in `</ a >`, **C217**), and then `>` (**C52**, **170**, **244**). A very similar network of SAE features was noted in Bricken et al. (2023).

**Code Block network**    Similarly, the model has distinct

*Table 6.* Token pattern categories and their definitions. Throughout the text we apply the indicated colors to tokens that follow a particular pattern. Boxed tokens indicate the pattern being illustrated.

| Pattern | Definition | Examples |
|---|---|---|
| **Word start** | A token that decodes to a space followed by lower or upper case letters | `be`, `R` ose , `The` |
| **Word part** | A non-word-end token that decodes to upper or lower case letters | S `ne` ed , `th` at , st `em` ed |
| **Word end** | A token that decodes to upper or lower case letters followed by a formatting token, delimiter or space | el im `inate` , differe `nces` ) , al `bum` |
| **Induction** | A sequence of tokens $uvUuv$ where $U$ is any sequence, $u, v$ are individual tokens, and $uv$ is not a common bigram ($q(v|u) \leq 0.05$) | the cat ... the `cat` |
| **Spacing** | A token made up of one or more characters from space, newline, tab, carriage return, or form feed | , \n , \t , \n\n |
| **Delimiter** | Brackets and composite tokens including parentheses, brackets, and their combinations | ) , ) , ] , ); , ( |
| **Formatting** | Tokens used for document structure and formatting beyond simple spacing | . , , , // |
| **Numeric** | Tokens containing numerical digits | 123 , 14 , 2024 |
| **Function** | Function words | the , and , to |
| **Capitalized** | Capitalized words and acronyms | Denver , CBRN , Enron |

clusters related to nesting depth and scope of programming logic (specifically in C-style languages). The sequence often starts with conditional keywords like `if (` (**C97**). The opening brace `{` marks the start of the block (**C86**, **267**, **337**). The structure of the body of the code block is governed by clusters such as 4-space indent (**C106**) versus deep 8-space indent (**C17**) and more subtle spacing (**C48**, **60**, **103**, **340**, **392**, **422**, **449**).

Note that in both cases the primary role of the clusters seems to be "structural" in the sense that it is the *form* rather than the *content* of both HTML tags and code blocks that seems to attract dedicated structure.

### G.0.1. MATHEMATICAL REASONING AND TYPESETTING

**Math Problem Network**   The `dm_mathematics` dataset consists of contexts with highly repetitive question formats generated by an algorithm, and provides a rich source of patterns for language models to learn. The model appears to treat math problems as a three-act structure. The problem begins with clusters for definition key-

words like `Let` or `Suppose` (**C333, 386, 458**), followed by variable assignments like "Let $x$ be..." (**C70, 129, 194, 360, 428, 476**). The model pivots to the question phase with imperative clusters like `Calculate` (**C438, 439**) or `What is` (**C169**). The sequence concludes with clusters specialized for answer formats, such as boolean `True`/`False` tokens (**C250**).

**Multiple Choice Network**   A network of clusters seems related to the patterns involved in the rigid formatting of multiple-choice options, ensuring consistent delimiters and spacing. Options are enclosed by specific opening parentheses `(` (**C241, 316**) and closing parentheses `)` (**C37, 437**). The labels themselves form a sequence, with specific clusters for `a` (**C285**) and `b` (**C315**). A unique cluster is dedicated to the double-space separator (**C100, 264**) that visually isolates options from one another.

**LaTeX Math Mode Network**   For scientific typesetting, the model toggles between text mode and math mode using dedicated clusters. The dollar sign delimiter `$` is handled by **C49** (opening) and **C68** (closing). Within math mode,

*Table 7.* Pythia models from 14M-1.4B parameters and the results of clustering.

| Model Name | Number of Components | Number of Clusters Found |
|---|---|---|
| `pythia-14m` | 32 | 510 |
| `pythia-31m` | 56 | 358 |
| `pythia-70m` | 56 | 254 |
| `pythia-160m` | 158 | 311 |
| `pythia-410m` | 410 | 241 |
| `pythia-1.4b` | 410 | 249 |

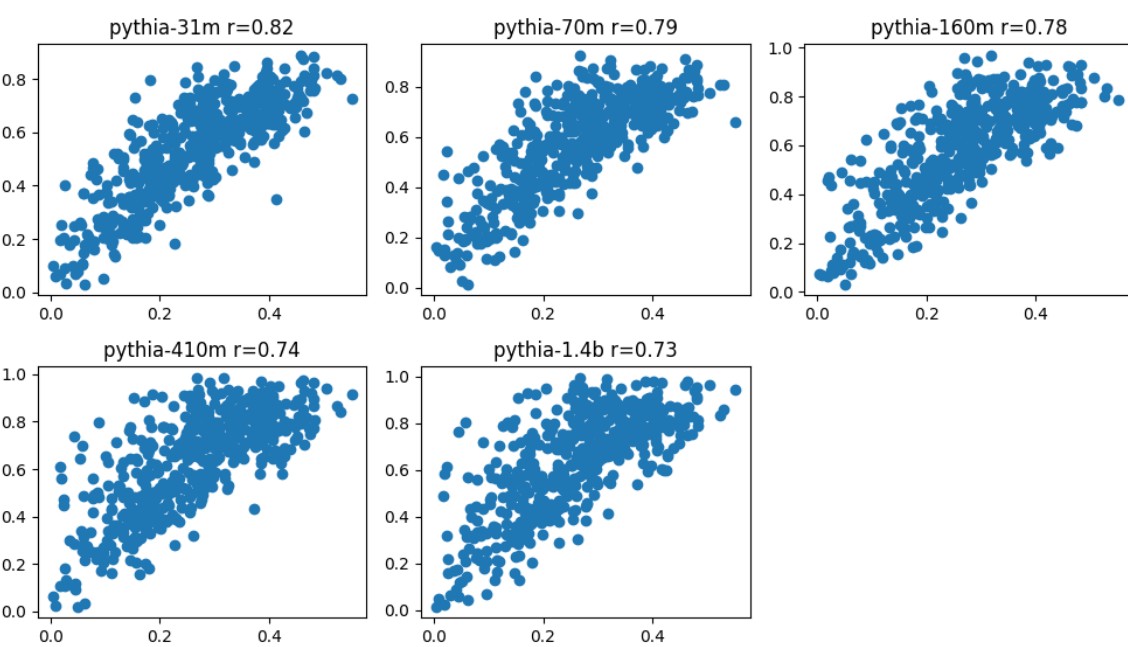

*Figure 4.* **Conductance of Pythia-14M clusters in larger models.** Each point represents one of the 510 clusters, with its conductance measured on the susceptibility distance graph of a larger Pythia model (y-axis) versus Pythia-14M (x-axis). Clusters with low conductance in both models correspond to patterns that persist across scale.

clusters track backslash commands like `\frac` (**C150**) and the mandatory opening brace `{` that follows them (**C173, 249**).

## H. SAE Comparison

To validate that our susceptibility clusters capture structure also identified by other interpretability methods, we compare them to sparse autoencoder (SAE) features. We evaluate pre-trained SAEs from Lan et al. (2025) on Pythia-70M against the token-in-context examples that populate each cluster. We use Pythia-70M because pre-trained SAEs were not available for Pythia-14M; the conductance analysis in Section 4.4 provides evidence that Pythia-14M clusters remain coherent in larger models. The SAEs were trained on the residual stream activations and are available for each layer; we use layers 2–4 (middle layers typically capture the most interpretable features, and we use three layers for redundancy).

The SAEs follow a standard architecture with pre-centering. Given a residual stream activation $a \in \mathbb{R}^d$ where $d = 512$ for Pythia-70M, the encoder produces a sparse latent representation $z \in \mathbb{R}^D$ with $D = 32768$ latent dimensions:

$$z = \mathrm{ReLU}\big(W_{\mathrm{enc}}(a - b_{\mathrm{dec}}) + b_{\mathrm{enc}}\big) \qquad (15)$$

where $W_{\mathrm{enc}} \in \mathbb{R}^{D \times d}$ is the encoder weight matrix, $b_{\mathrm{enc}} \in \mathbb{R}^D$ is the encoder bias, and $b_{\mathrm{dec}} \in \mathbb{R}^d$ is the decoder bias which serves as a pre-centering term. The decoder reconstructs the input as

$$\hat{a} = W_{\mathrm{dec}}z + b_{\mathrm{dec}} \qquad (16)$$

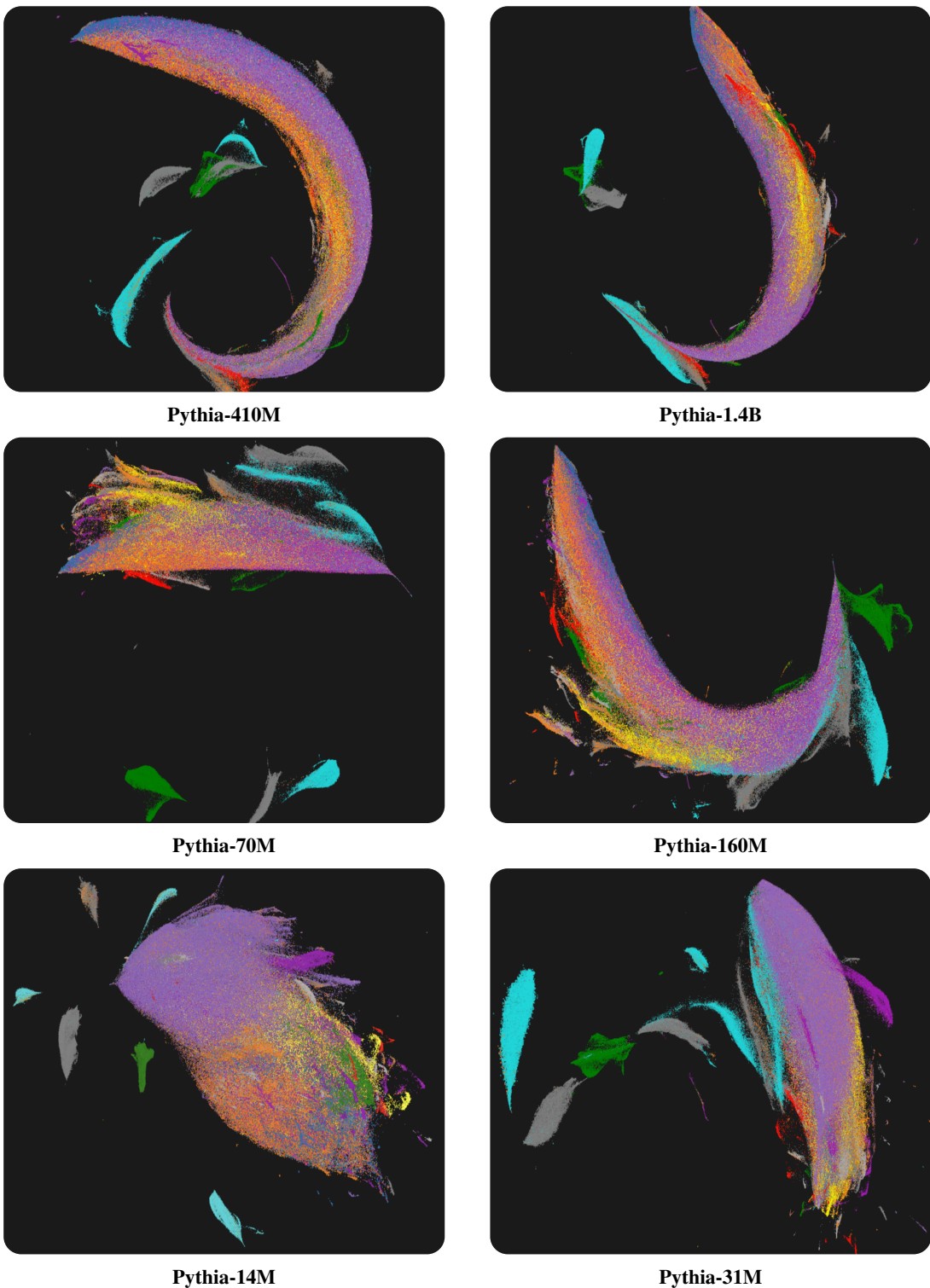

*Figure 5.* **UMAP visualizations of susceptibility vectors across the Pythia family.** Each panel shows the low-dimensional representation of susceptibility vectors for the same set of token sequences, computed from models ranging from 14M to 1.4B parameters. Points are colored by token pattern type. The overall structure – including the separation of spacing tokens, the clustering of similar patterns, and the broad organization into regions – persists across model scales, though the precise geometry varies.

where $W_{\text{dec}} \in \mathbb{R}^{d \times D}$. The ReLU activation in (15) induces sparsity in $z$, with each component $z_i$ corresponding to a learned "feature."

For each context belonging to a susceptibility cluster, we extract a window of 60 tokens centered at the target position. Specifically, if the target token $y$ appears at position $p$ in the full context, we extract tokens from position $\max(0, p - 30)$ to $\min(L, p + 30)$ where $L$ is the sequence length. We then run Pythia-70M on this window and extract residual stream activations at each position, obtaining SAE latent activations $z^{(t)} \in \mathbb{R}^D$ for each token position $t$ in the window.

To determine whether an SAE feature $i$ "matches" a susceptibility cluster, we introduce a binarized activation metric. For a given context, let $z_i^{(t)}$ denote the activation of feature $i$ at position $t$, and let $z_i^{(*)}$ denote its activation at the target token position. We compute:

1. The 90th percentile $\tau_{90}$ of $\{z_i^{(t)}\}_{t=1}^{T}$ across all positions in the window

2. An effective threshold $\tau = \max(\tau_{90}, 0.1)$

3. A binary indicator $\mathbf{1}[z_i^{(*)} \geq \tau]$

The *activation frequency* of feature $i$ for a cluster $\mathcal{C}$ is then the fraction of contexts in $\mathcal{C}$ for which this indicator equals 1:

$$\text{ActFreq}_i(\mathcal{C}) = \frac{\sum_{(x,y) \in \mathcal{C}} \mathbf{1}\big[z_i^{(*)}(x,y) \geq \tau(x,y)\big]}{|\mathcal{C}|} . \quad (17)$$

Intuitively, this measures how often feature $i$ is "unusually active" (relative to other positions in the same context) precisely at the target token. We say the SAE feature $i$ is a *match* for cluster $\mathcal{C}$ if $\text{ActFreq}_i(\mathcal{C}) \geq 0.8$.

To assess whether a matching feature is specific to a cluster rather than generically active, we compute a baseline activation frequency. For a given feature $i$ identified as matching cluster $\mathcal{C}$, we sample 20 clusters $\mathcal{C}_1, \ldots, \mathcal{C}_{20}$ uniformly at random from all clusters excluding $\mathcal{C}$. For each baseline cluster, we sample up to 30 contexts $\mathcal{C}_j' \subseteq \mathcal{C}_j$ and compute the same binarized activation measure. The baseline is then:

$$\text{Baseline}_i = \frac{\sum_{j=1}^{20} \sum_{(x,y) \in \mathcal{C}_j'} \mathbf{1}\big[z_i^{(*)}(x,y) \geq \tau(x,y)\big]}{\sum_j |\mathcal{C}_j'|}$$
$$(18)$$

A feature with high activation frequency on its matched cluster but low baseline frequency provides evidence that the SAE has learned a representation corresponding to the same pattern captured by the susceptibility cluster.

For each cluster, we analyze up to 30 sampled contexts and rank SAE features by their average activation at the target

position. From the top-ranked features, we identify the first (if any) with activation frequency $\geq 0.8$ as the candidate match, and compute its baseline.

We find that out of 510 susceptibility clusters, 259 (50.8%) have a matched SAE feature. Almost all the baseline frequencies for our matched SAE features are less than 5% (Figure 6).

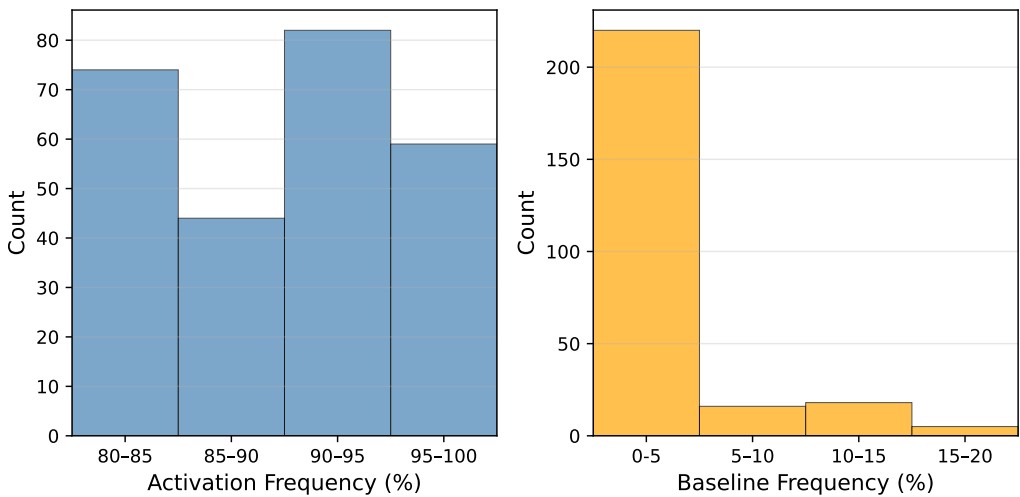

*Figure 6.* **Distribution of activation and baseline frequencies for matched SAE features**. *Left*: Activation frequencies for 259 matched features. *Right*: Baseline frequencies for the same features when measured across random clusters, demonstrating specificity with most baselines below 5%. Matched features are defined as those with activation frequency $\geq 80\%$.

Dataset: dm_mathematics

```
m ** 2 + 26 * m + 24 . Let w ( l ) = 3 * l ** 2 + 39 * l + 35 . Let o ( p ) = - 7 *
s ( p ) + 5 * w ( p ). Let g be o (- 13 ). Solve - 2 * u
```

Dataset: pubmed_abstracts

```
is otypes and specific ities . \n The EBV immortal ization technique was used to produce stable
clones , from B lymphocytes , secre ting human monoclonal antibodies to Rh ( D ), Rh ( G ),
Rh ( c ), Rh ( E ), Kell , A and A 1 blood group antigens . These clones were obtained from
```

Dataset: dm_mathematics

```
- 40 . What is c ( m ( g ))? \n - 134 19 * g ** 4 + 21 * g ** 2 - 40 \n Let r ( h ) =
4 * h . Suppose - 3 * w = - v + 2 * w + 127 44 , 4 * v - 2 *
```

Dataset: dm_mathematics

```
, - 5 * a = - l - b . Let k = 108 + l . Is k a multiple of 27 ? \n True \n Let y ( q
) = q ** 3 - 4 * q ** 2 + q + 4 . Let n be (- 100 )/ 36 - (- 4 )/(- 18
```

Dataset: dm_mathematics

```
z ). What is i ( l ( f ))? \n 2 * f ** 2 \n Let a ( y ) = - 8 * y . Let h ( i ) be
the third derivative of i ** 7 / 25 20 + i ** 4 / 12 - 2 * i ** 2 . Let v (
```

*Figure 7.* **Matching SAE for cluster C95**. Shown are five contexts from across `dm_mathematics` and `pubmed_abstracts` appearing in **C95**, with the outlined token being $y$. We show the activation of SAE feature 834 from layer 3 of Pythia-70M, which is strongly activated on these $y$ tokens. The activation ranges from $0.0$ (white) to $6.33$ (orange). The label for this cluster is: opening parenthesis `(` after function names in `dm_mathematics` algebra problems. The activation frequency of this feature on this cluster is 93.3% and baseline frequency 4.3%.

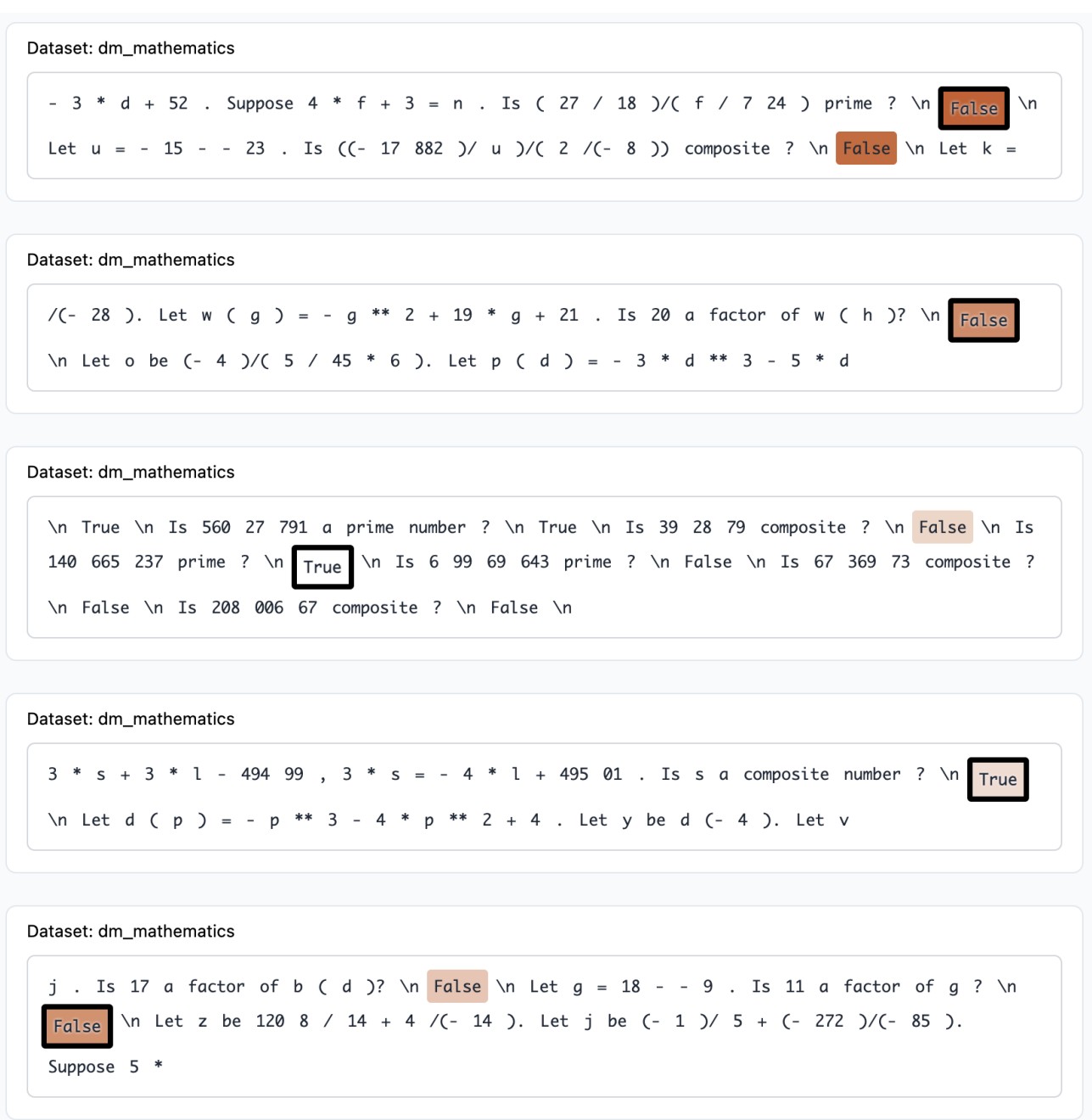

*Figure 8.* **Matching SAE for cluster C250**. Shown are five contexts from across dm_mathematics appearing in **C250**, with the outlined token being $y$. We show the activation of SAE feature 16089 from layer 4 of Pythia-70M, which is strongly activated on these $y$ tokens. The activation ranges from $0.0$ (white) to $5.95$ (orange). The label for this cluster is: `True` / `False` boolean answers after newlines in dm_mathematics math problems. The activation frequency of this feature on this cluster is 90% and baseline frequency 0.2%.

# I. Gaussian Posterior Susceptibilities

The SLT-based approach to interpretability developed in Baker et al. (2025); Wang et al. (2025) and this paper rests on a key hypothesis: that the internal structure of neural networks is encoded in the geometry of the loss landscape, and can be accessed by estimating posterior expectation values. It is therefore crucial that our methods actually probe the *posterior* – which is shaped by the loss landscape – rather than just a generic distribution around the trained weights.

In this appendix we test this by comparing susceptibilities to a simpler baseline that uses Gaussian noise instead of posterior samples. If the Gaussian baseline produced similar results, it would undermine the claim that loss landscape geometry matters; the fact that it performs substantially worse validates that the posterior structure is doing real work.

The per-token *Gaussian Posterior Susceptibility* of a component $C$ for a token context pair $(x, y)$ is

$$-\mathrm{Cov}_{N(w^*, \lambda I)} \left[ \phi_C, \ell_{xy}(w) - L(w) \right]. \tag{19}$$

This is identical to susceptibilities as usually defined, but the role of the quenched posterior has been replaced with a Gaussian distribution with uniform covariance.

Gaussian posterior susceptibilities are easily computed by modifying the experiment details in section B.2 to have $n\beta = 0$. They have a straightforward interpretation as a tool of mechanistic interpretability. Up to scale and constant term (19) measures the covariance between $\ell_{xy}$ and $L$ when the weights in component $C$ are perturbed by Gaussian noise.

We contrast the approach taken in the main text with this one by computing Gaussian posterior susceptibilities for Pythia-14M on the same set of data points, and repeating our analysis. Figure 11 shows these new susceptibilities. We observe low to moderate correlation between the two datasets, as described in Figure 9. We also observe that the component susceptibilities for Gaussian posterior susceptibilities are extremely well correlated with each other, as visible in Figure 10. This is also seen in PCA analysis. The top PC for Pythia-14M susceptibility data explains 71% of the variance, while the top PC for Gaussian posterior susceptibilities on the same dataset explains 87% of the variance.

From this, we conclude that Gaussian posterior susceptibilities are *meaningfully different* from susceptibilities done using the localized Gibbs posterior, and in many ways inferior. The high inter-component correlations suggest that the Gaussian posterior susceptibilities are less sensitive to difference between components.

This affects cluster analysis as well, since the clustering algorithm implicitly depends on the Euclidean metric on susceptibility space, having multiple highly correlated coefficients can distort distances between points and make low conductance sets harder to identify.

Running the clustering algorithm on the Gaussian posterior susceptibilities for Pythia-14M identified 288 clusters, compared to the 510 found for the normal data. There isn't a clear metric for cluster quality, but clusters found on normal susceptibility data were 72% larger on average (mean size 976 vs 567). Further the clusters for normal susceptibilities contained, on average, more diverse sets of tokens as measured by entropy. The clusters found on normal data had an average token entropy 31% higher (1.23 vs 0.94).

In Figure 11 we see that there is no separation of tokens containing double quotes into opening and closing clusters, like there is for normal susceptibilities. Nor do these tokens appear to have any particular organization in the body of the UMAP.

In Figure 12 we compare the susceptibility data with the original hyperparameters, which we refer to as $n\beta > 0$, to that for $n\beta = 0$. We perform this comparison for sequences $xy$ where $y$ contains "<" when decoded. In the $n\beta > 0$ data we see a separation into two regions (A) for `<` from opening tags (**C18**, **151**, **242**, **506**) and (B) for `</` from closing tags (**C190**, **266**). In the $n\beta = 0$ data there is a single `</` cluster (**D87**) but no `<` cluster for opening HTML tags, which is consistent with the more uniform appearance of these tokens in the susceptibility UMAP.

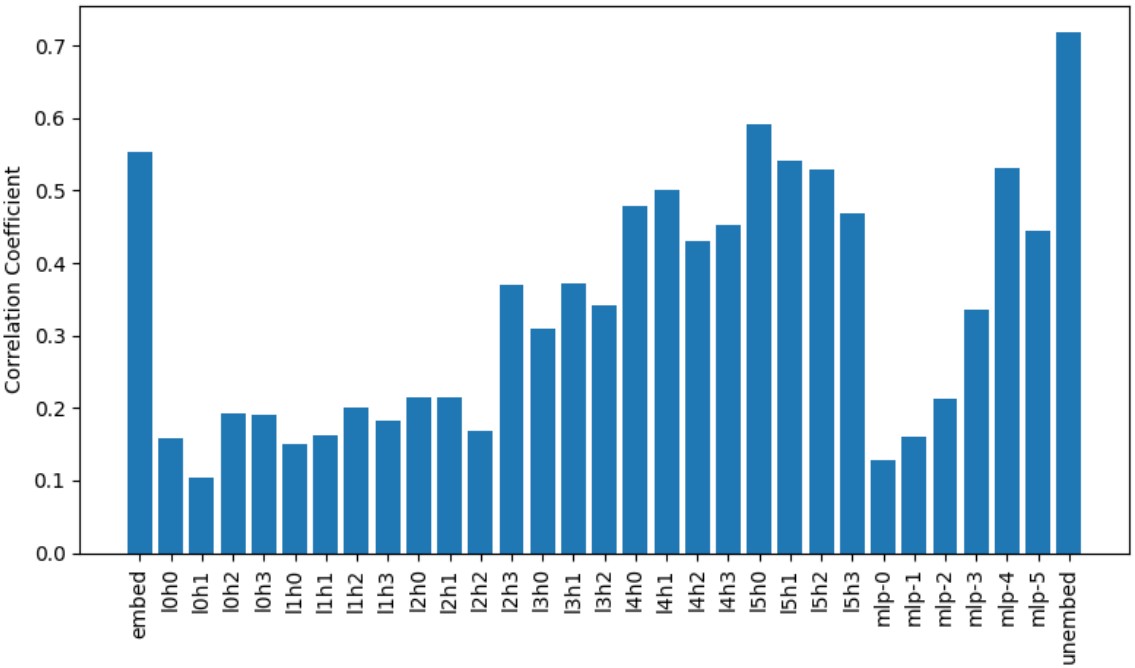

*Figure 9.* **Component-wise correlation between susceptibility types.** Pearson correlation between Gaussian posterior and usual susceptibilities, computed separately for each of the 32 components.

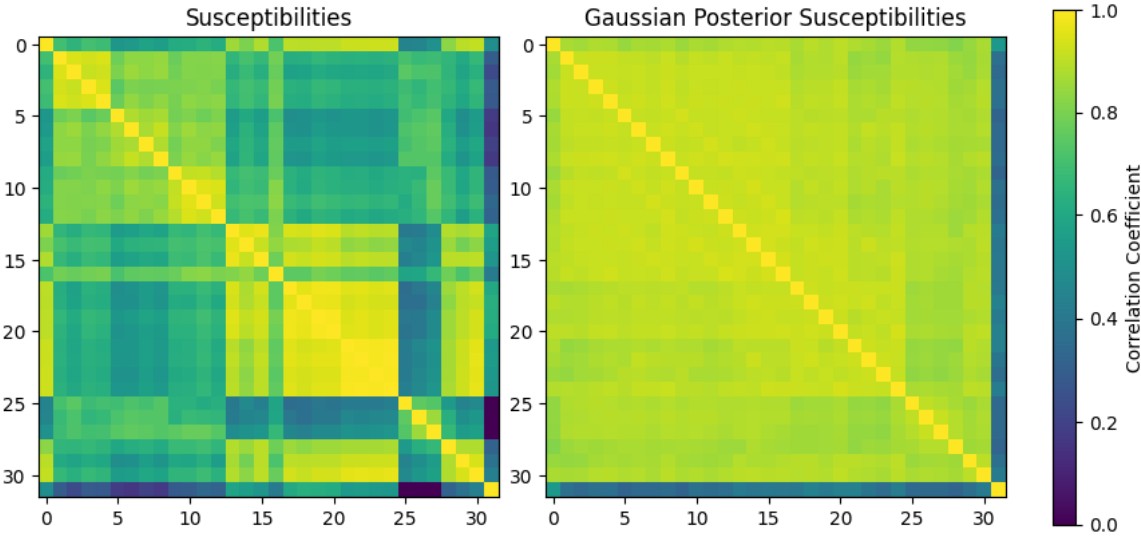

*Figure 10.* **Inter-component correlation structure.** Correlation matrices for usual susceptibilities (left) and Gaussian posterior susceptibilities (right) across the 32 components. The higher off-diagonal correlations on the right indicate that Gaussian posterior susceptibilities are less sensitive to differences between components.

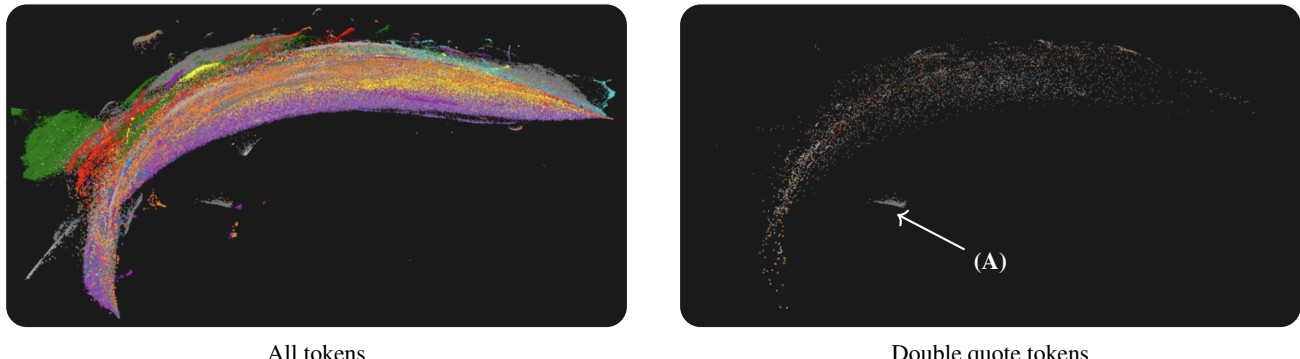

| All tokens | Double quote tokens |

*Figure 11.* **Double quote token clusters in** $n\beta = 0$ **data.** *Left:* Full low-dimensional representation of susceptibility vectors computed from Pythia-14M, with points colored by pattern type. *Right:* Filtered view showing only tokens which, when decoded, contain double quotes (e.g. `"` , `."` ). The cluster (A) consists of `="` tokens. We do not see separate clusters for `"` with the sense "opening" vs "closing" in this data.

| All tokens | Tokens containing "<" |

*Figure 12.* **Tokens containing "<" in both** $n\beta = 0$ **and** $n\beta > 0$ **data.** *First row:* data from Pythia-14M, usual hyperparameters. *Second row:* data from Pythia-14M with hyperparameter $n\beta = 0$. *Left:* Full low-dimensional representation of susceptibility vectors colored by pattern type. *Right:* Filtered view showing only tokens when decoded contain "<" (e.g. `<` , `</` ). Region (A) contains `<` from opening tags while (B) contains tokens `</` from closing HTML tags. Region (C) contains a mix of both `</` , `<` with more of the latter towards the bottom of the region and more of the former towards the top.

# J. Clusters

## J.1. Pythia-14M clusters

We divide token clusters into four main categories (and implicitly "Other"). These are

- Grammar: Syntax and grammatical patterns

- Math: Mathematical and scientific notation

- Code: Code, markup and technical syntax

- Formatting: Document structure and formatting

- Bigrams: Fixed phrases and bigrams (examples in Table 8)

For details of this taxonomy see Appendix J.3. The complete list of clusters is given in Table 9. Note that **Cx** stands for the cluster with index $1 \leq x \leq 510$ and **La-b** stands for the layer $a$ Pythia-70M SAE feature with index $b$. When an SAE feature appears in the row for a cluster it means they are a match in the sense of Appendix H.

## J.2. Bigrams and fixed phrases

The simplest example of a mode in Chen & Murfet (2025) is an "absolute" bigram, where a pair of tokens only ever occur together. Given the conjectured relation between modes and clusters we therefore expect that bigrams will appear in classifications of clusters in susceptibility UMAPs.

And indeed, the set of clusters for Pythia-14M contains numerous examples of bigrams and fixed phrases, by which we mean a cluster $\mathcal{C}$ consisting of token sequences $xy$ where $y$ is the same across almost all token sequences in the cluster, and $x = x'x''$ for some common sequence $x'' \in \Sigma^{k'}$. In Table 8 we enumerate all the bigrams and fixed phrases found among our clusters (see Appendix J.1 for the complete list).

*Table 8.* **Bigrams and fixed phrases.** A significant minority of discovered clusters correspond to rigid bigrams and idiomatic phrases where the continuation is highly deterministic. This suggests the model processes common multi-token sequences (like "United States" or "based on") as atomic units.

| Category | Example Bigram | Cluster | $y$ **Token** |
|---|---|---|---|
| **Fixed Idioms** | "United **States**" | **C373** | States |
| | "based **on**" | **C99** | on |
| | "associated **with**" | **C346**, **355** | with |
| | "such **as**" | **C236** | as |
| | "in **order**" | **C432** | order |
| | "a **lot**" | **C146** | lot |
| | "of **these**" | **C357** | these |
| | "the **following**" | **C359** | following |
| | "a **new**" | **C468** | new |
| **Conversational** | "Thank **you**" | **C457** | you |
| | "I **think**" | **C262** | think |
| | "I **have**" | **C433** | have |
| | "to **do**" | **C376** | do |
| **Math Setup** | "derivative **of**" | **C77** | of |
| | "Let [...] **be**" | **C425** | be |
| | "Collect the **terms**" | **C291** | terms |
| | "In **base**" | **C361** | base |
| | "How **many**" | **C470** | many |
| **Syntactic "The"** | "that **the**" | **C53** | the |
| | "to **the**" | **C408** | the |
| | "is **the**" | **C270** | the |
| | "be **the**" | **C364** | the |

| Cluster | Description | Category | SAE |
|---------|-------------|----------|-----|
| **C1** | Auxiliary verbs ( `is` , `have` , `has` , `are` ) after subject pronouns/nouns | Grammar | – |
| **C2** | `**` exponentiation operator after variables in `dm_mathematics` and code | Math | **L3-23541** |
| **C3** | Date separator `/` in formatted dates and paths | Formatting | – |
| **C4** | Indefinite article `a` following prepositions in technical/formal text | Grammar | **L4-6774** |
| **C5** | Underscore `_` in code/LaTeX formatting (`github-all`, `arxiv`, `stackexchange`) | Code | – |
| **C6** | Opening bracket of parenthetical in text | Formatting | **L2-7291** |
| **C7** | Minus sign in mathematical expressions, opening brackets in math and combinations of the two | Math | – |
| **C8** | Hyphen `-` after prefixes and in compound terms across scientific/technical text | Grammar | – |
| **C9** | Token `:` in "Category:" formatting in `wikipedia_en` | Formatting | **L4-22002** |
| **C10** | Single-letter variable names in `dm_mathematics` equations | Math | – |
| **C11** | Prepositions ( `to` , `on` , `by` , `in` , `for` ) after passive/descriptive phrases | Grammar | – |
| **C12** | Plural verb forms ( `are` , `have` , `were` ) after relative clauses and subjects | Grammar | – |
| **C13** | Mathematical operators and numbers following arithmetic symbols in `dm_mathematics`, `arxiv`, and `github-all` | Math | – |
| **C14** | Modal and auxiliary verbs after subjects in technical/legal text | Grammar | – |
| **C15** | Multiplication operator `*` and formatting tokens in mathematical expressions across `dm_mathematics`, `github-all`, and `arxiv` | Math | – |
| **C16** | Closing parentheses and brackets after numbers in technical/scientific text | Formatting | – |
| **C17** | Indentation spacing (8 spaces) after newlines in `github-all`, `enron_emails`, `stackexchange` | Formatting | **L2-30019** |
| **C18** | `<` XML/HTML opening bracket after whitespace in `github-all` and `stackexchange` | Code | **L2-20843** |
| **C19** | Comparison words ( `smaller` , `greater` , `bigger` ) after "is" in `dm_mathematics` | Math | **L2-2185** |
| **C20** | Closing parentheses after citations, references, and expressions in `pubmed_abstracts`, `stackexchange`, `arxiv`, `github-all`, and `pubmed_central` | | – |
| **C21** | End of text token after double newlines in `stackexchange` and `hackernews` | Formatting | **L2-3257** |
| **C22** | Double newline paragraph breaks in `cc`, `wikipedia_en`, `hackernews`, `arxiv`, and `github-all` | Formatting | – |

| Cluster | Description | Category | SAE |
|---|---|---|---|
| **C23** | `.` period after numbers in `dm_mathematics` | Formatting | – |
| **C24** | Preposition `for` after purpose/use descriptions in technical and discussion contexts | Grammar | – |
| **C25** | Token `:` in "Q:" formatting pattern in `stackexchange` question headers | Formatting | **L2-25505** |
| **C26** | Article `an` following prepositions and verbs in technical/legal text | Grammar | **L4-4098** |
| **C27** | Hyphen after `non` prefix in scientific/technical text | Formatting | **L4-32155** |
| **C28** | Prepositions after commas (`as`, `from`, `with`, `into`, `using`) | Grammar | – |
| **C29** | Token `.` playing various roles except the end of a sentence | | – |
| **C30** | `of` after nouns/phrases indicating possession, quantity, or part-whole relationships | Grammar | – |
| **C31** | Modal verb followed by `be` in technical/scientific text | Grammar | **L3-5228** |
| **C32** | Token `in` from "\in" set membership symbol in `arxiv` and `stackexchange` | Math | **L2-11825** |
| **C33** | `fig` after `="` in `pubmed_central` figure references | Code | – |
| **C34** | Pipe `\|` delimiter in `wikipedia_en` tables and formatting | Formatting | – |
| **C35** | Assignment operator after closing parenthesis in `dm_mathematics` | Math | **L4-27936** |
| **C36** | Sentence-initial capitalized words after periods in formal/technical text | Math | – |
| **C37** | Closing parenthesis `)` after function arguments in `dm_mathematics` and code, as well as closing parenthesis on multiple choice questions | Math | – |
| **C38** | Colons and semicolons following dates and times and email headers | Formatting | – |
| **C39** | `this` keyword induction in JavaScript code (`github-all`) | Code | **L4-22721** |
| **C40** | Exponents (3-5) after `**` in `dm_mathematics` polynomial expressions | Math | **L3-31787** |
| **C41** | Passive voice construction (`by` after past participles) | Grammar | – |
| **C42** | Double newline after closing brace `}` in `github-all` and `stackexchange` code | Formatting | **L2-17680** |
| **C43** | Sentence-initial tokens after periods in expository text | | – |
| **C44** | LaTeX/code closing delimiters and formatting tokens in `arxiv` and `stackexchange` | Math | – |
| **C45** | End of document/abstract in scientific and technical text (`<\|endoftext\|>` after `.` in `pubmed_abstracts`, `nih_exporter`, `uspto_backgrounds`) | Formatting | – |
| **C46** | `www` after `://` in URLs across `hackernews`, `enron_emails`, and `stackexchange` | Code | **L4-12160** |
| **C47** | Closing bracket `]` with various compounded tokens | Formatting | – |

*Continued on next page*

| Cluster | Description | Category | SAE |
|---------|-------------|----------|-----|
| **C48** | Whitespace indentation after newlines in code/technical documents | Code | – |
| **C49** | LaTeX math delimiter `$` opening a math environment in `arxiv` and `stackexchange` | Math | – |
| **C50** | Preposition `to`/`by` after "divided" or "with respect" in `dm_mathematics` and technical texts | Grammar | – |
| **C51** | Variable names `s` and `t` after `*` in `dm_mathematics` | Math | **L2-29842** |
| **C52** | HTML/XML closing angle bracket `>` after tag names in `github-all` and `stackexchange` | Code | **L2-28829** |
| **C53** | Token `the` as part of "that the" in technical/legal documents | Bigrams | **L4-18242** |
| **C54** | Equals sign `=` after numbers in `dm_mathematics` equations | Math | – |
| **C55** | Conjunction `and` after commas and list items | Grammar | **L2-1172** |
| **C56** | HTML closing angle bracket `">` after `.html` in href attributes | Code | **L3-24737** |
| **C57** | Multiplication operator `*` in `dm_mathematics` algebraic expressions | Math | **L4-23363** |
| **C58** | Variable names (`h`, `u`, `y`, `o`, `w`) after `*` in `dm_mathematics` equations | Math | – |
| **C59** | Conjunction `and` after comma in scientific/technical lists | Grammar | **L2-1172** |
| **C60** | Indentation whitespace after newlines in code/structured text | Formatting | – |
| **C61** | Newline `\n` after punctuation/delimiters in code and structured text | Code | – |
| **C62** | Whitespace and indentation continuation after newlines in `github-all` and `stackexchange` | Code | – |
| **C63** | Modal verbs (`will`, `is`, `should`, `would`, `must`) following subjects in technical/professional text | Grammar | – |
| **C64** | Prepositions after verbs/nouns (`with`, `as`, `between`, `using`) in technical/scientific text | Grammar | – |
| **C65** | Hexadecimal `00` after `0x` prefix in `github-all` | Code | – |
| **C66** | `other`, `others`, `them` after quantifiers/conjunctions in diverse technical/legal texts | Grammar | – |
| **C67** | Variable name `q` in `dm_mathematics` algebra problems | Math | **L2-29842** |
| **C68** | LaTeX math mode closing delimiter `$` in `arxiv` and `stackexchange` | Math | **L2-8199** |
| **C69** | Closing quotation marks of various kinds after quoted phrases in `freelaw` and `cc` | Grammar | – |
| **C70** | `Let` followed by `be` in `dm_mathematics` variable definitions | Math | **L3-7888** |
| **C71** | Tabs and whitespace after newlines in code/structured text | Code | – |
| **C72** | Variable names after `*` or `(` in `dm_mathematics` equations | Math | – |
| **C73** | Document/sentence start tokens after `<|endoftext|>` in `pubmed_abstracts` and `nih_exporter` | Formatting | – |

| Cluster | Description | Category | SAE |
|---|---|---|---|
| **C74** | Pronouns `his` / `he` following references to male individuals | Grammar | – |
| **C75** | Modal verb completions (`be`, `=`, `not`, `a`) following auxiliary verbs (`can`, `may`, `would`, `should`, `will`) in technical/scientific text | Grammar | – |
| **C76** | Percentage symbols and repeated zeros in scientific/technical text | Math | – |
| **C77** | Token `of` as part of the bigram "derivative of" in `dm_mathematics` calculus problems | Math | **L2-32107** |
| **C78** | Contractions and possessives after pronouns/nouns (`'s`, `'ll`, `'d`, `'ve`) | Grammar | – |
| **C79** | Capitalized words/names following punctuation or prepositions across `wikipedia_en`, `cc`, `enron_emails`, `pubmed_abstracts`, `nih_exporter` | Grammar | – |
| **C80** | Closing parentheses and brackets after expressions | Formatting | – |
| **C81** | Opening parenthesis `(` after function names in `dm_mathematics` equations | Math | – |
| **C82** | Preposition `to` and numbers after hyphens/references | Grammar | – |
| **C83** | Newlines in a variety of contexts. Sometimes the first of two newlines | Formatting | – |
| **C84** | *No description available* | – | |
| **C85** | LaTeX/math notation delimiters and escaped formatting in `arxiv` and `stackexchange` | Math | – |
| **C86** | Opening brace `{` after function/control structure in code | Code | **L2-19285** |
| **C87** | Question marks ending math problems in `dm_mathematics` | Math | **L2-5385** |
| **C88** | `get` and other common verbs after infinitive markers in `hackernews`, `cc`, `stackexchange`, `enron_emails` | Grammar | – |
| **C89** | Numeric identifiers in `pubmed_central` reference codes (e.g., `molecules-22-00821`, `pone.0196349`) | | – |
| **C90** | `HOU` after `/` in Enron email address paths | | **L4-27393** |
| **C91** | `such` after comma/preposition introducing examples in technical/legal text | Grammar | **L3-21249** |
| **C92** | Period after U.S. state abbreviations and legal citations | – | |
| **C93** | Newline after question mark or sentence end in `dm_mathematics` and structured text | Math | – |
| **C94** | Sentence-ending `.` after numbers in `dm_mathematics` and technical text | Math | – |
| **C95** | Opening parenthesis `(` after function names in `dm_mathematics` algebra problems | Formatting | **L3-834** |
| **C96** | Sentence-initial capitalized words after newlines in `dm_mathematics` | Math | – |
| **C97** | `if (` conditional statement opening in `github-all` code | Code | **L2-4280** |

*Continued on next page*

| Cluster | Description | Category | SAE |
|---|---|---|---|
| **C98** | Quantifiers and determiners after prepositions ( `two` , `different` , `we` , `three` , `many` ) | Grammar | – |
| **C99** | Token `on` from bigram "based on" in technical/scientific text | Bigrams | **L4-22796** |
| **C100** | Double space separator between multiple choice options in `dm_mathematics` | Math | **L4-9348** |
| **C101** | `also` after auxiliary/linking verbs in technical/scientific text | Grammar | **L2-18885** |
| **C102** | `Category` token after newline in `wikipedia` article footers | | **L4-28622** |
| **C103** | Newline followed by tab indentation in code/legal documents | Code | – |
| **C104** | Relative clause verbs after `which` / `that` in technical/formal text | Grammar | – |
| **C105** | Underscore `_` after identifiers in code/LaTeX (`github-all`, `arxiv`, `stackexchange`) | Math | – |
| **C106** | Code indentation (4 spaces) after newline in `github-all` and `stackexchange` | Code | **L2-30019** |
| **C107** | Domain suffix `org` after `.` in URLs (`hackernews`, `stackexchange`) | Code | **L3-9056** |
| **C108** | Exponentiation operator `**` after variables in `dm_mathematics` polynomial expressions | Math | **L3-23541** |
| **C109** | Comma after clause-final `that` and `and` in technical/legal text | Grammar | – |
| **C110** | Whitespace before line breaks in `enron_emails` and `stackexchange` | Formatting | – |
| **C111** | Delimiters and punctuation in code/math contexts ( `(` , `(` , `;` , `}` , `]` ) | Code | – |
| **C112** | Conjunction `and` after list items in scientific/technical text | Grammar | **L2-1172** |
| **C113** | Conjunctions after commas ( `or` , `but` , `and` ) in technical/legal text | Grammar | – |
| **C114** | `:` after `A` in `stackexchange` answer headers | | **L4-3827** |
| **C115** | Newline after math problem answers in `dm_mathematics` | Formatting | **L4-11424** |
| **C116** | Relative pronouns ( `that` , `which` , `who` ) introducing clauses in technical/legal text | Grammar | – |
| **C117** | Newline characters in `hackernews`, `freelaw`, and `stackexchange` discussions | Formatting | – |
| **C118** | Repeated `0` in comma-separated numeric arrays in `github-all` | Code | – |
| **C119** | Minus sign/subtraction operator in mathematical expressions | Math | – |
| **C120** | Indefinite article `a` after "as/is/have" across diverse texts | Grammar | – |
| **C121** | Article `an` | Grammar | – |
| **C122** | Newline after question in `dm_mathematics` | Math | **L2-26511** |
| **C123** | Apostrophe `'` after pronouns/nouns for contractions and possessives | Grammar | **L3-22772** |

| Cluster | Description | Category | SAE |
|---------|-------------|----------|-----|
| **C124** | Third-person plural pronouns ( `their` , `they` , `themselves` ) following entity references | Grammar | – |
| **C125** | Superlative/ordinal adjectives ( `first` , `most` , `best` , `following` ) after `the` in natural language | Grammar | – |
| **C126** | Colon after labels/headers in technical and structured text | Formatting | **L2-25505** |
| **C127** | Decimal point `.` after `0` in numeric sequences in `arxiv` | Math | **L2-32622** |
| **C128** | Preposition `in` after phrases requiring location/context specification | Grammar | **L2-24529** |
| **C129** | Variable name introduction after `Let` in `dm_mathematics` | Math | **L3-6264** |
| **C130** | Plural/past tense verbs ( `are` , `were` , `have` , `was` ) following plural nouns in scientific/technical text | Grammar | – |
| **C131** | Question marks ending sentences in `hackernews`, `stackexchange`, `cc`, and `enron_emails` | Formatting | – |
| **C132** | Token `same` as part of the bigram "the same" across technical/legal texts | Bigrams | **L2-22352** |
| **C133** | First-person pronoun `I` after sentence or clause boundaries in informal text | Grammar | – |
| **C134** | `&` after `0` in LaTeX matrix notation in `arxiv` | Math | – |
| **C135** | Hyphens and dashes as formatting separators in technical/academic text | Formatting | – |
| **C136** | `which` after comma in relative clauses | Grammar | **L2-5279** |
| **C137** | `we` after comma in academic/technical writing | Grammar | **L3-6738** |
| **C138** | `of` after nouns in technical/scientific contexts | Grammar | **L3-30829** |
| **C139** | Email address closing brackets in `enron_emails` ( `.com>` , `.edu>` followed by `,` or `;` ) | | **L3-17319** |
| **C140** | Auxiliary verbs after subjects (especially `is` ) in technical/scientific text | Grammar | – |
| **C141** | Mathematical expression closing parentheses followed by operators in `dm_mathematics` | Math | – |
| **C142** | Second-person pronouns ( `you` , `your` ) after conversational connectors in informal web text | Grammar | – |
| **C143** | Polynomial exponents ( `4` , `5` , `6` , `7` ) after `**` in `dm_mathematics` | Math | **L3-31787** |
| **C144** | Hyphens and dashes in scientific/technical text | | – |
| **C145** | Fraction `/` and minus sign `-` in mathematical expressions | Math | – |
| **C146** | Token `lot` as part of bigram "a lot" | Bigrams | **L4-9348** |
| **C147** | Tabs and code formatting tokens in `github-all` and technical contexts | Formatting | – |
| **C148** | The number `2` as both an exponent and in other math expressions | Math | – |
| **C149** | HTML anchor tag `a` after angle brackets in `github-all` | Code | **L4-2942** |

| Cluster | Description | Category | SAE |
|---------|-------------|----------|-----|
| **C150** | LaTeX math commands after backslash in `arxiv` and `stackexchange` | Math | – |
| **C151** | HTML/XML opening angle bracket `<` after newlines/whitespace in code | Code | **L4-16761** |
| **C152** | `it` after subordinating conjunctions (`because`, `if`, `when`, `since`) | Grammar | **L4-26404** |
| **C153** | Possessive `'s` after "it" and proper nouns | Grammar | **L2-15540** |
| **C154** | Comma after `ECT` in `enron_emails` email address lists | Formatting | **L4-11343** |
| **C155** | Multiplication operator `*` after numbers or variables in `dm_mathematics` equations | Math | – |
| **C156** | Time units after "many" in `dm_mathematics` and general text | Math | – |
| **C157** | Parenthetical expressions and clarifications in scientific/technical text | Grammar | **L2-7291** |
| **C158** | `**` exponentiation operator after variable names in mathematical expressions | Math | – |
| **C159** | Numeric tokens after arithmetic operators in `dm_mathematics` and mixed contexts | Math | – |
| **C160** | Sentence-initial capitalized words after periods in academic/formal text | Grammar | – |
| **C161** | Em dash separator `–` in informal/technical text | | **L2-20383** |
| **C162** | Newline after closing brace `}` in code (`github-all`, `stackexchange`) | Code | **L2-26486** |
| **C163** | URL path separators and formatting slashes | Code | – |
| **C164** | `pone` journal identifier in `pubmed_central` citation references | | **L4-6666** |
| **C165** | Document-initial tokens, mostly "Q", after `<|endoftext|>` in `stackexchange` and emails | Grammar | **L2-28741** |
| **C166** | Comma after numbers in lists and sequences | Formatting | – |
| **C167** | Underscore `_` for emphasis/italics formatting in `hackernews` and LaTeX subscripts in `arxiv` | Formatting | – |
| **C168** | Opening double quote after punctuation/text | Formatting | – |
| **C169** | `What` followed by `is` in `dm_mathematics` question patterns | Math | – |
| **C170** | HTML/XML closing angle bracket `>` after tag names in `github-all` and `stackexchange` | Code | – |
| **C171** | First token after `<|endoftext|>` in scientific/technical documents | Formatting | – |
| **C172** | Whitespace before line breaks in `enron_emails` and other text corpora | Formatting | – |
| **C173** | LaTeX command opening brace `{` after macro names in `arxiv` and `stackexchange` | Math | – |
| **C174** | Opening quotation marks and special characters that behave in a similar way, after articles/prepositions | Formatting | – |
| **C175** | Sentence-initial tokens in math problems in `dm_mathematics` | Math | – |

| Cluster | Description | Category | SAE |
|---|---|---|---|
| **C176** | Question marks ending sentences in `hackernews`, `stackexchange`, `enron_emails`, and `cc` | Formatting | **L2-5385** |
| **C177** | Closing brace `}` in code blocks (`github-all`, `stackexchange`) | Code | **L4-31580** |
| **C178** | `enron` after `@` in email addresses in `enron_emails` | | **L4-10333** |
| **C179** | `Category:` formatting pattern in `wikipedia_en` | | **L4-22002** |
| **C180** | Sentence-initial `These` after period in scientific/technical text | Grammar | **L4-28272** |
| **C181** | Conjunction `or` after alternative options in technical/scientific text | Grammar | – |
| **C182** | Closing quotation marks after quoted text | | – |
| **C183** | Superlative adjectives (`most`, `best`, `last`, `fastest`) after `the` | Grammar | – |
| **C184** | `term` after hyphen in "long-term" and "short-term" compounds | | **L4-14946** |
| **C185** | `=` often after closing parenthesis in function definition in `dm_mathematics` equations | Math | **L3-23065** |
| **C186** | `been` after auxiliary verbs (`have` / `has` / `had`) in perfect tense constructions | Grammar | **L2-9904** |
| **C187** | `people` and related plural nouns after quantifiers/determiners in `hackernews` and `cc` | Grammar | – |
| **C188** | Opening parenthesis `(` after conjunctions and punctuation in scientific/legal text | Formatting | – |
| **C189** | Newline after sentence-ending punctuation | Formatting | **L4-18943** |
| **C190** | XML/HTML closing tag delimiter `</` in code | Code | **L4-2503** |
| **C191** | `Pat` after "U.S." in patent citations in `uspto_backgrounds` | Bigrams | **L4-7189** |
| **C192** | Indefinite article `a` following prepositions/verbs in technical and narrative text | Grammar | **L4-6774** |
| **C193** | Relative clause and complement clause introducers (`that`, `which`, `who`) in technical/legal text | Grammar | – |
| **C194** | Variable definitions after `Let` in `dm_mathematics` | Math | – |
| **C195** | `://` after `http` / `https` in URLs across `hackernews`, `stackexchange`, `enron_emails` | Code | **L4-16482** |
| **C196** | `Invention` after "Field of the" in `uspto_backgrounds` patent documents | | **L4-27614** |
| **C197** | HTML anchor tag `href` attribute after `<a` in `github-all` | Code | **L3-2324** |
| **C198** | Sentence-ending `.` after variable names in `dm_mathematics` algebra problems | Math | – |
| **C199** | Hyphen after compound word prefixes (self-, high-, low-) | Formatting | – |
| **C200** | Consecutive newlines in `freelaw` legal documents | Formatting | **L4-18262** |
| **C201** | Not clear | | – |

| Cluster | Description | Category | SAE |
|---------|-------------|----------|-----|
| **C202** | Mathematical expressions after closing parentheses in `dm_mathematics` | Math | – |
| **C203** | Numeric digits after division operators or with minus signs in `dm_mathematics` | Math | **L3-9056** |
| **C204** | Minus sign after newline in `dm_mathematics` | Math | **L3-6699** |
| **C205** | Sentence-initial `The` after newline in expository text | Grammar | **L4-18242** |
| **C206** | 24-space indentation in `pubmed_central` table formatting | Formatting | – |
| **C207** | Token `be` as part of "to be" infinitive construction across diverse datasets | Bigrams | **L2-2297** |
| **C208** | 16-space indentation after newline in LaTeX `pubmed_central` documents | Formatting | – |
| **C209** | Time duration units after numeric quantifiers | | – |
| **C210** | Legal citation suffix `d` after F.2d/F.3d case reporters in `freelaw` | | – |
| **C211** | Contraction apostrophe `'t` after negation words (don't, doesn't, didn't, isn't, can't) in `hackernews` and `stackexchange` | Grammar | **L2-32558** |
| **C212** | Indentation and whitespace after newlines in `github-all`, `stackexchange`, and `freelaw` | Formatting | – |
| **C213** | Closing parentheses `)` after numbers/expressions in `dm_mathematics`, `arxiv`, `github-all`, and `pubmed_central` | Math | – |
| **C214** | Hexadecimal byte values `x` after `0` in `github-all` | Code | – |
| **C215** | `S` after `U.` in legal citations (U.S.C., U.S. Pat.) | Bigrams | **L4-6880** |
| **C216** | Indentation spaces after newlines in code (`github-all`, `stackexchange`) | Formatting | **L2-30019** |
| **C217** | Content (`a`, `b`, `td`, `ul`) of HTML/XML closing tags (`</a>`, ``, `</td>`, `</ul>`) | Code | – |
| **C218** | Pronoun + `was` after subject pronouns (`he`, `it`, `I`, `He`, `It`) | Grammar | **L2-26870** |
| **C219** | Variable name `z` in `dm_mathematics` algebra expressions after `*` | Math | **L2-29842** |
| **C220** | Hyphen `-` in compound words and formatting across technical/scientific text | Formatting | – |
| **C221** | `the` after "is" in `dm_mathematics` math questions | Bigrams | **L3-28171** |
| **C222** | `the` after prepositions in technical/scientific text | Grammar | **L2-13230** |
| **C223** | Ordinal position words (`second`, `third`, `remainder`, `smallest`, `biggest`) after `the` in `dm_mathematics` | Grammar | – |
| **C224** | Denominator digits after `/` in fractions in `dm_mathematics` | Math | **L3-19310** |
| **C225** | Document-initial capitalized words (`The`, `This`) after `<|endoftext|>` in scientific/technical texts | Grammar | – |

*Continued on next page*

| Cluster | Description | Category | SAE |
|---|---|---|---|
| **C226** | `ENRON` after `@` in email addresses from `enron_emails` | | **L4-20721** |
| **C227** | Female pronoun references (`her`/`she`) in narrative contexts | Grammar | – |
| **C228** | Contraction `'t` (don't, can't, isn't) in `cc` and `hackernews` | Grammar | **L2-32558** |
| **C229** | Newline after section dividers (` `, `----`, `======`) in `hackernews` and technical documents | Formatting | – |
| **C230** | `Category` after newline in `wikipedia` category lists | | **L4-28622** |
| **C231** | `S` after `U.` in `U.S.` abbreviation | Bigrams | **L2-14080** |
| **C232** | Article `an` after prepositions and verbs in technical/legal text | Grammar | – |
| **C233** | Contraction `'s` after "it" (it's) | Grammar | – |
| **C234** | `.html` file extension in URLs/paths in `github-all` | Code | **L4-11710** |
| **C235** | Negation phrases after pronouns ("don't", "haven't") in informal text | Grammar | **L4-18262** |
| **C236** | Token `as` as part of "such as" phrase completion in technical/scientific text | Grammar | – |
| **C237** | Token `g` completing "e.g." | Bigrams | **L4-3985** |
| **C238** | Contractions after `I` (`'m`, `'ve`, `'s`) in informal text | Grammar | – |
| **C239** | Token `in` in various contexts | Grammar | – |
| **C240** | Comparative phrases with `than` following `more`, `less`, `better`, `higher`, `greater` | Grammar | **L2-6584** |
| **C241** | Opening parenthesis `(` for multiple choice options in `dm_mathematics` | Math | **L3-31725** |
| **C242** | HTML/XML opening angle bracket `<` after newlines/whitespace in `github-all` and `stackexchange` | Code | – |
| **C243** | Question marks ending math problems and queries in `dm_mathematics`, `hackernews`, and `stackexchange` | Formatting | **L2-5385** |
| **C244** | HTML closing tag `>` after `li` and other elements in `github-all` | Code | **L2-28829** |
| **C245** | `16` after `DSP` in DSP16 function calls in `github-all` | | – |
| **C246** | First token after `<\|endoftext\|>` token | Formatting | – |
| **C247** | Common verbs after plural subjects (`have`, `are`, `know`, `to`) in explanatory text | Grammar | – |
| **C248** | `present` after `the`/`The` in `uspto_backgrounds` patent descriptions | Grammar | – |
| **C249** | LaTeX command opening brace `{` after math commands in `arxiv` | Math | **L2-463** |
| **C250** | `True`/`False` boolean answers after newlines in `dm_mathematics` math problems | Math | **L4-16089** |
| **C251** | HTML/XML attribute value delimiter `="` after attribute names in `github-all` and `stackexchange` | Code | **L2-15267** |

| Cluster | Description | Category | SAE |
|---|---|---|---|
| C252 | Digits after minus sign in `dm_mathematics` equations | Math | – |
| C253 | Ordinal suffix `'t` as in "'th" after variable names in `dm_mathematics` sequence term problems | Grammar | **L4-2426** |
| C254 | Document-initial tokens after `<\|endoftext\|>` in scientific/professional text | Formatting | – |
| C255 | Preposition `for` after purpose/use descriptions in technical and scientific text | Grammar | – |
| C256 | `){#` figure/table reference syntax in `pubmed_central` markdown | Formatting | **L2-11800** |
| C257 | Semicolon `;` after list items in technical/scientific text | Formatting | **L2-31196** |
| C258 | `of` after math terms like "term", "digit", "multiple", "factors", "derivative" in `dm_mathematics` | Grammar | **L4-7086** |
| C259 | Sentence-initial capitalized words after newlines | Grammar | – |
| C260 | Top-level domain extensions (`.com`, `.org`, `.edu`) after domain names in URLs | Code | – |
| C261 | Token `is` as part of "which is" pattern in `dm_mathematics` comparison questions | Grammar | – |
| C262 | Token `think` as part of "I think" in informal text (`hackernews`, `cc`, `enron_emails`) | Bigrams | **L4-8133** |
| C263 | Token `e` as part of "i.e." | Bigrams | **L2-16862** |
| C264 | Double space separator between multiple choice options in `dm_mathematics` | Formatting | – |
| C265 | `A` after newline in `stackexchange` Q&A answer markers | | **L2-30166** |
| C266 | XML/HTML closing tag delimiter `</` in code and markup | Code | **L2-9024** |
| C267 | Opening brace `{` after indentation in code blocks (`github-all`, `stackexchange`) | Code | **L4-20517** |
| C268 | Prepositions after verbs/phrases (`from`, `as`, `into`, `by`, `to`) | Grammar | – |
| C269 | Variable names `j` and `o` in `dm_mathematics` | Math | – |
| C270 | Token `the` as part of "is the" bigram in `dm_mathematics` comparison questions | Bigrams | **L4-13470** |
| C271 | `are` after "minutes" in time calculation questions in `dm_mathematics` | Grammar | **L4-6666** |
| C272 | Token ` example as part of "for example" in technical/patent text | Bigrams | **L4-18943** |
| C273 | Sentence-ending punctuation (`!`, `?`, `...`) in informal text | Formatting | – |
| C274 | `for` after variable name in `dm_mathematics` solve equations | Grammar | **L4-18262** |
| C275 | Sentence-initial `I` after newline in `hackernews` and `stackexchange` discussions | Grammar | **L4-4232** |

| Cluster | Description | Category | SAE |
|---|---|---|---|
| **C276** | Email quote continuation marker `>` after newline in `enron_emails` and `hackernews` | Formatting | **L2-18642** |
| **C277** | `----` comment separator in `hackernews` discussions | Formatting | **L4-1060** |
| **C278** | `com` after `.` in URLs across `hackernews`, `enron_emails`, and `stackexchange` | Code | – |
| **C279** | Multiplication operator `*` after numbers in `dm_mathematics` equations | Math | – |
| **C280** | Sentence-initial conjunctive adverbs (`However`, `Thus`, `Therefore`, `Furthermore`, `Moreover`) in academic/legal texts | Grammar | – |
| **C281** | Token `this` as part of "In this" in scientific/technical discourse | Grammar | **L3-27379** |
| **C282** | `with` after `patients` in medical/scientific literature | Grammar | – |
| **C283** | Relative pronouns (`which`, `who`, `whose`) in technical/formal writing | Grammar | **L2-5279** |
| **C284** | Exponent `3` - `7` after `**` in `dm_mathematics` polynomial expressions | Math | **L3-31787** |
| **C285** | `a` after `(` for multiple choice option labels in `dm_mathematics` | | **L4-21437** |
| **C286** | `://` after `http`/`https` in URLs across `hackernews`, `github-all`, `enron_emails` | Code | – |
| **C287** | Slash `/` and hyphen `-` as formatting delimiters in URLs, paths, fractions, and compound terms | Formatting | – |
| **C288** | Contrastive conjunctions (`but`, `while`) and similar words after commas | Grammar | **L2-27816** |
| **C289** | `case` after "the/this/in" in legal and technical contexts | Grammar | – |
| **C290** | Newline after `True`/`False` answers in `dm_mathematics` | Formatting | **L4-11424** |
| **C291** | Token `terms` in bigram "the terms" in `dm_mathematics` "Collect the terms" expressions | Bigrams | **L2-25825** |
| **C292** | `prime` after "a" or "the" in `dm_mathematics` primality questions | | – |
| **C293** | `type` after `-` in "ref-type=" attribute patterns in `pubmed_central` | Code | **L4-9348** |
| **C294** | Months of the year following prepositions like `in` and `on` in date expressions | | – |
| **C295** | `<|endoftext|>` token after sentence-ending punctuation in scientific/technical documents | Formatting | – |
| **C296** | URL domain suffix `com` after period in `enron_emails` and `hackernews` | Code | **L2-23912** |
| **C297** | Complementizer `that` after verbs/phrases expressing belief, knowledge, or observation | Grammar | – |

*Continued on next page*

| Cluster | Description | Category | SAE |
|---|---|---|---|
| **C298** | Paragraph breaks after sentence-ending punctuation in `freelaw` legal documents | Formatting | **L2-17680** |
| **C299** | Conjunction `and` in various contexts | Grammar | – |
| **C300** | Relative pronouns `that` / `which` introducing clauses | Grammar | – |
| **C301** | Hyphen `-` in ISO date lists (YYYY-MM-DD format) in `stackexchange` | Code | **L4-18262** |
| **C302** | LaTeX subscript closing `}_` and HTML `HREF` attributes in `arxiv` and `github-all` | Math | – |
| **C303** | Newline after HTML closing tags `>` or `">` in `stackexchange` | Formatting | **L4-13671** |
| **C304** | Asterisk `*` in code documentation comments (JSDoc/Javadoc style) in `github-all` | Math | – |
| **C305** | Sentence-initial capitalized words after periods (`In`, `At`, `Since`) in scientific/technical text | Grammar | **L2-9902** |
| **C306** | `~~~` separator between comments in scraped `hackernews` comments | Formatting | **L2-17795** |
| **C307** | Opening brace `{` after LaTeX commands and code blocks in `github-all`, `stackexchange`, and `arxiv` | Math | – |
| **C308** | Period after "U.S" in legal citations (`freelaw`, `uspto_backgrounds`) | Formatting | **L2-11214** |
| **C309** | First-person pronoun `I` after subordinate clauses in informal text | Grammar | **L4-4232** |
| **C310** | Hyphen after "anti" prefix in biomedical/scientific text | Formatting | **L4-23580** |
| **C311** | `Collect` / `Expand` math problem keywords in `dm_mathematics` | Math | **L4-3475** |
| **C312** | `D` prefix in DSP16 function calls in `github-all` | | **L3-1074** |
| **C313** | `I` after newline starting Stack Exchange questions | Grammar | **L4-4232** |
| **C314** | `closest` after "the" in `dm_mathematics` proximity questions | Math | **L4-21638** |
| **C315** | Multiple choice option `b` after `(` in `dm_mathematics` | Math | **L2-23839** |
| **C316** | Opening parenthesis `(` before multiple choice options in `dm_mathematics` | Formatting | **L3-31725** |
| **C317** | Newline after code statement endings in `github-all` and `stackexchange` | Formatting | – |
| **C318** | Token `as` as part of "as well as" | Bigrams | – |
| **C319** | Variable names (single letters like `j`, `q`, `z`, `h`, `i`) in `dm_mathematics` equations and scientific text | Math | – |
| **C320** | Hyphen/dash after newlines and in compound terms across `dm_mathematics`, `pubmed_central`, `arxiv` | Formatting | – |
| **C321** | Underscore `_` formatting character in code/LaTeX contexts | Math | – |
| **C322** | URL/path separator `/` after domain names and path segments | Code | – |
| **C323** | Numeric `0` after comma in arrays/lists in `github-all` | Code | – |

*Continued on next page*

| Cluster | Description | Category | SAE |
|---|---|---|---|
| **C324** | `return` keyword in code after indentation in `github-all` and `stackexchange` | Code | **L3-13106** |
| **C325** | Double newline continuation in `freelaw` legal documents | Formatting | **L4-18262** |
| **C326** | Ordinal suffix `h` after apostrophe in "n'th term" patterns in `dm_mathematics` | Grammar | **L2-24180** |
| **C327** | Ordinal position + `biggest` / `smallest` in `dm_mathematics` value comparisons | Math | **L4-19908** |
| **C328** | Function word `the` after prepositions/conjunctions | Grammar | **L2-13230** |
| **C329** | `ECT` token in Enron email address patterns (e.g., `/HOU/ECT@ECT`) | | **L4-22254** |
| **C330** | HTML tag attribute `class` after element names like `div`, `span`, `a`, `li` in `github-all` and `stackexchange` | Code | – |
| **C331** | Legal citation suffix `d` after F.2d/F.3d case reporters in `freelaw` | | – |
| **C332** | Double newline after `<|endoftext|>Q:` in `stackexchange` | Formatting | **L2-3257** |
| **C333** | `Let` after newline in `dm_mathematics` math problems | Math | **L3-1074** |
| **C334** | Markdown emphasis/italic markers (`_`) in `hackernews` discussions | Formatting | – |
| **C335** | LaTeX right and left commands in mathematical expressions | Math | – |
| **C336** | LaTeX math notation closing braces and delimiters in `arxiv`/`stackexchange` | Math | – |
| **C337** | Opening brace `{` after function/control structure declarations in code | Code | **L3-26738** |
| **C338** | Newline after closing brace `}` in code (`github-all`, `stackexchange`) | Formatting | – |
| **C339** | Opening parenthesis after `if` / `while` in code | Code | **L2-4280** |
| **C340** | Code indentation with tab and newline characters in `github-all` | Code | – |
| **C341** | Email quote continuation marker `>` in `enron_emails` | Code | – |
| **C342** | `was` / `is` after pronouns (`it`, `there`, `he`) in expository text | Grammar | **L2-26870** |
| **C343** | Single-digit numbers after minus signs in `dm_mathematics` | Math | – |
| **C344** | `Collect` followed by `the` in `dm_mathematics` polynomial term collection problems | Math | **L3-512** |
| **C345** | Numeric references, citations and section numbers in technical/legal documents | | – |
| **C346** | Token `with` as part of "associated with" bigram in scientific/technical text | Bigrams | **L2-24995** |
| **C347** | `we` after comma in scientific/academic text (`nih_exporter`, `arxiv`, `pubmed`) | Grammar | **L3-6738** |
| **C348** | `~~~` separator between comments in scraped `hackernews` forum comments | Formatting | **L2-17795** |

| Cluster | Description | Category | SAE |
|---------|-------------|----------|-----|
| **C349** | Opening parenthesis `(` after function names in `dm_mathematics` | Code | **L3-834** |
| **C350** | Closing quotation marks after quoted text in `uspto_backgrounds`, `cc`, `arxiv`, and `freelaw` | Formatting | – |
| **C351** | Sentence-initial capital letter `A` after period in scientific/technical text | Grammar | – |
| **C352** | `Enron` after `/` in email addresses from `enron_emails` | | **L4-10333** |
| **C353** | Exponent `2` after `**` in mathematical expressions | Math | **L2-14139** |
| **C354** | Sentence-initial `I` after newlines in informal text (emails, forums) | Grammar | **L4-4232** |
| **C355** | Token `with` as part of "associated with" bigram in biomedical/scientific texts | Bigrams | – |
| **C356** | Equation assignment `=` after `0` in `dm_mathematics` | Math | **L3-23065** |
| **C357** | Token `these` as part of "of these" bigram in scientific/legal text | Bigrams | – |
| **C358** | `="` after `ref-type` in `pubmed_central` XML cross-reference formatting | Code | **L4-21337** |
| **C359** | Token `following` as part of "the following" bigram in `arxiv` and other technical writing | Bigrams | **L2-7454** |
| **C360** | Variable names as function arguments and after `Let` in `dm_mathematics` | Math | **L2-30901** |
| **C361** | Token `base` as part of "In base" questions in `dm_mathematics` | Bigrams | **L3-22772** |
| **C362** | `is` after `What` / `Which` in `dm_mathematics` questions | | **L4-15064** |
| **C363** | Sentence-initial `They` / `Their` / `There` / `Those` after period | Grammar | **L4-19240** |
| **C364** | Token `the` as part of "be the" bigram in `dm_mathematics` derivative definitions | Bigrams | **L4-12133** |
| **C365** | Sentence-initial `A` after period in scientific/technical text | Grammar | – |
| **C366** | Contrastive conjunctions after commas (`but`, `while`, `because`) | Grammar | – |
| **C367** | Conjunction `and` in scientific/technical text | Grammar | **L2-1172** |
| **C368** | LaTeX `{` delimiter after label, begin, end commands in `arxiv` | Math | – |
| **C369** | Indefinite article `a` in math divisibility questions in `dm_mathematics` | Math | – |
| **C370** | `things` and `people` after various context words in `hackernews`, `cc`, and `stackexchange` informal text | Grammar | **L3-2279** |
| **C371** | Token `s` as part of Possessive "'s" in `cc` and `freelaw` | Grammar | – |
| **C372** | Closing double quote `"` after quoted text in mixed sources | Formatting | – |
| **C373** | `States` after `United` in country name bigram | Bigrams | **L4-25349** |
| **C374** | Dollar sign `$` as LaTeX math delimiter in `arxiv` and `stackexchange` | Math | – |

| Cluster | Description | Category | SAE |
|---------|-------------|----------|-----|
| **C375** | Auxiliary verbs after pronouns/subjects ( `is` , `can` , `was` , `has` ) in technical/formal text | Grammar | – |
| **C376** | Token `do` as part of "to do" bigram after infinitive marker | Bigrams | **L3-24016** |
| **C377** | HTML tag names after `<` in code/markup | Code | **L4-16622** |
| **C378** | Citation reference `B` after `[@` in `pubmed_central` | | **L2-6093** |
| **C379** | Variable name `v` after `Let` in `dm_mathematics` | Math | **L3-1074** |
| **C380** | `Solve` followed by minus sign `-` in `dm_mathematics` equations | Math | **L3-9056** |
| **C381** | Function word `the` after math problem instructions in `dm_mathematics` | | **L2-13230** |
| **C382** | Closing parenthesis `)` after function arguments, denominators or other contexts in `dm_mathematics` | Math | – |
| **C383** | Escaped backslash before LaTeX `usepackage` commands in `pubmed_central` document preambles | Math | **L3-5220** |
| **C384** | Sentence-initial capitalized words after newlines | Grammar | – |
| **C385** | Quotation marks and delimiters in code/markup formatting | Formatting | – |
| **C386** | Sentence-initial keywords ( `Let` , `Suppose` , `Is` ) after periods in `dm_mathematics` | Math | – |
| **C387** | Newline after sentence/statement endings in `dm_mathematics` and technical documents | Formatting | – |
| **C388** | `){` after reference IDs in `pubmed_central` markdown cross-references | Formatting | **L4-9348** |
| **C389** | Time format `00` after colons/periods in timestamps (`enron_emails`, `github-all`) | | **L2-23953** |
| **C390** | Sentence-initial capitalized words after periods in technical/scientific text | | – |
| **C391** | Variable names and mathematical symbols after operators/delimiters in `dm_mathematics`, `arxiv`, and `freelaw` | Math | – |
| **C392** | Code indentation with tabs ( `\n \t \t` and `\n \t \t \t` ) in `github-all` | Code | **L4-27050** |
| **C393** | Newline after sentence-ending punctuation in `enron_emails`, `stackexchange`, `hackernews`, `wikipedia_en`, `freelaw` | Formatting | – |
| **C394** | Options `c` , `d` in `dm_mathematics` multiple choice questions | Math | – |
| **C395** | HTML list item closing and opening tags ( `><` after `li` , `span` ) in `github-all` | Code | – |
| **C396** | Single lowercase letters, often in words where every letter is separated by a space | | – |
| **C397** | `Suppose` followed by minus sign `-` in `dm_mathematics` equations | Math | **L3-9056** |
| **C398** | Token `.,` as comma following "i.e." or "e.g." in scientific/legal text | Formatting | **L2-3597** |

| Cluster | Description | Category | SAE |
|---------|-------------|----------|-----|
| **C399** | Token `usepackage` as part of LaTeX "\usepackage" in `pubmed_central` documents | Math | **L3-1074** |
| **C400** | Modal verb `can` (also `could`, `may`, `would`) after subject pronouns in informal/technical text | Grammar | **L4-31978** |
| **C401** | `/` after `Corp` or `com` in URLs and email paths in `enron_emails` and `github-all` | Code | **L3-29181** |
| **C402** | `example` after "for" or "For" in technical/legal documents | Bigrams | – |
| **C403** | Nested parenthetical expressions `((-` in `dm_mathematics` | Formatting | **L3-18590** |
| **C404** | JavaScript/code comment markers `//` in `github-all` and `stackexchange` | Code | **L2-22162** |
| **C405** | Numeric tokens after mathematical operators and delimiters in `dm_mathematics` | Math | – |
| **C406** | Newline after horizontal rule separators in `hackernews` | Formatting | **L4-2328** |
| **C407** | `(` after function/method names and `-` in expressions across `dm_mathematics`, `stackexchange`, `github-all`, and `arxiv` | Code | – |
| **C408** | Token `the` as part of "to the" bigram in `dm_mathematics` rounding problems | Code | **L3-18437** |
| **C409** | Contraction `'re` after pronouns in `hackernews` and `stackexchange` discussions | Grammar | **L4-16875** |
| **C410** | Modal verb completions (`be`, `been`, `not`, `go`, `also`) following auxiliaries | Grammar | – |
| **C411** | `will` after research proposal descriptions in `nih_exporter` | Grammar | **L2-12242** |
| **C412** | `had` following pronouns/subjects in past perfect constructions | Grammar | **L4-32084** |
| **C413** | Ordinal words (`first`, `second`, `third`) after `the` in `dm_mathematics` derivative problems | | **L3-5943** |
| **C414** | `Solve` keyword after newline in `dm_mathematics` algebra problems | Math | **L4-17290** |
| **C415** | Comparison/relation markers (`with`, `as`, `in`, `=`) in technical and mathematical text | | – |
| **C416** | Six-space indentation after newline in `github-all` HTML/XML code | Formatting | **L2-9186** |
| **C417** | `HOU/` path separator in Enron email addresses | | **L3-1074** |
| **C418** | Citation reference letter `B` after `[@` in `pubmed_central` | | **L2-6093** |
| **C419** | `/` after "coq" in Coq library file paths in `github-all` | Formatting | **L3-29181** |
| **C420** | `the` after newline in `hackernews, freelaw,` and `enron_emails` | Grammar | **L4-18242** |
| **C421** | Numeric literal `0` after equals/operators in `dm_mathematics` | Math | **L3-8250** |
| **C422** | Indentation whitespace after newlines in code | Formatting | **L2-30019** |

*Continued on next page*

| Cluster | Description | Category | SAE |
|---------|-------------|----------|-----|
| **C423** | Newline after `True`/`False` answers and section headers in `dm_mathematics`, `pubmed_central`, and `arxiv` | Formatting | – |
| **C424** | Conjunctive adverbs (`thus`, `therefore`, `hence`) after coordinating conjunctions in technical/scientific text | Grammar | **L2-8884** |
| **C425** | Token `be` as part of "Let ... be" phrase in `dm_mathematics` derivative problems | Grammar | **L4-24505** |
| **C426** | HTML tag contents (`li`, `td`, `b`), a mix of opening and closing but mostly the former | Code | – |
| **C427** | Minus signs (sometimes combined with other characters) and left brackets in function arguments in `dm_mathematics` | Math | – |
| **C428** | Variable name `h` after `Let` in `dm_mathematics` | Math | **L3-6264** |
| **C429** | Newline after multiple consecutive newlines in `enron_emails` and `freelaw` | Formatting | **L2-3257** |
| **C430** | Domain name TLDs and email/URL components (`.com`, `.org`, `.edu`) | Code | – |
| **C431** | Second newline in double newlines in structured text | Formatting | – |
| **C432** | Token `order` as part of "in order" bigram | Bigrams | **L4-18262** |
| **C433** | Token `have` as part of "I have" bigram in informal/technical text | Bigrams | **L4-32084** |
| **C434** | Token `no` as part of negation phrases after "is/was/were" | Grammar | **L3-624** |
| **C435** | `OMNI` XML tag names in `enron_emails` | Code | – |
| **C436** | `)?` question delimiter in `dm_mathematics` math problems | Formatting | **L4-6666** |
| **C437** | Closing parenthesis `)` after `a` in multiple choice options in `dm_mathematics` | | **L3-24655** |
| **C438** | `Simplify` command after newline in `dm_mathematics` algebra problems | Math | **L2-31862** |
| **C439** | `Calculate` after newline in `dm_mathematics` math problems | Math | **L3-23953** |
| **C440** | Uppercase letter sequences in base64/encoded data | Code | – |
| **C441** | Token `|` as a separator in wiki tables and LaTeX formatting | Formatting | **L4-13088** |
| **C442** | `ref-type` attribute hyphen in `pubmed_central` XML references | Formatting | **L4-22796** |
| **C443** | Newline after username in `hackernews` comment replies | Formatting | – |
| **C444** | `found` after passive/discovery verbs in scientific and factual texts | Grammar | – |
| **C445** | Prepositions `in`/`with` in technical/scientific text | Grammar | – |
| **C446** | Token `:` as part of "Sent:" field delimiter in email headers and metadata formatting | Formatting | – |
| **C447** | `States` after `United` in legal/encyclopedic contexts | Bigrams | **L2-633** |
| **C448** | Citation reference markers in `pubmed_central` and `arxiv` | Formatting | **L3-10916** |

| Cluster | Description | Category | SAE |
|---|---|---|---|
| **C449** | Newline-following tokens (indentation/formatting) in code and structured text | Formatting | – |
| **C450** | Token `the` as part of "that the" bigram in legal/technical documents | Bigrams | **L4-18242** |
| **C451** | `what` after comma in `dm_mathematics` base arithmetic questions | | **L2-5836** |
| **C452** | Punctuation ( `,` or `.` ) after `<|endoftext|>` token | Formatting | – |
| **C453** | Modal verb followed by `be` (passive/future constructions) | Grammar | – |
| **C454** | Token `to` as part of bigrams "is to" and "was to" | Bigrams | – |
| **C455** | Capital letters as part of capitalized words/names following punctuation and conjunctions | Grammar | – |
| **C456** | Token `term` as part of "th term" in math sequences, `ories` in file paths (theories/setoid\_ring) | Math | – |
| **C457** | Token `you` as part of various forms of "Thank you" bigrams | Bigrams | **L2-19835** |
| **C458** | `Suppose` keyword starting math problems in `dm_mathematics` | Math | **L4-17290** |
| **C459** | Exponent `3` or `4` after `**` in `dm_mathematics` polynomial expressions | Math | **L3-31787** |
| **C460** | Contraction apostrophe after `don`, `doesn`, `didn`, `isn`, `can` in `hackernews`, `cc`, `stackexchange` | Grammar | **L2-32558** |
| **C461** | `00` following `-` in `pubmed_central` citation reference IDs (e.g., `viruses-10-00682` ) | | **L4-6666** |
| **C462** | Minus sign/subtraction operator after equals sign in `dm_mathematics` | Math | **L3-14513** |
| **C463** | Percentage symbol `%` after numeric values in scientific/medical texts | Math | **L3-23877** |
| **C464** | Exclamation marks and ellipses in informal/emphatic text | | – |
| **C465** | Newline after line-wrapped legal text in `freelaw` | Formatting | **L4-18262** |
| **C466** | Token `.` after parts of abbreviations or as decimal points | Formatting | – |
| **C467** | Single-digit numbers after minus signs in `dm_mathematics` | Math | – |
| **C468** | Token `new` as part of bigram "a new" across diverse technical and legal contexts | Bigrams | **L3-11996** |
| **C469** | Closing double quote after quoted text/strings | Formatting | |
| **C470** | Token `many` as part of bigram "How many" in `dm_mathematics` time calculation questions | Bigrams | **L4-31179** |
| **C471** | Token `nearest` as part of bigram "the nearest" in `dm_mathematics` | Bigrams | **L4-8078** |
| **C472** | Numeric digits after decimal points in scientific/technical text | Math | – |
| **C473** | Token `the` as part of bigram "to the" across technical/legal documents | Bigrams | **L4-18242** |
| **C474** | Period after `U.S` abbreviation in legal citations and formal documents | Formatting | **L2-11214** |
| **C475** | Conditional `you` after `if` / `If` / `when` / `what` in conversational text | Grammar | **L4-21933** |

| Cluster | Description | Category | SAE |
|---------|-------------|----------|-----|
| **C476** | Token `a` as part of bigram "Let a" in `dm_mathematics` variable definitions | Bigrams | **L3-16348** |
| **C477** | Newline after `<\|endoftext\|>` token in `hackernews` and `freelaw` | Formatting | – |
| **C478** | `the` followed by `remainder` / `closest` in `dm_mathematics` math word problems | Math | – |
| **C479** | `Enron` after `@` in `enron_emails` email addresses | | **L2-30764** |
| **C480** | Token `control` in scientific/technical contexts | | **L2-4778** |
| **C481** | Sentence-initial capitalized words after newlines in `dm_mathematics` | | – |
| **C482** | 24-space indentation in `pubmed_central` tables | Formatting | – |
| **C483** | Double newline paragraph breaks in `cc`, `pubmed_central`, `wikipedia_en`, and `arxiv` | Formatting | **L2-3257** |
| **C484** | `What` after newline in `dm_mathematics` question sequences | | **L4-21647** |
| **C485** | Uppercase letters and short tokens in base64/encoded strings and technical content | Code | – |
| **C486** | Prepositions `to`, `in` in mathematics and code and a range of other tokens | Grammar | – |
| **C487** | Relative pronoun `who` after person references in descriptive clauses | Grammar | **L4-13411** |
| **C488** | Auxiliary verbs `is`, `has`, `have` after subjects in technical/scientific text | Grammar | – |
| **C489** | LaTeX superscript `^` and subscript `_{` operators in mathematical notation | Math | – |
| **C490** | Two-digit numeric date/time components after delimiters | | – |
| **C491** | `\n \t \t \t` newline-tab indentation in HTML templates in `github-all` | Formatting | **L4-26957** |
| **C492** | Exponent `3` after `**` in polynomial expressions in `dm_mathematics` | Math | **L3-31787** |
| **C493** | `been` after auxiliary verbs (`has` / `have` / `had`) in perfect tense constructions | Grammar | **L3-14473** |
| **C494** | Abbreviation punctuation `.,` after "e.g" and similar Latin abbreviations in technical/legal text | Formatting | – |
| **C495** | Token `etc` after a comma | Grammar | **L4-22683** |
| **C496** | Digit `0` after decimal points in numeric data (`github-all`, `arxiv`) | | – |
| **C497** | Cyrillic text and non-ASCII characters in `stackexchange` and `github-all` | | – |
| **C498** | `Court` and `court` after "the/this/Supreme/District" in `freelaw` legal documents | | **L3-6380** |
| **C499** | `including` after comma in list introductions | Grammar | **L4-22796** |

| Cluster | Description | Category | SAE |
|---|---|---|---|
| **C500** | Division operator after closing parenthesis in `dm_mathematics` expressions | Math | – |
| **C501** | Sentence-initial ` You ` after periods in `hackernews, cc,` and `stackexchange` | Grammar | **L4-21933** |
| **C502** | ` > ` quote marker after newline in `hackernews` and `enron_emails` discussions | Formatting | **L3-31415** |
| **C503** | ` with ` following comma in lists | Grammar | **L4-22796** |
| **C504** | Italic text marker ` * ` in scientific/academic writing (`pubmed_central,` `arxiv`) | Formatting | **L2-30291** |
| **C505** | ` In ` after newline in `dm_mathematics` base arithmetic problems | | – |
| **C506** | Opening angle bracket ` < ` in HTML/XML markup in `github-all` and `stackexchange` | Code | **L3-14450** |
| **C507** | Contraction ` 's ` after "it" and "that" in informal text | Grammar | **L3-22099** |
| **C508** | Prepositions ` in `, ` on `, ` at `, often in technical contexts | Grammar | – |
| **C509** | Sentence-initial capitalized words after periods ( ` In `, ` It `, ` With `, ` Since `, ` First ` ) in scientific/technical text from `nih_exporter,` `wikipedia_en, cc, uspto_backgrounds, pubmed_central` | Grammar | – |
| **C510** | LaTeX escaped backslash ` \\ ` often after newlines in `arxiv` math equations | Math | – |

**J.3. Cluster taxonomy in Pythia-14M**

Below we provide a taxonomy of the susceptibility clusters, classifying them into six categories: Formatting, Grammar, Code, Math, Bigrams and Other.

J.3.1. SYNTACTIC & GRAMMATICAL PATTERNS

- **Auxiliary & Modal Verbs**
  - Modal verbs like `can`, `may` **C14, 63, 400** and the infinitive `be` **C453**.
  - Auxiliary verbs `is`, `has`, `have` **C1, 140, 488**.

- **Determiners & Articles**
  - `a` following prepositions **C4, 192**.
  - `an` **C26, 121, 232**.
  - `the` in specific bigrams like "that the" or "to the" **C53, 270, 408**.

- **Pronouns & References**
  - Relative pronouns `which`, `who`, `whose` **C104, 116, 283**.
  - Third-person pronouns `he`, `she`, `they` **C74, 124, 227**.

- **Prepositions & Conjunctions**
  - Conjunctions `and`, `but`, `or` **C55, 113, 299**.
  - Prepositions `in`, `on`, `at`, `with` **C11, 28, 64**.

- **Other**:
  - Passive constructions like `by` after past participle `caused`, `supported` **C41**.

J.3.2. MATHEMATICAL & SCIENTIFIC NOTATION

- **Algebraic Operations**
  - Exponentiation `**` **C2, 40, 108**.
  - Multiplication `*` **C15, 57, 155**.
  - Division/Fractions `/` **C145, 224**.
  - Equality `=` **C54, 356**.

- **Variable Definitions**
  - "Let [variable] be..." **C70, 129, 194**.
  - Single-letter variables (`x`, `y`, `n`) **C10, 51, 67**.

- **Problem Statements**
  - "What is..." **C169**.
  - "Calculate..." **C439**.
  - "Collect the terms..." **C344**.

- "nearest to" **C471**.

- **LaTeX Typesetting**
  - Math delimiters `$` **C49, 68, 374**.
  - Commands `\begin`, `\frac`, `\usepackage` **C150, 249, 399**.

- **Numeric & Unit Formatting**
  - Decimals and floating points **C23, 127, 496**.
  - Negative numbers/Minus signs **C7, 204, 343**.

J.3.3. CODE, MARKUP & TECHNICAL SYNTAX

- **Markup & Tags (HTML/XML)**
  - Opening/closing tags `<` `>` **C52, 170, 242**.
  - Attributes `ref-type`, `class` **C330, 358**.
  - Specific tags `div`, `span`, `li` **C426**.

- **Control Structures & Syntax**
  - Conditionals `if` `(` **C97, 339**.
  - Braces `{` `}` for blocks **C86, 162, 337**.
  - Comments `//` **C404**.

- **Indentation & Whitespace**
  - Indentation (4 or 8 spaces) **C17, 106**.
  - Newlines in code blocks **C60, 103, 491**.

- **Infrastructure & Metadata**
  - URL protocols `http` `://` **C195, 286**.
  - Domain suffixes `.` `com`, `.` `org` **C107, 296**.
  - Email headers "Sent:" **C446**.

J.3.4. DOCUMENT STRUCTURE & FORMATTING

- **Sentence & Document Boundaries**
  - End of text token `<|endoftext|>` **C45, 171**.
  - Sentence starters (Capitalized words) **C36, 96, 384**.
  - Question marks **C87, 131**.

- **Citations & References**
  - Brackets in citations and references within text **C20, 256**.
  - Legal citations "U.S.C.", "F.2d" **C210, 215**.
  - Figure references **C33**.

- **Visual Separators**
  - Horizontal rules `--`, `===` **C229, 306, 406**.
  - Double newlines for paragraphs **C22, 325, 483**.

- **Lists & Enumeration**
  - Multiple choice options "(a)", "(b)" **C315, 437**.
  - List bullets and numbers **C166**.

### J.3.5. BIGRAMS

- **Bigrams**

    - "United `States`" **C373**.
    - "associated `with`" **C346**.
    - "Thank `you`" **C457**.
    - "based `on`" **C99**.

## J.4. Low-level vs high-level patterns

The clusters discovered by our methodology vary in their level of abstraction. We attempt here to characterize the distinction between "low-level" or "syntactic" clusters and "high-level" or "abstract" clusters. The distinction is not sharp, and even our most abstract examples are still quite low-level. Nonetheless this is an interesting qualitative aspect of the clusters to track as we look at larger models in the future: one can see for example a distinction between the rather low-level SAEs in Bricken et al. (2023) for a one-layer transformer with the much more abstract examples in Templeton et al. (2024).

Low-level clusters tend to be defined by a specific token in a specific positional pattern, depending on immediate, local features such as "what character precedes this one?" Examples include **C148** (the digit `2` as an exponent), **C17** (8-space indentation after newlines), **C154** (comma after `ECT` in Enron email address lists), and **C389** (time format `00` after colons in timestamps).

Higher-level clusters tend to abstract over token identity to capture a shared functional role, or require recognizing compositional structure across longer distances.

- **C10**, **58**, **269**, **319**: Variable names in `dm_mathematics`

- **C11**: Prepositions ( `to`, `on`, `by`, `in`, `for` )

- **C19**: Comparison words ( `smaller`, `greater`, `bigger` )

- **C63**: Modal verbs ( `will`, `is`, `should`, `would`, `must` ) following subjects

- **C100**: Double space separator between multiple choice options in `dm_mathematics`

- **C125**: Superlative/ordinal adjectives ( `first`, `most`, `best`, `following` ) in natural language

- **C183**: Superlative adjectives ( `most`, `best`, `last`, `fastest` )

- **C217**: Content of closing HTML tags (e.g. `a`, `b`, `td` )

- **C247**: Common verbs after plural subjects ( `have`, `are`, `know`, `to` )

- **C252**: Digits after minus sign in `dm_mathematics`

- **C280**: Sentence-initial conjunctive adverbs ( `However`, `Thus`, `Therefore`, `Furthermore`, `Moreover` )

- **C410**: Modal verb completions ( `be`, `been`, `not`, `go`, `also` )

- **C424**: Conjunctive adverbs ( `thus`, `therefore`, `hence` )

- **C426**: HTML tag contents ( `li`, `td`, `b` ), both opening and closing but mostly the former

- **C440**: Uppercase letter sequences in base64/encoded data

## J.5. Detailed Example of Clusters

Below we give 30 randomly selected samples from each of the twelve clusters showcased in Figure 2, as well as an evaluation of whether this sample is "on theme".

Note that the clusters are strongly monothematic. **C455** (which was selected for having high entropy among its tokens) is the only cluster with fewer than 75% of contexts clearly having the same theme. Seven out of the twelve clusters (including every cluster that was randomly selected) have greater than 90% of examples matching the cluster's theme. We view this as compelling evidence that the clusters are identifying context-token pairs that are related to each other, with little outside noise, and that the labels given to each cluster are broadly accurate. It is also interesting to note that when contexts in a cluster are "off-theme", they are frequently related. For example in **C294** ( months and seasons) the examples that fall out of the theme include years and the word "date".

*Table 10.* **Cluster 455**    size = 5122    theme: Capital letters, often at start of names    score: 19 / 30

| # | J | Context |
|---|---|---------|
| 1 | **X** | esModule: true };\n},{"core-js/library/fn/symbol/iterator": 75 }],56 |
| 2 | **X** | ().execute(fileOutputStream2, stream_url);\n } catch (FileNotFoundException e 5 ) {\n |
| 3 | ✓ | Keith, a woman who had gone missing in March 1993.\n\nSubsequent developments\n\nAfter Y arborough passed |
| 4 | ✓ | participated in the Eurovision Song Contest 2014 in Copenhagen, Denmark. The Slovenian entry was selected through E MA 2014, |
| 5 | ✓ | Lau Ying, who is now the director of Putien. They have two children, Fong Ch ak Wai |
| 6 | **X** | t e o f F r a u d s. " B e a z l e y v |
| 7 | **X** | $ and $\mu_1$ (corresponding to $X|Y=0$ and $X | Y=1 |
| 8 | **X** | \n <a href="../07_ruby_on_rails/d02/underscore- t pl.html |
| 9 | ✓ | used in this study are as follows: PKM1-F, 5-CAGCCAAAG G GGACTAT |
| 10 | **X** | uin colonies: relationships with host breeding activity.\nA survey of the temporal pattern of population structure and feeding activity of the |
| 11 | **X** | of protein was fractionated on 10 to 17.5% SDS–polyacrylamide gel electrophoresis gel and transferred to a nitro |
| 12 | **X** | store called Samsung Music, that is an official Samsung …... How to Download Music to Samsung Galaxy S 6 /S6 |
| 13 | ✓ | re looking for a financial advisor who is honest, trustworthy and interested in you as an individual, Roger Cal dwell is that |
| 14 | ✓ | enlisted in the 101st together in 1942.\n\nIn Mark Twain's The Diary of Adam and Eve ( |
| 15 | ✓ | tress that gave the Children much of their power. The Children used their magic to shatter the Arm of Dorne |
| 16 | ✓ | Muslims had their armed 28th Division in the "safe area," had not been disarmed by the UN , had plenty |
| 17 | ✓ | Shammi (Punjabi) Shammi, Santosh Kumar, Ajmal, Sh ola, Gh |
| 18 | ✓ | 4ff. W. Witten, Physics Today, April 1996, pp.24-30. M . Maggi |
| 19 | ✓ | E.O.C. The Court interpreted the two provisions to hold that a charge filed with the E .E. |
| 20 | ✓ | were transfected with Nrf2-siRNA and analyzed by Western blots for expression of Shh, G li1, |
| 21 | ✓ | to investigate why Baird increased the membership of the board of directors from seven to nine - and why Ra itt, while |
| 22 | ✓ | anger management issues" and a significant history of relationships involving domestic\nviolence.\n Mother told D CFS she |
| 23 | ✓ | C3 _________________________________ Vin < -Vref -1 S2A S4B S 4C S |
| 24 | ? | suggesting signaling functions in tumor cells (Non Patent Literature 22).\nIn immunohistochemical analysis using clinical samples, h TROP2 |
| 25 | ? | 1067?\n-16*n**2 + 356*n - 493\nWhat is the g 'th term |

*Table 10.* **Cluster 455**     size = 5122     theme: Capital letters, often at start of names     score: 19 / 30

| # | J | Context |
|---|---|---------|
| 26 | ✓ | (\mathcal I^*,\mathcal L^*)\cdot C&=& N_{\mathcal I^*} \cdot C + N_{\ |
| 27 | ✗ | as opposed to rejected and completely changed, can't it?\n\n~~~\nloup-vaillant\nThis |
| 28 | ✓ | 0\n\t\t\tunsigned rsvd4 : 28;\t\t// 31:4\n\t\t} HIEISR |
| 29 | ✓ | \nVoxy\|Director of Engineering\|New York, NY or Sao Paolo, BR\|Full-Time\| |
| 30 | ✓ | professor in the Department of Epidemiology and Biostatistics at the University of California at San Francisco. He also served as |

*Table 11.* **Cluster 194**     size = 1737     theme: Variable names     score: 21 / 30

| # | J | Context |
|---|---|---------|
| 1 | ✗ | }^2)\n (\hat{u}-m_{\tilde{q}_{l^{\prime}}}^ 2)}\n |
| 2 | ✓ | = 37, -2*t - g = 5 for b.\n5\nSuppose 2*a - 3* |
| 3 | ✗ | is the nearest to 19? (a) 3 (b) 0 (c) 0.4 |
| 4 | ✗ | y@ECT\nTo: Kay Mann/Corp/Enron@Enron\ncc: \n\n Subject: Re: |
| 5 | ✓ | \n-54*c - 2\nLet o(b) = -b. Let y(g) be the |
| 6 | ✓ | 4*f - 794 = -f. Is l a composite number?\nFalse\nLet n (y) |
| 7 | ✓ | )**2\nLet d(c) = c**2 - 11*c + 6. Let b be d( |
| 8 | ✓ | Let v = -2 + 1.4. Let z = 0.4 - v. Let i = z - |
| 9 | ✗ | control contractile responses whereas longer term Ca2+ signals regulate cell growth and proliferation through Ca2+-mediated transcriptional control. |
| 10 | ✓ | j - 28 = 2*d, 3*j = -3*d - 21. Let m = -1 |
| 11 | ✓ | . Let a(u) = 0. Calculate u.\n-43, -2\nLet v (w) |
| 12 | ✓ | = -2*b. What is the units digit of n(b)?\n1\nSuppose j - 51 - |
| 13 | ✗ | keeps track of a separate history about target appearance changes, the proposed algorithm is effective to handle multi-modal target appearances and |
| 14 | ✓ | 69/276) is divided by 7/5 + (-2)/5.\n0\nLet i = -17 |
| 15 | ✗ | higher spin states for $N_f = 2+1$'\n—\n\nIntroduction\n ============ \n\nCh |
| 16 | ✓ | - 4*t + 1/4*t**4 + 0*t**3. Let d (u) |
| 17 | ✗ | is the third biggest value in 0.2, m, y?\ny\nLet v = - 15.8 |
| 18 | ✓ | \nSolve 2*f + 0*p - 10 = -4*p, -2*f - 9* |
| 19 | ✓ | What is the closest to -4 in o, 4, -3?\n-3\nLet z be (0 |
| 20 | ✓ | (a) m (b) p (c) 1/3\na\nLet l be (-2 |

*Table 11.* **Cluster 194**    size = 1737    theme: Variable names    score: 21 / 30

| # | J | Context |
|---|---|---------|
| 21 | X | modifications from internet but didnt allowed me to post after the first successful time\nForm: \n< form action="post |
| 22 | X | T;H^s(S))$ in equation such that $\vec{u}(T)=\vec { 0}$, that |
| 23 | ✓ | 0. List the prime factors of k.\n2, 3, 11\nSuppose -4* r - o - |
| 24 | ✓ | for s.\n-14\nSolve 47*i - 34 = 46*i - 2* z , 22 = |
| 25 | ✓ | /5\nLet p(a) = a**2 + 33*a - 82. Let l be p(- |
| 26 | ✓ | 1, o in ascending order.\n-1, o, j\nSuppose o = 4* i + 24, |
| 27 | ✓ | *c + 12\nLet v(z) = z**2 + z + 1. Let u = 4 + |
| 28 | ✓ | m + 2. Let p be g(c). Give i(p).\n3\nLet o (d) |
| 29 | ✓ | .5, 4?\n4\nLet k = -779 - -775.05. Let d = k - |
| 30 | ✓ | the prime factors of 3 - 34*q/6?\n2, 3\nSuppose 4* v = 14 + |

*Table 12.* **Cluster 11**    size = 7071    theme: Prepositions    score: 25 / 30

| # | J | Context |
|---|---|---------|
| 1 | ✓ | and the office.\n\nThe Thrival Company is a social entrepreneurial management consulting firm that focuses on leading organizations to |
| 2 | ✓ | than from the father. In this larger survey of the parental transmission of disease in Huntington chorea 12 of 13 patients whose |
| 3 | ✓ | tptacek\nSo should we have "neutrality" in every market with large barriers to entry?\n |
| 4 | ✓ | ANTIHERO".<\|endoftext\|>Sunday morning, South Korea announced that it was extending its air defense zone to include a tiny |
| 5 | ✓ | all of\nthese systems were affected by different issues.\n\nYou're better off using a jump to conclusions board rather |
| 6 | ✓ | .\n\nThe most iconic companies in NewSpace are Blue Origin and SpaceX. Both are driven by long-term |
| 7 | ✓ | having demanded cancellation of the lease in these letters, plaintiffs are estopped from thereafter urging non-performance on the part of |
| 8 | X | the desired plot with regular plots.\nd <- rbind(\n data.frame(value = 1:100 |
| 9 | ✓ | proportion of rural dwellers accessed the grant. The grant holders displayed significantly more problems related to mobility and to technology and policies |
| 10 | ✓ | aming group displaying higher mean scores compared with the free-play group. Young children who were randomly assigned to the exerg |
| 11 | ✓ | <\|endoftext\|>Thelymitra petrophila\n\nThelymitra petrophila, commonly known as the granite sun |

*Table 12.* **Cluster 11**    size = 7071    theme: Prepositions    score: 25 / 30

| # | J | Context |
|---|---|---------|
| 12 | ✓ | many other low and middle income countries (LMIC), the burden of disease has gradually shifted from infectious to non-commun |
| 13 | ✓ | who shot straight at De Gea. Ronaldo soon had his own share of show boating to entertain the crowd |
| 14 | ? | one paid offer. If you happen to miss the fact that one of the offers costs money, you ’ re out $ |
| 15 | ✓ | of single crystal x-ray diffraction techniques. An attempt will be made to relate the structures thus determined to the structure of |
| 16 | ✓ | required to be in accordance with \neither the Rules framed by the RBI or otherwise specifically approved by the \n |
| 17 | ✓ | and the remaining ones are considered synchronized. The first and the last bits are equal, while the middle one is different ([ |
| 18 | X | !== ’function’) {\n\t\t\tthrow new TypeError();\n\t\t}\n\n\t\tvar res = []; \n\t\tvar |
| 19 | ✓ | creditor has reasonable cause to believe his debtor insolvent, constitute a preference? The correct answer is believed to be ‘yes |
| 20 | X | , keyIndex, value, isToggleBool) {\n if (isToggleBool === void 0 ) {\n |
| 21 | X | R^n$) found that if he wanted a square root of $\Delta = - \sum \ partial _i^ |
| 22 | ✓ | utable, Greif faulted the\n 3 Liquidating Trustee for not adopting that position in his\n 4 |
| 23 | ✓ | Court for the Southern District of New York. These records relate to proceedings which occurred subsequent to the ruling on demurrer |
| 24 | ✓ | Q:\n\nA simple scatterplot example in D3.js?\n\nI’m looking for an example of |
| 25 | ✓ | better analytical performance for the glucose detection compared with the IL-SPE based biosensor. The linear range for the detection of |
| 26 | ✓ | was promoted to the big job.\n\nSubscribe to Sports Now newsletter\n\n“After what happened on the sideline |
| 27 | ✓ | ol’ pat on the back for bringing us red bulls eye lovers a new easy way to save on specific items. |
| 28 | ✓ | Number Ten\n\nThe Battle of Island Number Ten was an engagement at the New Madrid or Kentucky Bend on the Mississippi River |
| 29 | ✓ | <\|endoftext\|>a’) show that the active site is required to be highly electroph |
| 30 | ✓ | \nmade the Holocaust so terrible, but truth be, any other country could have\nfallen victim to the same propaganda |

*Table 13.* **Cluster 167**    size = 71    theme: Text emphasizers, commonly ’_’, and ’_’ as subscript indicator    score: 29 / 30

| # | J | Context |
|---|---|---------|
| 1 | ✓ | hypothesis on this: it’s clear that one skill that used to be\nimportant is now _much_ less important, |

*Table 13.* **Cluster 167**    size = 71    theme: Text emphasizers, commonly '_', and '_' as subscript indicator    score: 29 / 30

| # | J | Context |
|---|---|---------|
| 2 | ✓ | , the concentration of endogenous choline did not change significantly in all transgenic plants expressing the *codA * gene ([@B |
| 3 | ✓ | ]\]. To generate myeloid-specific Ninj1-deficient mice, Ninj1^fl/fl ^ mice were bred |
| 4 | ✓ | Plassys evaporator. However, due to the substantially different regimes of critical currents, $I _ 0$, required |
| 5 | ✓ | with a situation\nlike this?\n\nNot only do I dislike the freedom limitations _in principle _, but I |
| 6 | ✓ | in most cases it'll only discourage customer attendance.\n\nTo be a member of _the queue _ of an establish-ment |
| 7 | ✓ | for the 7 year offset in the database\n return x + 7\n \n def process _ record(record |
| 8 | ✓ | Golbraikh method \[[@B30-molecules-23-01377]\]. Acceptable q^2 ^ values are equal |
| 9 | ✓ | longer lifespan, better health, and seemingly equivalent\nhappiness, it's arguable that _not _ keeping a cat |
| 10 | ✓ | apsack" (see Garey and Johnson(1979)): Given a sequence of integers $a _ 1 \ldots |
| 11 | ✓ | ~~~\nbootload\n_"... This is well known. This "news" is very old..." _ \n\nOld |
| 12 | ✓ | I have been doing this, there have been probably around 3-4\ntimes when I _have _ wanted to customize |
| 13 | ✓ | the control loop.\n\nAlso, the section about proximity fuzing: cruise missiles _are _ intended to\n |
| 14 | ✓ | behavior. As a consistent picture emerges, we tend to\nswitch from thinking of people as _acting _ a certain way |
| 15 | ✓ | 004\n\n###### Table of stable isotope values for the leaves of *Mangifera indica * (Common name |
| 16 | ✓ | the\n> bandwidth needed.\n\n _Very_ sound advice. Doing this has produced _amazing _ results for us |
| 17 | ✓ | [@Dill1997NSB]. Here, however, we prefer to save the name *free energy * for the one |
| 18 | ✓ | failed to\nconnect the dots until exactly the moment I read your comment.\n\nThere _is _ a void, |
| 19 | ? | \frac{c_H^2 \, R^4}{c_b^2\,Z _* \, \alpha |
| 20 | ✓ | Scores for all participants were calculated at the time of the first interview of each interview set^b ^ Both participants who |
| 21 | ✓ | shut down, and the scammed will fight you.\n\nScammed investors will _defend _ the scammer |
| 22 | ✓ | peripheries, but the core's comparative advantage becomes in _renting out\nsurplus capital _ (which produces |
| 23 | ✓ | . We consider strong, moderate and weak eccentricity damping, corresponding to $t_e/t _ a\n\ |
| 24 | ✓ | really relevant, I was making the general point that modern cars have\ngotten a _ridiculous _ amount of power |
| 25 | ✓ | opposite hypothesis and still claim the\nnumbers support it. The use of gaming consoles and smartphones _improve _ IQ,\n |
| 26 | ✓ | ate a bit - but in every single line of your CV, you should be\n_showing _ not telling. |
| 27 | ✓ | link:\nhttp://www.nyiso.com/services/documents/groups/weekly _ updates/ |
| 28 | ✓ | competitors naive._\n\nYeah, and that's the primary reason I increasingly _don 't want _ to be an |

*Table 13.* **Cluster 167**     size = 71     theme: Text emphasizers, commonly '_', and '_' as subscript indicator     score: 29 / 30

| # | J | Context |
|---|---|---------|
| 29 | ✓ | your assumptions are correct:\n<http://en.wikipedia.org/wiki/Millennium _ Challenge_ |
| 30 | ✓ | 's a fun way to approach the subject from a different angle.\n\nMany kids _hate _ math but might |

*Table 14.* **Cluster 161**     size = 60     theme: Em dashes     score: 27 / 30

| # | J | Context |
|---|---|---------|
| 1 | ✓ | been little affected \nby the search firm nomination process)\nCA-ISO.\nArea included – CA IOU |
| 2 | ✓ | but the exceptions – some of which are pretty big, eg. the entire Australian\nCapital Territory – will kill you |
| 3 | ✓ | </script><!– HubSpot Call-to-Action Code –> <!– hs-cta-wrapper – ></ |
| 4 | ✓ | \n "!! your program said tw={:d} but we thought it was {:d} – please send your |
| 5 | ✓ | profile machine learning research team within Amazon.  On my\nfirst day, I knew something was off immediately – the manager literally |
| 6 | ✓ | This will also result in \nCalifornia tax revenues being spent on power transmission and power \ngeneration – which the private |
| 7 | ✓ | \n\nI'm 3 months in on Keto, planning on taking blood tests in another 3 months – \nI'm |
| 8 | ✓ | work out a better system with your leads?\n\nThat's the tip of the ice berg – I think if |
| 9 | ✓ | ask any rank-and-file Google engineer, they'll totally\nagree that the sweatshop layout – oops, |
| 10 | ✓ | SHM technology into one of the simplest load-bearing biomaterial formulations – two component acrylic bone cement – with the most |
| 11 | X | o v. ACLU,\n521 U.S. 844, 874 (1997) (" In evaluating the |
| 12 | ✓ | AccountMinder organizes all your account statements – from banks, credit \ncards, and more – into one easy |
| 13 | ✓ | level than the national \nllevel. You can build a network of contacts over time, too – the Web makes |
| 14 | ✓ | To:\tDorland, Chris\nSubject:\t \n\nhey toughguy – whats the story – how's everything |
| 15 | ✓ | from). To my surprise, it was very rewarding. I met some\nvery knowledgeable and interesting people – even some that |
| 16 | ✓ | and \nsimple, is unconscionable price-gouging by the big energy producers – most \n |
| 17 | ✓ | decorating with its unique style and atmosphere – I could have sworn it was haunted by kindred spirits – I hope so |
| 18 | ✓ | , you\nthen went on to study for 2 years for 'A' (advanced) level exams – roughly\nequivalent |
| 19 | ? | Triphonon", dedicated to the great composer. On 22 October the album Haris Alexiou - A tribute to |
| 20 | ? | page of the NYISO web site:\nhttp://www.nyiso.com —> Services – -> Committees |
| 21 | ✓ | did.\n\nARLINGTON, Va., May 21, 2012 /PRNewswire/ – Westinghouse |

*Table 14.* **Cluster 161**    size = 60    theme: Em dashes    score: 27 / 30

| # | J | Context |
|---|---|---------|
| 22 | ✓ | cstross\nKey point: for decades in the UK, school education forked at age 16 – the\npoint |
| 23 | ✓ | , Linda J.\nSubject:\tname change\nImportance:\tHigh\n\nLinda – would you please |
| 24 | ✓ | collected eventually?\n\n~~~\nRichardCA\nGood:\n\n \n \n git log – all –graph |
| 25 | ✓ | omial model, you would want to assess whether you have both the variance and the correlation modelled reasonably well – you might need |
| 26 | ✓ | – a\nbig assumption but one I'm willing to play along with for purposes of this\nthread – how about fat |
| 27 | ✓ | of these treatments on sexual satiety development and recovery; additionally flutamide or tamoxifen treatments – alone or together |
| 28 | ✓ | ref2][@ref3]\] Clinical rabies manifests mainly in two forms, encephalitic (furious – more common) |
| 29 | ✓ | <dt>Command</dt>\n <dd>opam install -y – show-action |
| 30 | ✓ | \nclosing).\n \n\n^A&I wanna see the phone glued to your ear!^A8 – Ed Baugh |

*Table 15.* **Cluster 217**    size = 204    theme: Single letters in html tags    score: 27 / 30

| # | J | Context |
|---|---|---------|
| 1 | ✓ | </a>()</div>\n<div class="line"><a name="l00189"></ a >\n<div class="line"><a name="l00212"></ a >\n\t\t\t\t<li><a href="namespace-Monolog.Formatter.html">Formatter</ a >\n\t\t\t\t\t\t\t |
| 4 | ✓ | a href="Ergo.core.Context.html#_preConstruct">_preConstruct</ a ></li> |
| 5 | ✓ | \n\n You et al, 2009[@ b 46-cia |
| 6 | ✓ | \n \n <li><a href="../classes/Plugin Light.html">Plugin Light</ a ></li> |
| 7 | ✓ | vantage.com/HW_52809.GIF width=80 border=0></ a >\n</ |
| 8 | ✓ | Components#isEmpty"><a href="Ergo.core.Components.html#isEmpty">isEmpty</ a ></li> |
| 9 | ✓ | archivado-texto" href="/blog/archivo/#html5">HTML5</ a >\n |
| 10 | ✓ | velocity_verlet">reset</ a ></p></ |
| 11 | ✓ | HREF="package-summary.html" target="_top">NO FRAMES</ A > & |
| 12 | ✓ | "><a href="consumer-int-msg-init-con.html">Browser Init Connection</ a ></li> |
| 13 | ✓ | Bundle.CsrfFormLoginBundle.html">CsrfFormLoginBundle</ a >\n\t\t\t\t\t\t |
| 14 | ✓ | 8</div>\n<div class="line"><a name="l04613"></ a >\n<div class="line"><a name="l01837"></ a >\n Community\n <b class="caret"></ b ></a>< |
| 17 | X | , resulting in endophthalmitis.[@b17-opth-8-2307],[@ b 19-op |

*Table 15.* **Cluster 217**   size = 204   theme: Single letters in html tags   score: 27 / 30

| # | J | Context |
|---|---|---------|
| 18 | ✓ | span class="index-entry-level-1">resizer_type</a></ p ></li> |
| 19 | X | \n//bottom\nfor(; $b_btm < imagesy($img); ++$ b _btm |
| 20 | X | cc_lib::ACMP_GET_TX_CONNECTION_COMMAND_TIMEOUT_ MS },\n |
| 21 | ✓ | level"><a href="data-data-access-base-resource.html">Base Resource</ a ></li> |
| 22 | ✓ | _1_1internal_1_1_internals.html">internal::Internals</ a > I;</ |
| 23 | ✓ | At"><a href="Ergo.core.Components.html#removeAt">removeAt</ a ></li> |
| 24 | ✓ | 160; }</div>\n<div class="line"><a name="l00213"></ a >\n\t\t\t\t\tRequests\t\t\t\t\t</ a >\n\n\t\t\t\t\t\t |
| 26 | ✓ | in org.eclipse.emf.cdo.common">sessions</ a >.</div |
| 27 | ✓ | a href="consumer-int-msg-sub-events.html">Subscribe Conversation Content</ a ></li> |
| 28 | ✓ | > </div>\n<div class="line"><a name="l04599"></ a >Formatting Dates Using Language Resource Bundles</ a >\n |
| 30 | ✓ | >\n <A HREF="../../../../../allclasses-noframe.html">All Classes</ B ></A> |

*Table 16.* **Cluster 411**   size = 226   theme: 'will' in the context of research   score: 21 / 30

| # | J | Context |
|---|---|---------|
| 1 | ? | characteristic makes the ion trap potentially accessible to a wide range of researchers in the biomedical field. This work seeks to achieve ultra |
| 2 | ✓ | DNA from these samples, quality control assessment of DNA samples, sample tracking and sample storage. All samples will have specific sample |
| 3 | ✓ | document.\nWhen the vendor finally exports the data as a PostScript file for printing, the file will refer to the |
| 4 | ✓ | using both a serotonin precursor and inhibitor will determine if serotonin is sufficient to rescue behavioral decline. Our findings will expand our understanding |
| 5 | ? | tool to be used. Whether the state is likely to enforce the values he favors is a question he leaves un-ex |
| 6 | ✓ | doses of either S(-)- or R(+)-nornicotine. In another experiment, rats will be pretreated repeatedly |
| 7 | X | and intra-user stability (with ICC values \>0.90) were selected via ICC analysis. Second , the continuous |
| 8 | ✓ | knack, no one can take it away from you. With this book and CD, your career will benefit for years |
| 9 | ✓ | The recommendations for partitioning of men into those with normal, low and high E1 and E2 levels will be guided by |
| 10 | ✓ | direct role in the neuromuscular pathology of a mouse model of SMA. If successful, these studies will reveal an important |

*Table 16.* **Cluster 411**     size = 226     theme: 'will' in the context of research     score: 21 / 30

| # | J | Context |
|---|---|---------|
| 11 | **X** | the approach below and discuss polynomial kernels.\\nAssume we have a kernel $$K(x,x$ ') $=\langle\$ |
| 12 | **?** | hesus macaques. This is an important tool for the separation of genetic and environmental factors which must occur before satisfactory |
| 13 | ✓ | /c mice, the investigators will track the fate of the antigen-specific T-cells. They will distinguish between an |
| 14 | ✓ | test gives results identical to reference laboratory tests. If Phase I is successful, the lateral-flow devices will be used in |
| 15 | ✓ | an average wage of $28,000 per year.\nD. The Community recognizes that the Project will bring direct and |
| 16 | **X** | proposed as an improved variant of post-treatment system for UASB treating domestic wastewater. This paper evaluates the potential of |
| 17 | **X** | that Pakistan too must now cast off its yoke. They preach hate.\n\nThe traveling circus consists of the leaders |
| 18 | **?** | also hoping to graduate within four years but is concerned it might actually take five.\n\nSSU requires 120 units to |
| 19 | **X** | launch a retrieving dummy, and a solenoid connected to actuate the trigger mechanism. A receiver unit receives signals from a |
| 20 | ✓ | in which the observed burst amplitude will be evaluated as a function of enzyme and substrate concentration. This assay will serve as an |
| 21 | ✓ | of the 2008 and 2009 conferences ($20,000 in each year). Financial support from NIGMS will allow us to |
| 22 | ✓ | tool to link IRB compliance and GCRC processes to strengthen overall human subjects protections. The new database will also keep track |
| 23 | ✓ | , an autosomal recessive human disease due to deficiency of acid neuraminidase. Heterozygote identification will also be made |
| 24 | ✓ | by phosphorus-31 NMR and, collaboratively, by X-ray crystallography. The complexes which will be most vigorously |
| 25 | ✓ | disciplinary actions and a bi-annual cohort survey of 9th and 11th graders. They will also measure the |
| 26 | ✓ | be used suboptimally with inconsistent results. When these clinical and research deficiencies are corrected, passive motion will attain its proper |
| 27 | ✓ | Vt., for $180 million. With the completion of the Vermont Yankee sale, Entergy will own a nuclear |
| 28 | ✓ | immune targeted anti-glioma therapies by inhibiting immune suppressive myeloid-derived cells. The hypothesis will be addressed by |
| 29 | ✓ | HT transporter and 5-HT2A receptor number in smokers, and that these 5-HT markers will down-regulate |
| 30 | ✓ | with hormone levels of tumors and plasma as evaluated by radioimmunoassays. Quantitative immunocyto-chemistry will be used to |

*Table 17.* **Cluster 207**     size = 330     theme: 'be' as part of 'to be'     score: 29 / 30

| # | J | Context |
|---|---|---------|
| 1 | ✓ | my voicemail, attached below is a copy of the Andy Katz article \nwhich is to be substituted for the |
| 2 | ✓ | ~~~\nGoodMalts\nWell, almost. To properly fit in the meter it would have to be three syllables |
| 3 | ✓ | claim it. But he ends up setting fire to the front room, damaging the presents that were to be opened after lunch |
| 4 | ✓ | \n\t\t\t// timeout handle\n\t\t\ttimeoutTimer,\n\n\t\t\t// To know if global events are to be dispatched\n\t\t\t |
| 5 | ? | the end of the first series, Howard and Mel were wed, despite the many mishaps that had be fallen the |
| 6 | ✓ | <\|endoftext\|> and EA2192 has never been the EPA standard method, rather, the method had to be developed in cooperation |
| 7 | ✓ | q, t$, and the limits of the sum are the variables that the closed form would need to be a function of |
| 8 | ✓ | \n\nIn the case of a bat swing, to bat the ball faster, your muscles need to be \nstronger |
| 9 | ✓ | that literate programming has to\nbe tangled and weaved into a different order, in order to be considered "true |
| 10 | ✓ | That discipline will come through the social union framework agreement when we allow ourselves to look at outcomes, to be held accountable to |
| 11 | ✓ | break the normal\nflow.\n\nAlmost all exceptions in for instance the python mysql layer need to be \nhandled |
| 12 | ✓ | : errors of commission \nare much better than errors of omission. H e does want to be personally \n |
| 13 | ✓ | , the internet, global warming, pollution, and cell phones? if the constitution was simply only to be interpreted on a |
| 14 | ✓ | relatively little is known about how the ENS accomplishes these tasks. Serotonin is known to be an important signaling |
| 15 | ✓ | , to establish a diocesan printing house and address office in the town of Aarhus to be used for the |
| 16 | ✓ | monies, or anything that may arise related to the same.\n\nEntrants agree to be bound by these |
| 17 | ✓ | the same address.\n\n<\|endoftext\|>Q:\n\nHow to reverse words?\n\nI need to be able to reverse |
| 18 | ✓ | imbalances on NNG. Terry and Gerry will work with the teams\nto be sure everyone is |
| 19 | ✓ | These approaches have proved, due in part to the high inductance of the vertical yoke, to be expensive, slow |
| 20 | ✓ | stick out in my memory.\n\n#1 - That he told jokes during his lecture. To be wheelchair bound, |
| 21 | ✓ | /Time\nNewOpen_Time - Date/Time\n\nNewOpen_Time is to be updated with the |
| 22 | ✓ | processing of the immature form of IL-1(pro-IL-1). The central hypothesis to be tested is that |
| 23 | ✓ | , and supported employment (totally community developed and rigorously tested outside the categorical psychiatric services) to be two kinds of |
| 24 | ✓ | 82000 a year by a 501(c)3 non-profit; how would YOU like to be paid 82 G |
| 25 | ✓ | you've wanted to go since you were 9 years old, then go. Do the research to be sure you know |

*Table 17.* **Cluster 207**     size = 330     theme: 'be' as part of 'to be'     score: 29 / 30

| # | J | Context |
|---|---|---------|
| 26 | ✓ | as of\nFebruary 28, 2003, UMB Bank represented the value of his retirement assets to  be  $724 |
| 27 | ✓ | the accounts between the parties and thereby render and administer justice between them. The further action here sought to  be  required of the |
| 28 | ✓ | one yet(granted the way Pandora handled it was worse)\n\n* They claim to  be  able to handle |
| 29 | ✓ | evidence has been preserved and provided to defense counsel. As evidence that the government never considered the investigation to  be  over, the |
| 30 | ✓ | . Clinton, by contrast, has the extensive experience that proves that she too lacks the good judgment to  be  President.\n |

*Table 18.* **Cluster 187**     size = 261     theme: The word 'people'     score: 19 / 30

| # | J | Context |
|---|---|---------|
| 1 | ✓ | the loyalty of 20somethings.\nThey've seen large companies lay off hordes of  people  for dumb reasons |
| 2 | ✓ | knowledgeable and interesting people – even some that I learned from a\nlot. Just having a bunch of  people  that I can |
| 3 | ✓ | opposite, that opiate treatment improves their ability to\nthink. Whereas with pot, it seems most  people  are seeking a |
| 4 | X | market first then the retail public will begin 12/21/2001. As long as there are no  problems  that surface with |
| 5 | ✓ | , it was (at age 55) my first time as an inpatient.\n\nI know many  people  who've |
| 6 | X | , very bad," Glenn Hubbard, \nchairman of President Bush's Council of Economic Advis ers , says. |
| 7 | X | \n\n~~~\ncm2187\nAs a heavy VB user, XML literals is the one  thing  I never found |
| 8 | ✓ | finance don't make or sell anything. They're just greedy. Too\nmany of the smartest  people  go into it |
| 9 | X | only 2 black people that applied, but they were both joke applications.\nThere's simply no use-ful  information  you can gather |
| 10 | ✓ | and tons and tons of photographs of family members ranging from great great grandparents to baby pictures of all the  people  who I knew |
| 11 | ✓ | Eye in PPC-3100S rear cover gives a clear signal of system operating status, helping  people  detecting and analysis |
| 12 | ✓ | The system as described empowers the obsessive aggressor far more than\nimproving the lives of  people  who don't |
| 13 | ✓ | . After the book was published, its readers began visiting the hut as well, with 2,000  people  in one year |
| 14 | X | are placed into the drawers, the various tools are often interlaced with each other after many  times  of drawing/ |
| 15 | X | story but I think it's ridiculous that EDF wants us to do them a favour when they dit ched  us when Enron |
| 16 | X | You may be victim of the "usb fuse" issue. I soldered some 1ohm res isters \nacross |

*Table 18.* **Cluster 187**    size = 261    theme: The word 'people'    score: 19 / 30

| # | J | Context |
|---|---|---------|
| 17 | ✓ | in the area, but it seems\nlike there are potentially showstopper questions about how to stop ==people== \ngaming |
| 18 | **X** | . My high regard and respect for the Government Affairs group and for each individual has been validated many ==times== over during |
| 19 | ✓ | "Australia and New Zealand have already taken international rugby abroad to the USA and to Hong Kong, and ==people== will continue to |
| 20 | **X** | is extensive and it is the most common form of browning in heated food systems.\nThere are ==many== reviews of Ma |
| 21 | ✓ | naval intelligence, Jim Morrison from The Doors' dad as\nwell, plus a ton of other ==people== . Tons |
| 22 | ✓ | decline)\n\nRegardless of the technical cause have a feeling this will make things worse\nand ==people== more nervous. |
| 23 | ✓ | it and fix the conventions yourself. Style conventions are arbitrary and meaningless. Reduce the friction here for ==people== who bother to |
| 24 | **?** | be done against the NCGC Pharmacological Compound Collection, which includes several small molecules approved by several regulatory ==bodies== . Animal models |
| 25 | ✓ | streets looking for angels\nEveryone I meet looking for angels\n\nSo many nations with so many hungry ==people== \nSo many |
| 26 | ✓ | you know nobody broke in?\n\nAnd if you're responsible for storing personal data for millions of ==people== in\nyour |
| 27 | ✓ | that judicial power could only be exercised by a Chapter III Court, and that the removal and detention of ==people== was exclusively a |
| 28 | ✓ | of District Human Resources 6368 Mail Service Center Raleigh,<\|endoftext\|>The job\n\nThese days, most ==people== go to work |
| 29 | **X** | a bad way to implement security. Finger prints and\niris scans can easily be stolen many ==times== by just browsing |
| 30 | ✓ | in the US?\nThanks so much for saving him. Its so great that there are such caring ==people== in the world |

*Table 19.* **Cluster 294**    size = 312    theme: Months, seasons or years    score: 27 / 30

| # | J | Context |
|---|---|---------|
| 1 | ✓ | anniversary date (that is, November 25 of 2008, ==2009== , or |
| 2 | ✓ | common stock to unsuspecting investors. \nIf you bought the common stock of Enron between ==January== 18, 2000 |
| 3 | ✓ | crash was speeding\n\nAssociated Press and AP, WTSP12:54 PM. EDT ==June== 19, 2014 |
| 4 | ✓ | woman premier. In NSW, Kristina Keneally became the first female premier of the state in ==2009== and was defeated |

*Table 19.* **Cluster 294**    size = 312    theme: Months, seasons or years    score: 27 / 30

| # | J | Context |
|---|---|---|
| 5 | ✓ | It became the leading information and press body of Soviet public organizations. The constituent conference was held on 21 February 1961. The |
| 6 | ✓ | his own routine for six weeks, but his routine involved very little\nactivity. He had joined the summer sports programme for |
| 7 | ✓ | (53.1 percent), for 2,643 yards, 21 touchdowns and just seven interceptions in 2010 as a senior |
| 8 | X | \n\n<\|endoftext\|>Q:\n\nHow to delete a portlet in Liferay 6.1 program matically from code\n |
| 9 | ✓ | Standard Offer Service load in \nthe State of Maine, covering at least March 1, 2001 - Feb 28, 2002 |
| 10 | ✓ | at the ADMA (Association for Data-Driven Marketing & Advertising) Global Forum in August 2015.\n |
| 11 | ✓ | Bill Cleveland\n\nWilliam Jennings Cleveland, Sr., known as Bill Cleveland (October 19, 1902 – December 16, 1974 |
| 12 | ✓ | necessary to show waiver is present in this case. First, process was served upon Charterers on September 18, 2000 |
| 13 | ✓ | transactions related to a partnership that for a time was headed by Mr. Fastow. \nIn July , Mr. |
| 14 | ✓ | , is lower than during the day. A full statewide nighttime belt use observation survey was conducted in 2004 . This survey |
| 15 | ✓ | 5 May 2008. The first single for the album, "Built to Fail", was released in March 2008 and received |
| 16 | ✓ | Scott, June 7th, accepted another order "under protest, especially as to delivery date." On June 11th, |
| 17 | ✓ | omannan assay (GMA) screening on IPA diagnosis in children. Between January 2010 and December 2011 , all children |
| 18 | ✓ | 2001\nAgence France-Presse \n(Copyright 2001) \n\nLAGOS, Feb 14 (AFP |
| 19 | ✓ | 7.5 million outpatient visits, 640,000 ED visits, and 157,000 hospitalization visits in 2008 . Recent work |
| 20 | ✓ | *Journal for Public Health Management and Practice*devoted two issues entirely to GIS applications. In May , 2003, |
| 21 | ✓ | Bernard Kolélas\n\nBernard Bakana Kolélas (12 June 1933 – 13 November 2009) was |
| 22 | ✓ | \[[@pone.0153192.ref053]\], using images taken by the Landsat 8 satellite during August -October 2013 |
| 23 | X | /clean\n\nYou need the permissions to look like this (provided you'll run the script as root which you probably |
| 24 | X | was simplicity and good proportions.\n\nAntique mirrors and lighting, perennial customer favourites at the Dec orative Fair |
| 25 | ✓ | ache Spreads.\n\n\n15\nPromptly after filing the Cresswell I complaint in March 1983, Swan |
| 26 | ✓ | Excellence Gala, to be held June 23 in New Orleans.\n\nAn advocacy group established in 1975 in Washington, |
| 27 | ✓ | October 23, 1986, but was dismantled during November 15–16, 1986, and stored until May 1989. The |
| 28 | ✓ | it was Guy IV, who had entered Rome along with Lambert and his mother Angiltrude in January 897, |

*Table 19.* **Cluster 294**    size = 312    theme: Months, seasons or years    score: 27 / 30

| # | J | Context |
|---|---|---|
| 29 | ✓ | and in need of being able to communicate fluently in standard American English." – Midwest Book Review, March 2006\n\n |
| 30 | ✓ | several personal losses. She was devastated when her mother, to whom she was close, died suddenly in August 1929. Four |

*Table 20.* **Cluster 316**    size = 201    theme: Open parenthesis for multiple choice answers    score: 29 / 30

| # | J | Context |
|---|---|---|
| 1 | ✓ | the nearest to -0.2? (a) o (b) -5 ( c) j |
| 2 | ✓ | .6 - 13. Which is the nearest to 1/4? (a) 4 ( b) 0 |
| 3 | ✓ | Which is the third smallest value? (a) 0.2 (b) r ( c) y |
| 4 | ✓ | - 0.55. Which is the second smallest value? (a) 0.1 ( b) i |
| 5 | ✓ | is the third smallest value? (a) -0.1 (b) f ( c) j |
| 6 | ✓ | + -2.5. Which is the biggest value? (a) 0.1 ( b) - |
| 7 | ✓ | + 8 = 6*t. Which is the third biggest value? (a) 1 ( b) t |
| 8 | ✓ | -3? (a) -1/4 (b) -2/9 ( c) 2 |
| 9 | ✓ | \nWhat is prob of picking 2 x<\|endoftext\|>/2 (b) -5/3 ( c) - |
| 10 | ✓ | Which is the third biggest value? (a) 5 (b) 2797787 ( c) - |
| 11 | ✓ | Let k = 0.158 + -0.058. Which is the closest to 6? ( a) 5 |
| 12 | ✓ | ) 0.4 (b) -0.3 (c) 8/7 ( d) 4 |
| 13 | ✓ | t - -19. Which is the closest to 1/4? (a) 14 ( b) l |
| 14 | ✓ | ) 4 (f) 69125.7\na\nWhich is the smallest value? ( a) - |
| 15 | ✓ | 2 -<\|endoftext\|>5, -501?\n0.2\nWhich is the smallest value? ( a) - |
| 16 | ✓ | is the nearest to -0.1? (a) 5 (b) 0 ( c) s |
| 17 | ✓ | 6, 595, 0.1?\n0.1\nWhich is the smallest value? ( a) 1 |
| 18 | ✓ | Let w = 4 + k. Which is the closest to w? (a) 1 ( b) 2 |
| 19 | ✓ | (c) 3/707044\nc\nWhich is the fourth smallest value? ( a) 3 |
| 20 | ✓ | )/(-104))/(6/4). Which is the second biggest value? (a) i ( b) 0 |
| 21 | X | Let u(a) = a**2 - 5*a - 1. Calculate z*m ( r) - |
| 22 | ✓ | . Which is the biggest value? (a) 2/5 (b) 3 ( c) f |
| 23 | ✓ | be ((-81)/6)/(-9)*(-10)/3. Which is the third biggest value? ( a) 3 |
| 24 | ✓ | Let d be (-3 + 61/21)*-3. Which is the third biggest value? ( a) 0 |

*Table 20.* **Cluster 316**    size = 201    theme: Open parenthesis for multiple choice answers    score: 29 / 30

| # | J | Context |
|---|---|---------|
| 25 | ✓ | = y - -1.48. Which is the biggest value? (a) r **(** b) - |
| 26 | ✓ | 1\nWhich is the closest to 1? (a) 3 (b) 0 **(** c) 54 |
| 27 | ✓ | + -2. Which is the third biggest value? (a) -1/3 **(** b) i |
| 28 | ✓ | l. Which is the nearest to -2/5? (a) 3/2 **(** b) h |
| 29 | ✓ | 0.2?\n0.2\nWhich is the nearest to -0.1? **(** a) - |
| 30 | ✓ | = 53.26 + -53.29. Which is the closest to 1/4? **(** a) - |

*Table 21.* **Cluster 244**    size = 183    theme: Closing brackets in html    score: 29 / 30

| # | J | Context |
|---|---|---------|
| 1 | ✓ | aging-operations-messaging-conversation.html">Messaging Conversation</a></li **>** \n |
| 2 | ✓ | datatable-recordtype.html">Sortable generated columns</a>\n </li **>** \n |
| 3 | ✓ | title="class in com.model.player">Location</a> getPosition()</pre **>** \n</li |
| 4 | ✓ | \n \n \n </ul>\n </li **>** \n |
| 5 | ✓ |  java.time.LocalDateTime getTime()</pre>\n</li **>** \n</ul |
| 6 | ✓ | /coq/user-contrib/TLC/LibMultiset.glob</li **>** \n < |
| 7 | ✓ | "><a href="Ergo.core.Components.html#each">each</a></li **>** \n \n |
| 8 | ✓ | " height="1" width="1"></td>\n </tr>\n <tr **>** \n < |
| 9 | ✓ | Brocot/Q_to_R.glob</li>\n <li **>** 222 K < |
| 10 | ✓ | object_dom_read.html">Read HTML to data objects</a>\n\t\t\t\t\t\t\t</li **>** \n\t\t\t\t\t\t\t< |
| 11 | ✓ | href="Ergo.core.Context.html#_destroy">_destroy</a></li **>** \n \n |
| 12 | ✓ | ing/cell-border.html">Base style - cell borders</a>\n\t\t\t\t\t\t\t</li **>** \n\t\t\t\t\t\t\t< |
| 13 | ✓ | \thref="../general/creating_libraries.html">Creating Libraries</a></li **>** \n\t\t\t\t\t\t\t< |
| 14 | ✓ | .Bundle.SwiftmailerBundle.Command.html">Command</a>\n\t\t\t\t\t\t\t</li **>** \n\t\t\t\t\t< |
| 15 | X | '/>\n <ul>\n <li>A</li>\n <li **>** B</li |
| 16 | ✓ | autokad\n> "Western PA in general seems to be slipping back into (as I've **been** told it\n |
| 17 | ✓ | a></li>\n</ul>\n<ul class="subNavList">\n<li **>** Detail:&nbsp |
| 18 | ✓ | class="reference internal"\n\t\t\t\t\t\t\t\thref="../general/security.html">Security</a></li **>** \n\t\t\t\t\t\t\t< |
| 19 | ✓ | lib/coq/user-contrib/TLC/LibNat.vo</li **>** \n < |
| 20 | ✓ | ul class="subNavList">\n<li>Summary: </li>\n<li **>** Nested& |
| 21 | ✓ | 00047989">Ogre::OverlayElement</a>\n</li>\n<li **>** _setWindow |

*Table 21.* **Cluster 244**    size = 183    theme: Closing brackets in html    score: 29 / 30

| # | J | Context |
|---|---|---------|
| 22 | ✓ | html?com/history/Triple.html" target="_top">Frames</a></li > \n<li |
| 23 | ✓ | /bootstrap.html">Bootstrap 3</a>\n\t\t\t\t\t\t\t</li>\n\t\t\t\t\t\t\t<li > \n\t\t\t\t\t\t\t< |
| 24 | ✓ | ative_php.html">Alternate PHP Syntax\n\t\t\t\t\t\t\t\tfor View Files</a></li > \n\t\t\t\t\t\t\t< |
| 25 | ✓ | \n</div>\n<div>\n<ul class="subNavList">\n<li > Summary:&nbsp |
| 26 | ✓ | lib/coq/user-contrib/TLC/LibOption.vo</li > \n < |
| 27 | ✓ | "\n\t\t\t\t\t\t\thref="../tutorial/news_section.html">News section</a></li > \n\t\t\t\t\t\t\t< |
| 28 | ✓ | -messaging-interactions-conversations.html">Conversations</a></li > \n |
| 29 | ✓ | BackAndForth.html" target="_top">No Frames</a></li > \n</ul |
| 30 | ✓ | asings.QuadEaseOutBackAndForth.html">Use</a></li > \n<li |

