# OpenReview forum: "Towards Spectroscopy: Susceptibility Clusters in Language Models"
_ICML.cc/2026/Conference — ICML 2026 regular_

### Official Review · Reviewer_SD8B · 2026-02-26

**Soundness:** 4
**Presentation:** 3
**Significance:** 2
**Originality:** 3
**Overall Recommendation:** 4
**Confidence:** 4

**Summary:**

The paper proposes to interpret language model behavior through *susceptibility scores*—scalar values $\chi_{xy}$ that measure the covariance between the activations of specific model components (e.g., attention heads) and the action of upweighting the continuation token $y$ in the context $x$. Susceptibility scores provide a concise representation of what model components “respond to”.  Aggregating these scores across all model components gives a vector token--context pair $(x, y)$.
The authors first decompose of these vectors into a sum over modes of the data distribution (singular vectors of the true distribution $q(y \mid x)$ viewed as a matrix), which factorizes the vector into data-dependent *propensities* (how much a mode explains why $y$ follows $x$) and model-dependent responses (how strongly a component reacts to that mode). The decomposition implies that token pairs which follow their contexts for similar reasons will have similar susceptibility vectors, providing a justification for *clustering* in susceptibility space.
The authors then compute susceptibility vectors for 780,000 token-context pairs from the Pile using Pythia-14M and cluster them into 510 clusters. They validate their findings by comparing the clusters to SAE features from Pythia-70M, finding a 50.8% match rate, and by showing that the cluster structure persists when susceptibilities are recomputed from larger models (up to 1.4B parameters).

**Compliance With Llm Reviewing Policy:**

Affirmed.

**Final Justification:**

The authors addressed my questions, which enabled me to raise my confidence score.

**Key Questions For Authors:**

- As I mentioned above, I am not certain about the usefulness of the 50 % overlap between SAE features and the identified clusters. What overlaps would you consider useful and evidence of a good match?
- I really don’t think this is a large weakness, but I nonetheless want to ensure I understand your plans for applying these methods to larger models. How do you foresee methods like these being reliably applied to and used to interpret larger models?

**Limitations:**

yes

**Strengths And Weaknesses:**

- Soundness:
	- The provided experiments seem thorough, useful, and well-designed.
	- The theoretical results connecting susceptibility scores to mode-level quantities seem well justified and correct.
- Presentation:
	- The presentation is formal and splits the content between the main text and the appendix well.
	- Nonetheless, the content is very dense for someone not familiar with susceptibility scores, and there are a lot of references to Baker et al. (2025) (understandably).
	- The presentation of clusters in the tables, figures, and text is well laid out and easy to follow.
- Significance:
	- I really like how the proposed method very naturally clusters tokens not only based on the token identity but together with the context they appear in.
		- In particular, the theoretical connection between per-token susceptibilities and the ground-truth data distributions is useful, novel, and interpretable.
	- However, I am unsure how this sort of analysis could ever be useful for larger models, simply because of the issues of sampling in the larger parameter spaces.
	- I’m also not sure how interpretable 500 clusters are (or perhaps even more that would arise in larger models). It also doesn’t seem like 500 would be enough to capture an entire large LM.
	- I don’t necessarily find the 50 % overlap with the SAE features convincing; it seems a bit low. This might be because of the SAE features or the clusters, though.
- Originality:
	- Proposes something new based on recently developed methodology
	- I liked the connection to larger-scale models despite the main results being on smaller models.

---

> ### Author Rebuttal · Authors · 2026-03-31
>
> We are grateful for your feedback, which helps clarify and improve the paper.
>
> First, a note on scalability: Since publication, we have found success scaling our methods to larger models.  The key improvements are:
>
> 1) We replaced the row standardization preprocessing step in the paper (Section 3.2) with principal component whitening, which increased the number of discovered clusters.
>
> 2) Using an auxiliary model trained on the existing susceptibilities to generate susceptibilities for a larger set of tokens
> These methods increased the number of clusters found on Pythia-14m from 510 to 77,140, and the number of clusters found on Pythia 1.4B from 249 to 57,236. These new clusters contain more abstract patterns than those studied in the paper under review. Forthcoming work describes these changes and showcases the new results.
>
> With regard to your question about SAE feature overlap, any comparison of features to clusters is inevitably messy and difficult to judge. Our primary goal in making such a comparison was not to benchmark discovering a set percentage of SAE features with clusters. Rather, our goal was to give readers an outside reference to make it easier for them to judge the sorts of things we find in clusters. This establishes that clusters find patterns that are not wildly different to the kinds of patterns SAEs find. Further, the Pythia-70m SAEs are a known quantity to many in our audience, which hopefully lets them serve as a useful point of reference.

---

> > ### Author Rebuttal · Reviewer_SD8B · 2026-03-31
> >
> > Thank you for the response and the interesting follow-up. One follow-up question: Are the many more clusters that were discovered related to the previous ones (do they shatter larger clusters into smaller ones, or are they mostly unrelated?).

---

> > > ### Author Response · Authors · 2026-04-01
> > >
> > > Both. To explain: the original 510 clusters were derived from 780K tokens, and the quoted 77,140 clusters is for 46M tokens.
> > >
> > > Qualitatively, what we see is that the original 510 clusters are present in the new larger set of clusters (in the sense that for a given cluster in the original data, there is typically one or more clusters in the new data whose tokens follow the same pattern) but there are many clusters representing patterns that were not seen in the original clusters (e.g. a cluster of tech brand names).
> > >
> > > As you suggest, for a given cluster in the original data there is sometimes a "shattering" effect where (with more tokens) we see higher resolution sub-patterns (e.g. from newlines in general to newlines specifically in licenses like the GNU license).

---

### Official Review · Reviewer_xZ7Y · 2026-03-12

**Soundness:** 3
**Presentation:** 2
**Significance:** 3
**Originality:** 3
**Overall Recommendation:** 4
**Confidence:** 3

**Summary:**

The paper introduces susceptibility clustering, a method that builds on susceptibility analysis from a previous paper to identify clusters of context-token pairs that share similar "fingerprints" in the model's internal representations. The method is applied to the Pythia 14M model, where the authors identify 510 interpretable clusters with distinct patterns. The paper also shows that these identified clusters are also present in larger Pythia models, suggesting that the method can discover patterns that generalize across model scales. The paper argues that susceptibility clustering can help uncover the internal mechanisms of language models, while also offering a decent percentage of clusters with matching SAE features trained on the same model series.

**Compliance With Llm Reviewing Policy:**

Affirmed.

**Final Justification:**

My questions are all answered by the authors and given the novelty of the paper, my overall positive evaluation remains unchanged

**Key Questions For Authors:**

1. Following up on the scalability concern, how does the method scale with longer context windows? I see the paper mainly focuses on the Pythia model family. Does the method work well with other model families, especially those with longer context windows?

2. Could the discovered patterns potentially guide how to steer the model's generation, e.g., removing harmful generation as in the steering examples from Anthropic's attribution graphs work (Anthropic, 2025)?

3. Can you provide more clarification on the SVD and clustering steps? Specifically, how susceptibility decomposition may be related to the topic model which people use to cluster documents. Does this method exploit any low-rank structure in the probability and susceptibility matrices?

**Strengths And Weaknesses:**

## Strengths

1. Strong Theoretical Grounding: The method is built on the well-established framework of susceptibility analysis, which provides a principled way to measure the influence of model components on specific context-token pairs. This idea connects interpretability to statistical physics response theory.

2. The paper shows with concrete evidence that susceptibility clusters correspond to patterns in the training data like sentence boundaries, HTML syntax etc. This suggests that the method can discover meaningful patterns that are relevant to the model's behavior. It is a strength that the authors cross-validate the clusters with both larger models and with SAE features, which adds credibility to the findings.

In regard to the above point, I also find several weaknesses in the paper, which I will discuss below.

## Weaknesses

1. **Computational complexity concern with scaling context space.**
   The method may suffer from a dramatic increase in the context space ($\Sigma^k$) as the sequence length and vocabulary grow, rendering high computational complexity. In contrast, SAE neurons are typically shared across each sequence position, avoiding this issue (the position-independent could also be a limitation in some cases though).

2. **Analysis restricted to "next token" patterns.**
   The clustering only answers which part of the model is responsible for generating a given context-token pair, and how different context-token pairs share similar "fingerprints." It is unclear whether the method is capable of discovering more sophisticated circuits that drive complex reasoning, which typically requires understanding more refined cross-layer interactions and long-range dependencies in the context (e.g., Anthropic, 2025, "On the Biology of a Large Language Model")

3. **Scalability claim is indirect.**
   The scaling analysis is restricted to showing that clusters detected on Pythia happen to appear in larger models as well. This is different from showing that the method is directly scalable: larger models could contain more refined patterns beyond the small model's capability, and clustering on a small model does not guarantee discovering those patterns.

4. **Presentation could be improved with a notation table.**
   Although the paper and notation seem to be self-contained, it would benefit from a table of notations. Additionally, mathematical definitions (like the Dirac delta) should appear in a separate notation section rather than inline.

---

> ### Author Rebuttal · Authors · 2026-03-31
>
> Thank you for your review and help improving our paper!
>
> First, a note on scalability: Since publication, we have found success scaling our methods to larger models.  The key improvements are:
> 1) We replaced the row standardization preprocessing step in the paper (Section 3.2) with principal component whitening, which increased the number of discovered clusters.
> 2) Using an auxiliary model trained on the existing susceptibilities to generate susceptibilities for a larger set of tokens
> These methods increased the number of clusters found on Pythia-14m from 510 to 77,140, and the number of clusters found on Pythia 1.4B from 249 to 57,236. These new clusters contain more abstract patterns than those studied in the paper under review. Forthcoming work describes these changes and showcases the new results.
>
> To respond specifically to Question 1. Larger context windows would only affect susceptibility estimation insofar as it generally slows down forward and backward passes for the model. These are necessary for running Stochastic Gradient Langevin Dynamics and also to evaluate model losses at each sample produced by SGLD. In other words, susceptibility estimation would be affected by roughly the same proportion as training itself is affected.
>
> In response to the other questions:
>
> Q2) Regarding guiding the model’s generation. It seems possible to take an identified susceptibility cluster, and use the activations of the model on token sequences in that cluster in the usual ways that steering is done (this would roughly be analogous to “steering on the associated SAE feature to a cluster”, if one existed). This has not been tried. In a different direction in “Patterning: The Dual of Interpretability” (Wang et al, 2026) researchers used susceptibilities to intervene during training in a toy model setting. They were able to control the algorithm learned in a bracket matching problem (this is intervening in training, not generation).
>
> Q3) The decomposition of a data distribution into modes (Section 2.3) is akin to topic modeling in that it breaks down a complex probability distribution into key factors which describe it. That said, this analogy is complicated because mode decomposition does not require strong assumptions of topic modeling like the breakdown of text into documents, or the bag-of-words model.  If a corpus was generated according to a topic model, we think it likely that the mode decomposition would recover the topics used.
>
> In response to point 2 of the weaknesses section: Though preliminary, we have made some observations that clusters responsible for different parts of the one larger pattern may be colocated in susceptibility space, indicating that this technique has the ability to detect circuits, as well as characterize individual tokens.  See Appendix F for discussion.
>
> Finally, we genuinely appreciate the feedback that a notation table would have been useful, and will add one to future work

---

### Official Review · Reviewer_ddV9 · 2026-03-13

**Soundness:** 4
**Presentation:** 4
**Significance:** 3
**Originality:** 2
**Overall Recommendation:** 5
**Confidence:** 3

**Summary:**

This work scales a previous approach based on defining susceptibilities of language models as a way of understanding the connection between data structure and what the model has learned. They analyze a larger model and perform clustering and a detailed analysis of the learned susceptibilities.

**Compliance With Llm Reviewing Policy:**

Affirmed.

**Final Justification:**

This empirical study provides a scaled-up version of an interesting approach to interpretability. The ability to scale this method seems to be the main difficulty to overcome. Understanding what can be extracted this way should provide insight into whether this approach can be more broadly useful. The authors have clarified the methodology, so I raised my score as a result.

**Key Questions For Authors:**

1. How robust is your approach to perturbations in hyperparameters for both the susceptibilities and the clustering method? Can you give a general approach to tuning these?
2. This method feels very general. Is it unique to LLMs or is it possible to apply to other models in other domains? If so, have you checked that it gives reasonable results in other applications? Is there an implementation available for others to check?
3. Since susceptibilities are linear measures of correlation, what exactly do they see and what would they potentially miss? How might this affect potential intervention methods based on this idea?
4. How does this approach compare with SAEs beyond the similarities in identified features? Why would you expect them to be similar or to differ?

**Limitations:**

A better description of what exactly susceptibilities capture vs what they do not would be helpful for understanding how much to read into these results.

**Strengths And Weaknesses:**

**Soundness**
The methodology appears sound, and the resulting clusters are individually identified and compared with other approaches like SAEs.

**Presentation**
The paper is well-written and covers the key methodology of susceptibilities and clustering well.

**Significance**
Robust methods of interpreting LLMs and other deep learning models provide key tools for AI safety and just general understanding. This work pursues a promising avenue for such interpretability.

**Originality**
This paper is primarily a study in scaling a prior approach with a careful analysis of a larger model.

---

> ### Author Rebuttal · Authors · 2026-03-31
>
> Thank you for your feedback and questions on our paper!
>
> In response to your questions, as you have numbered them:
>
> 1) Susceptibility hyperparameters are constrained by the difficulty of having an SGLD chain converge in a high dimensional loss landscape. See “From Global to Local: A Scalable Benchmark for Local Posterior Sampling“ (Hitchcock et al, 2025)  for a discussion of the problems involved.  The hyperparameters for the clustering method are robust to perturbation. Tuning them mostly involves determining what size of cluster you are expecting to encounter and then setting the parameters of Page Rank to make sure that a sufficiently large region of the graph is explored.
>
> 2) There is forthcoming work looking at susceptibilities in the context of reinforcement learning models and finds they remain a useful tool for interpretability. We are also working to make the codebase public, and expect to push that change within a month
>
> 3) Susceptibilities only capture first order change of an observable in the direction of a perturbation, and higher order effects may be missed. Any intervention method based on susceptibilities needs to be mindful of this. Since we only measure the effects of an infinitesimal perturbation, we can only Taylor approximate the effects of a real, non-infinitesimal perturbation.
>
> 4) Clusters differ from SAEs in that tokens can easily participate in multiple unrelated SAE features simultaneously, but have a strong limit in how many clusters they can be members of, based on the geometry of susceptibility space. More fundamentally, SAE features are based on activations while susceptibilities are based on model weights which may cause them to find fundamentally different phenomena. We hope in the future to provide a more detailed comparative analysis of what is perceivable with these two tools.

---

> > ### Author Rebuttal · Reviewer_ddV9 · 2026-04-04
> >
> > Thank you, my concerns have been addressed.

---

### Decision · Program_Chairs · 2026-04-30

**Decision:**

Accept (regular)

**Comment:**

This paper builds on the work of Baker et al. (2025), who introduced susceptibility-based clustering for transformers. In this work, the authors scale up the method and apply it to a larger language model (Pythia-14M) to extract 510 clusters ranging from grammatical to code-related structures. While the empirical analysis is thorough, the novelty is somewhat limited given that the core approach has been introduced elsewhere. Furthermore, the reported 50% agreement with sparse autoencoder features leaves open questions about what this overlap actually signifies and whether it should be interpreted as validation or a sign of low concordance. While all three reviewers ultimately recommended acceptance (one “Accept,” two “Weak Accept”), the overall enthusiasm is tempered by questions about the method's practical impact on larger models and its ability to capture complex reasoning.